# On the Complexity of Differentially Private Best-Arm Identification with Fixed Confidence

**Achraf Azize**
Équipe Scool, Univ. Lille, Inria,
CNRS, Centrale Lille, UMR 9189- CRIStAL
F-59000 Lille, France
achraf.azize@inria.fr

**Marc Jourdan**
Équipe Scool, Univ. Lille, Inria,
CNRS, Centrale Lille, UMR 9189- CRIStAL
F-59000 Lille, France
marc.jourdan@inria.fr

**Aymen Al Marjani**
UMPA, ENS Lyon
Lyon, France
aymen.al_marjani@ens-lyon.fr

**Debabrota Basu**
Équipe Scool, Univ. Lille, Inria,
CNRS, Centrale Lille, UMR 9189- CRIStAL
F-59000 Lille, France
debabrota.basu@inria.fr

## Abstract

Best Arm Identification (BAI) problems are progressively used for data-sensitive applications, such as designing adaptive clinical trials, tuning hyper-parameters, and conducting user studies to name a few. Motivated by the data privacy concerns invoked by these applications, we study the problem of *BAI with fixed confidence under $\epsilon$-global Differential Privacy (DP)*. First, to quantify the cost of privacy, we derive a *lower bound on the sample complexity* of any $\delta$-correct BAI algorithm satisfying $\epsilon$-global DP. Our lower bound suggests the existence of two privacy regimes depending on the privacy budget $\epsilon$. In the high-privacy regime (small $\epsilon$), the hardness depends on a coupled effect of privacy and a novel information-theoretic quantity, called the *Total Variation Characteristic Time*. In the low-privacy regime (large $\epsilon$), the sample complexity lower bound reduces to the classical non-private lower bound. Second, we propose AdaP-TT, an $\epsilon$-global DP variant of the Top Two algorithm. AdaP-TT runs in *arm-dependent adaptive episodes* and adds *Laplace noise* to ensure a good privacy-utility trade-off. We derive an asymptotic upper bound on the sample complexity of AdaP-TT that matches with the lower bound up to multiplicative constants in the high-privacy regime. Finally, we provide an experimental analysis of AdaP-TT that validates our theoretical results.

## 1 Introduction

We study the stochastic multi-armed *bandit* problem [LS20], which allows us to reflect on fundamental information-utility trade-offs involved in interactive sequential learning. Specifically, in a bandit problem, a learning *agent* is exposed to interact with $K$ unknown probability distributions $\{\nu_1, \ldots, \nu_K\}$ with bounded expectations, referred as the *reward distributions* (or *arms*). $\boldsymbol{\nu} \triangleq \{\nu_1, \ldots, \nu_K\}$ is called *a bandit instance*. At every step $t > 0$, the agent chooses to interact with one of the reward distributions $\nu_{A_t}$ for an $A_t \in [K]$, and obtains a sample (or *reward*) $r_t$ from it. The goal of the agent can be of two types: (a) maximise the reward accumulated over time, or equivalently to minimise the regret, and (b) to find the reward distribution (or arm) with highest expected reward. The first problem is called the regret-minimisation problem [ACBF02], while the second one is called the *Best Arm*

*Identification (BAI)* problem [KCG16]. In this paper, we focus on the BAI problem, i.e. to compute

$$a^\star \triangleq \arg\max_{a \in [K]} \mathbb{E}_{r \sim \nu_a} [r] \triangleq \arg\max_{a \in [K]} \mu_a. \qquad \text{(BAI)}$$

With its advent in 1950s [Bec54, Bec58] and resurgence in last two decades [MT04, GGL12, JMNB14, KCG16, DKM19], BAI has been extensively studied with different structural assumptions (Fixed-confidence: [JN14]; Fixed-budget: [CL16]; Non-stochastic: [JT16]; Best-of-both-worlds: [AYBG$^+$18]; Linear: [SLM14]). In this paper, we specifically investigate the *Fixed Confidence BAI problem*, in brief FC-BAI, that yields a $\delta$-correct recommendation $\hat{a} \in [K]$ satisfying $\Pr(\hat{a} \neq a^\star) \leq \delta$. FC-BAI is increasingly deployed for different applications, such as clinical trials [AKR21], hyper-parameter tuning [LJD$^+$17], communication networks [LPJ22], online advertisement [CLK$^+$14], crowd-sourcing [ZCL14], user studies [LEHT22], and pandemic mitigation [LVR$^+$19] to name a few. All of these applications often involve the sensitive and personal data of users, which raises serious data privacy concerns [TBD$^+$16], as illustrated in Example 1.

**Example 1** (Adaptive dose finding trial)**.** *In a dose-finding trial, one physician decides $K$ possible dose levels of a medicine based on preliminary studies (Typically, $K \in \{3, \ldots, 10\}$ in practice [AKR21]). At each step $t$, a patient is chosen from a local pool of volunteers and a dose level $a_t \in [K]$ is applied to the patient. Following that, the effectiveness of the dose on the patient, i.e. $r_t \in \mathbb{R}$ is observed. The goal of the physician is to recommend after the trial, which dose level is most effective on average, i.e. the dose level $a^*$ that maximises the expected reward. Here, every application of a dose level and the patient's reaction to it exposes information regarding the medical conditions of the patient. Additionally, at each step $t$ of an adaptive sequential trial, the physician can use an FC-BAI algorithm that observes the previous history of dose levels $\{a_s\}_{s<t}$ and their effectiveness $\{r_s\}_{s<t}$ to decide on the next dose level $a_t$ to test. When releasing the experimental findings of the trial to health authorities, the physician should thoroughly detail the experimental protocol. This includes the dose allocated to each patient $\{a_s\}_{s \leq t}$ and the final recommended dose level $a^\star$. Thus, even if the sequence of reactions to doses $\{r_s\}_{s \leq t}$ is kept secret, publishing the sequence of chosen dose levels $\{a_s\}_{s \leq t}$ and the final recommended dose level $a^\star$ computed using the history can leak information regarding patients involved in the trial.*

This example demonstrates the need for privacy in best-arm identification. In this paper, we investigate *privacy-utility trade-offs for a privacy-preserving algorithm in FC-BAI*. Specifically, we use the celebrated Differential Privacy (DP) [DR14] as the framework to preserve data privacy. DP ensures that an algorithm's output is unaffected by changes in input by a single data point. By limiting the amount of sensitive information that an adversary can deduce from the output, DP renders an individual corresponding to a data point '*indistinguishable*'. A popular way to achieve DP is to inject a calibrated amount of noise, from a Laplace [DR14] or Gaussian distribution [DRS22], into the algorithm. The scale of the noise is set to be proportional to the algorithm's sensitivity and inversely proportional to the privacy budget $\epsilon$. Specifically, we study $\epsilon$*-global DP*, where users trust the centralised decision-maker with access to the raw sensitive rewards. For example, in an adaptive dose-finding trial, the patients trust the physician conducting the trial. Thus, at any time $t$, she has access to all the true history $\{a_s, r_s\}_{s<t}$, and it is her duty to design an algorithm such that *publishing $\{a_s\}_{s \leq t}$ and the recommended optimal dose $a^\star$ obeys $\epsilon$-DP given the sensitive input*, i.e. the effectiveness of the dose levels on the patients $\{r_s\}_{s \leq t}$. We define the notion of $\epsilon$*-global DP* for BAI rigorously in Section 2.

For different settings of bandits, the cost of $\epsilon$-global DP and optimal algorithm design techniques are widely studied in the regret-minimization problem [MT15, TD16, SS19, SS18, NR18, BDT19, AB22]. Recently, a problem-dependent lower bound on regret of stochastic multi-armed bandits with $\epsilon$-global DP and an algorithm matching the regret lower bound is proposed by [AB22]. In contrast, DP is meagerly studied in the FC-BAI problem of bandits [SS19, KNSS21]. Though *efficient* algorithm design in FC-BAI literature is traditionally propelled by deriving tight lower bounds, we do not have any explicit sample complexity lower bound for FC-BAI satisfying $\epsilon$-global DP. Here, by an 'efficient' algorithm, we refer to the FC-BAI algorithms that aim to minimise the expected number of samples required (alternatively called, expected *stopping time*) to find a $\delta$-correct recommendation. Presently, we know neither the minimal cost in terms of sample complexity for ensuring DP in FC-BAI, nor the feasibility of efficient algorithm design to achieve the minimal cost.

Motivated by this gap in the literature, we aim to address two questions:

1. *What is the fundamental hardness of ensuring Differential Privacy in the Best-Arm Identification with fixed confidence problem ($\epsilon$-DP-FC-BAI) in terms of a lower bound on the sample complexity?*

2. *How to design an efficient $\epsilon$-DP-FC-BAI algorithm that achieves the lower bound order-optimally?*

**Our contributions.** These questions have led to the following contributions:

1. *Hardness as lower bounds:* We commence our study by deriving the lower bound on sample complexity (or expected stopping time) of any FC-BAI algorithm ensuring $\epsilon$-global DP, in brief $\epsilon$-**DP-FC-BAI** (Section 3). In Theorem 2, we prove that the sample complexity of $\epsilon$-DP-FC-BAI depends on the minimum of two information-theoretic quantities one that depends on privacy, and the other that originates from the classical sample complexity of non-private FC-BAI [KCG16]. The term dependent on privacy depends on the privacy budget $\epsilon$, and a novel information-theoretic quantity, referred to as the *Total Variation Characteristic Time* $T^*_{\mathrm{TV}}$. $T^*_{\mathrm{TV}}$ depends on the Total Variation (TV) distance between the reward distributions in the bandit problem and corresponding most confusing instance. The lower bound also indicates that, in a similar spirit as the regret minimisation with $\epsilon$-global DP [AB22], there are two regimes of hardness for $\epsilon$-DP-FC-BAI. For lower level of privacy (i.e. higher $\epsilon$), the sample complexity of $\epsilon$-DP-FC-BAI is identical to that of the non-private FC-BAI. But for higher level of privacy (i.e. lower $\epsilon$), the sample complexity depends on $\epsilon$ and $T^*_{\mathrm{TV}}$.

2. *Algorithm design:* Following the lower bounds, we aim to design an efficient $\epsilon$-DP-FC-BAI algorithm that simultaneously achieves the lower bound order-optimally and is computationally efficient (Section 4). Due to the superior empirical performance and computational efficiency of Top Two algorithms, we design an $\epsilon$-DP variant of Top Two algorithms, named AdaP-TT. Specifically, we show two simple design techniques, i.e. adaptive episodes for each arm and Laplacian mechanism, if properly used, lead to AdaP-TT from the non-private TTUCB [JD22]. We further derive an asymptotic (as $\delta \to 0$) upper bound on the sample complexity of AdaP-TT (Theorem 5). In the high-privacy regime, the sample complexity upper bound of AdaP-TT coincides with the lower bound up to multiplicative constants. Thus, it is an order-optimal $\epsilon$-DP-FC-BAI algorithm in this regime, whereas the looseness in the low-privacy regime is as same as the looseness of the non-private Top Two algorithm. In Section 5, we experimentally show that AdaP-TT is more sample efficient than the existing $\epsilon$-DP-FC-BAI algorithm, i.e. DP-SE [SS19]. We also show that its sample complexity is independent of the privacy budget in the low-privacy regime, as already indicated by the lower bound.

3. *Technical tools:* (a) To derive the lower bound, we provide an $\epsilon$-global DP version of the transportation lemma [KK13] (Lemma 1), which we prove using a sequential coupling argument. We also define and study the TV characteristic time ($T^\star_{\mathrm{TV}}$) quantifying the hardness of FC-BAI in high privacy regimes (Proposition 1). (b) To design the algorithm, we propose a *generic wrapper, which adapts the existing FC-BAI algorithm to tackle DP-FC-BAI.* It builds on two components: (i) Adaptive episodes with per-arm doubling and forgetting, (ii) A private GLR stopping rule obtained by plugging in private empirical means in the non-private GLR stopping rule used of FC-BAI with Gaussian distributions. To use this proposed wrapper, one can choose among the numerous existing sampling rules to tackle FC-BAI. In this work, we consider the Top Two algorithms since they have good theoretical guarantees and empirical performance. To provide the sample complexity upper bound, we study step-by-step the effects of doubling, forgetting, and adding noise on the performance of the algorithm. Building on [JDB+22], we provide a generic analysis of the class of Top Two algorithms when combined with our wrapper.

## 1.1 Related works

**Lower bound.** Efficient algorithm design in BAI literature is propelled by the derivation of lower bounds on sample complexity. [KCG16] derive the first lower bounds for classical fixed-confidence BAI setting without privacy, which is further improved in [GK16] by introducing KL characteristic time $T^\star_{\mathrm{KL}}$ (Corollary 1). Motivated by this, *we prove the first-known lower bound on sample complexity for FC-BAI with $\epsilon$-global DP* (Theorem 2). The proof employs a similar sequential coupling argument as in the regret lower bound for bandits with $\epsilon$-global DP [SS18, AB22]. This similarity is also reflected in the existence of two privacy regimes depending on the privacy budget $\epsilon$. And for both lower bounds, the Total Variation (TV) appears to be the information-theoretic measure that captures the hardness in the high privacy regime. Specifically, the TV characteristic time ($T^\star_{\mathrm{TV}}$, Corollary 1) serves as the BAI counterpart to the TV-distinguishability gap ($t_{\mathrm{inf}}$) in the problem-dependent regret lower bound for bandits with $\epsilon$-global DP as in [AB22, Theorem 3].

**Algorithms for BAI with fixed confidence (FC-BAI).** The optimal sample complexity for the non-private FC-BAI problem (i.e., $T_{KL}^\star$) is well-understood [GK16], and algorithms are proposed with the aim to achieve this lower bound. Early approaches involved Successive Elimination (SE) based algorithms [EDMMM06] with uniform sampling to find the optimal arm. Inspired by the success of Upper Confidence Bound (UCB) algorithms in the regret setting, the Lower Upper Confidence Bound (LUCB) algorithm was proposed [GGL12]. However, neither SE nor LUCB algorithms achieve asymptotic optimality. The Track-and-Stop (TnS) algorithm introduced in [GK16] was the first to asymptotically achieve the exact optimal sample complexity $T_{KL}^\star$. TnS attains asymptotic optimality by solving a plug-in estimate of the lower bound optimization problem at each step. The game-based approach presented in [DKM19] relaxes this requirement by casting the optimization problem as an unknown game and proposing sampling rules based on iterative strategies to estimate and converge to its saddle point. Finally, the Top Two algorithms arose as an identification strategy based on the praised Thompson Sampling algorithm for regret minimization [Rus16]. In recent years, numerous variants have been analyzed and shown to be asymptotically near optimal [JDB+22]. At every step, a Top Two sampling rule selects the next arm to sample from among two candidate arms, a leader and a challenger. In addition to their great empirical performance, and easy implementation compared to TnS and Game-based algorithms, the Top Two algorithms achieve near asymptotic optimality. In Sec. 4, *we derive an $\epsilon$-global DP version of a Top Two algorithm:* AdaP-TT.

**$\epsilon$-DP BAI algorithms.** DP-SE [SS19] is an $\epsilon$-global DP version of the Successive Elimination algorithm. Although the algorithm was proposed and analysed for the regret minimisation setting in [SS19], it is possible to derive a sample complexity from the analysis in [SS19]. We compare in-depth DP-SE and AdaP-TT, both theoretically (Section 4) and experimentally (Section 5). *In both aspects, our proposed algorithm* AdaP-TT *outperforms DP-SE*. Another adaptation of DP-SE, namely DP-SEQ, is proposed in [KNSS21] for the problem of privately finding the arm with the highest quantile at a fixed level. But this is a different setting of interest than the present paper. [RBCS23] also studies privacy for BAI under fixed confidence but with multiple agents. They propose and analyse the sample complexity of DP-MASE, a multi-agent version of DP-SE. They show that multi-agent collaboration leads to better sample complexity than independent agents, even under privacy constraints. While the multi-agent setting with federated learning allows tackling large-scale clinical trials taking place at several locations simultaneously, we study the single-agent setting, which is relevant for many small-scale clinical trials (see Example 1).

## 2  Differential privacy and best arm identification

**Background: Differential Privacy (DP).** DP ensures protection of an individual's sensitive information when her data is used for analysis. A randomised algorithm satisfies DP if the output of the algorithm stays almost the same, regardless of whether any single individual's data is included in or excluded from the input. This is achieved by adding controlled noise to the algorithm's output.

**Definition 1** (($\epsilon, \delta$)-DP [DR14])**.** *A randomised algorithm $\mathcal{A}$ satisfies ($\epsilon, \delta$)-Differential Privacy (DP) if for any two neighbouring datasets $\mathcal{D}$ and $\mathcal{D}'$ that differ only in one entry, i.e. $d_{Ham}(\mathcal{D}, \mathcal{D}') = 1$, and for all sets of output $\mathcal{O} \subseteq \mathrm{Range}(\mathcal{A})$,*

$$\Pr[\mathcal{A}(\mathcal{D}) \in \mathcal{O}] \leq e^\epsilon \Pr[\mathcal{A}(\mathcal{D}') \in \mathcal{O}] + \delta, \tag{1}$$

*where the probability space is over the coin flips of the mechanism $\mathcal{A}$, and for some $(\epsilon, \delta) \in \mathbb{R}^{\geq 0} \times \mathbb{R}^{\geq 0}$. If $\delta = 0$, we say that $\mathcal{A}$ satisfies $\epsilon$-DP. A lower privacy budget $\epsilon$ implies higher privacy.*

The Laplace mechanism [DNPR10, DR14] ensures $\epsilon$-DP by injecting controlled random noise into the output of the algorithm, which is sampled from a calibrated Laplace distribution (as specified in Theorem 1). We use $Lap(b)$ to denote the Laplace distribution with mean 0 and variance $2b^2$.

**Theorem 1** ($\epsilon$-DP of Laplace mechanism (Theorem 3.6, [DR14]))**.** *Let us consider an algorithm $f : \mathcal{X} \to \mathbb{R}^d$ with sensitivity $s(f) \triangleq \max\limits_{\mathcal{D}, \mathcal{D}' \, s.t \, |\mathcal{D} - \mathcal{D}'|_{\mathrm{Hamming}} = 1} |f(\mathcal{D}) - f(\mathcal{D}')|1$. Here, $\|\cdot\|_1$ is the $L_1$ norm on $\mathbb{R}^d$. If $d$ noise samples $\{N_i\}_{i=1}^d$ are generated independently from $Lap\left(\frac{s(f)}{\epsilon}\right)$, then the output injected with the noise, i.e. $f(\mathcal{D}) + [N_1, \ldots, N_d]$, satisfies $\epsilon$-DP.*

**Background: BAI with fixed confidence.** Now, we describe the canonical best-arm identification problem with fixed confidence (*FC-BAI*). BAI is a variant of pure exploration, where the goal is to

---
**Algorithm 1** Sequential interaction between a BAI strategy and users
---
1: **Input:** A BAI strategy $\pi = (S_t, \text{Rec}_t)_{t \geq 1}$ and Users $\{u_t\}_{t \geq 1}$ represented by the table $\underline{\mathbf{d}}$
2: **Output:** A stopping time $\tau$, a sequence of samples actions $\underline{a}^\tau = (a_1, \ldots, a_\tau)$ and a recommendation $\hat{a}$ satisfying $\epsilon$-global DP
3: **for** $t = 1, \ldots$ **do**
4: $\quad \pi$ recommends action $a_t \sim S_t(. \mid a_1, r_1, \ldots, a_{t-1}, r_{t-1})$
5: $\quad$ **if** $a_t = \top$ **then**
6: $\quad\quad$ Halt. Return $\tau = t$ and $\hat{a} \sim \text{Rec}_t(. \mid a_1, r_1, \ldots, a_{t-1}, r_{t-1})$
7: $\quad$ **else**
8: $\quad\quad u_t$ sends the **sensitive** reward $r_t \triangleq \underline{\mathbf{d}}_{t,a_t}$ to $\pi$
9: $\quad$ **end if**
10: **end for**
---

identify the optimal arm. In FC-BAI, the learner is provided with a confidence level $1 - \delta \in (0, 1)$ [1]. Learner aims to recommend an arm that is optimal with probability at least $1 - \delta$, while using as few samples as possible. To achieve this, the learner defines a FC-BAI strategy to interact with the bandit instance $\boldsymbol{\nu} = \{\nu_a : a \in [K]\}$. We denote the action played at step $t$ by $a_t$, and the corresponding observed reward by $r_t \sim \nu_{a_t}$. $\mathcal{H}_t = (a_1, r_1, \ldots, a_t, r_t)$ is the history of actions played and rewards collected until time $t$. We augment the action set by a *stopping action* $\top$, and write $a_t = \top$ to denote that the algorithm has stopped before step $t$. A FC-BAI strategy $\pi$ is composed of

**i. A pair of sampling and stopping rules** $(S_t : \mathcal{H}_{t-1} \to \mathcal{P}([|1, K|] \cup \{\top\}))_{t \geq 1}$. For an action $a \in [K]$, $S_t(a \mid \mathcal{H}_{t-1})$ denotes the probability of playing action $a$ given history $\mathcal{H}_{t-1}$. On the other hand, $S_t(\top \mid \mathcal{H}_{t-1})$ is the probability of the algorithm halting given $\mathcal{H}_{t-1}$. For any history $\mathcal{H}_{t-1}$, a consistent sampling and stopping rule $S_t$ satisfies $S_t(\top \mid \mathcal{H}_{t-1}) = 1$ if $\top$ has been played before $t$.

**ii. A recommendation rule** $(\text{Rec}_t : \mathcal{H}_{t-1} \to \mathcal{P}([|1, K|]))_{t>1}$. A recommendation rule dictates $\text{Rec}_t(a \mid \mathcal{H}_{t-1})$, i.e. the probability of returning action $a$ as a guess for the best action given $\mathcal{H}_{t-1}$.

We denote by $\tau$ the **stopping time** of the algorithm, i.e. the first step $t$ demonstrating $a_t = \top$. A BAI strategy $\pi$ is called $\delta$-**correct** for a class of bandit instances $\mathcal{M}$, if for every instance $\boldsymbol{\nu} \in \mathcal{M}$, $\pi$ recommends the optimal action $a^\star(\boldsymbol{\nu}) = \arg\max_{a \in [K]} \mu_a$ with probability at least $1 - \delta$, i.e. $\mathbb{P}_{\boldsymbol{\nu}, \pi}(\tau < \infty, \hat{a} = a^\star(\boldsymbol{\nu})) \geq 1 - \delta$.

**FC-BAI with $\epsilon$-global DP ($\epsilon$-DP-FC-BAI).** Now, we formally define $\epsilon$-global DP for FC-BAI, where the BAI strategy (a.k.a. the centralised decision maker) is trusted with all the intermediate rewards. We represent each user $u_t$ by the vector $\mathbf{x}_t \triangleq (x_{t,1}, \ldots, x_{t,K}) \in \mathbb{R}^K$, where $x_{t,a}$ represents the **potential** reward observed, if action $a$ was recommended to user $u_t$. Due to the bandit feedback, only $r_t = x_{t,a_t} \sim \nu_{a_t}$ is observed at step $t$. We use an underline to denote any sequence. Thus, we denote the sequence of sampled actions until $T$ as $\underline{a}^T = (a_1, \ldots, a_T)$. We further represent a set of users $\{u_t\}_{t=1}^T$ until $T$ by **the table of potential rewards** $\underline{\mathbf{d}}^{\mathbf{T}} \triangleq \{\mathbf{x}_1, \ldots, \mathbf{x}_T\} \in (\mathbb{R}^K)^T$. First, we observe that $\underline{\mathbf{d}}^T$ is the sensitive input generated through interaction with the users, and $(\underline{a}^T, \hat{a}, T)$ is the output of the BAI strategy. Hence, we define the probability that the BAI strategy $\pi$ samples the action sequence $\underline{a}^T$, recommends the action $\hat{a}$, and halts at time $T$, as

$$\pi(\underline{a}^T, \hat{a}, T \mid \underline{\mathbf{d}}^T) \triangleq \text{Rec}_{T+1}(\hat{a} \mid \mathcal{H}_T) S_{T+1}(\top \mid \mathcal{H}_T) \prod_{t=1}^T S_t(a_t \mid \mathcal{H}_{t-1}) \tag{2}$$

where $T$ users under interaction are represented by the table of potential rewards $\underline{\mathbf{d}}^T$

Thus, a BAI strategy satisfies $\epsilon$-global DP if the probability defined in Eq. (2) is similar when the BAI strategy interacts with two neighbouring tables of rewards differing by a user (i.e. a row in $\underline{\mathbf{d}}^T$).

**Definition 2** ($\epsilon$-global DP for BAI). *A BAI strategy satisfies $\epsilon$-**global DP**, if for all $T \geq 1$, all neighbouring table of rewards $\underline{\mathbf{d}}^T$ and $\underline{\mathbf{d}}'^T$, i.e. $d_{Ham}(\underline{\mathbf{d}}^T, \underline{\mathbf{d}}'^T) = 1$, all sequences of sampled actions $\underline{a}^T \in [K]^T$ and recommended actions $\hat{a} \in [K]$ we have that*

$$\pi(\underline{a}^T, \hat{a}, T \mid \underline{\mathbf{d}}^T) \leq e^\epsilon \pi(\underline{a}^T, \hat{a}, T \mid \underline{\mathbf{d}}'^T).$$

---

[1]We remind not to confuse risk level $\delta$ with the $\delta$ of $(\epsilon, \delta)$-DP. Hereafter, we consider $\epsilon$-global DP as the privacy definition, and $\delta$ always represents the risk (or probability of mistake) of the BAI strategy.

Definition 2 can be seen as a BAI counterpart of the $\epsilon$-global DP definition proposed in [AB22] for regret minimization. We demonstrate the BAI strategy-Users interaction in Algorithm 1.

**Remark 1.** *It is possible to consider that the output of a BAI strategy is only the final recommended action $\hat{a}$, i.e. not publishing the intermediate actions $\underline{a}^T$. This gives a weaker definition of privacy compared to Definition 2, since the latter defends against adversaries that may look inside the execution of the BAI strategy, i.e. pan-privacy [DNP$^+$10]. In addition, Definition 2 is needed in practice. For example, in the case of dose-finding (Example 1), the experimental protocol, i.e. the intermediate actions, needs to be published too.*

**The goal** in $\epsilon$-DP-FC-BAI is to design a $\delta$-correct $\epsilon$-global DP algorithm, with $\tau$ as small as possible.

## 3 Lower bound on sample complexity for FC-BAI with $\epsilon$-global DP

The central question that we address in this section is

*How many additional samples a BAI strategy must select for ensuring $\epsilon$-global DP?*

In response, we prove a lower bound on the sample complexity of any $\delta$-correct $\epsilon$-DP BAI strategy. Our lower bound features problem-dependent characteristic times reminiscent of the FC-BAI setting.

Let $\boldsymbol{\nu} \triangleq \{\nu_a : a \in [K]\}$ be a bandit instance, consisting of $K$ arms with finite means $\{\mu_a\}_{a \in [K]}$. Now, we define the set of alternative instances as $\mathrm{Alt}(\boldsymbol{\nu}) \triangleq \{\boldsymbol{\lambda} : a^\star(\boldsymbol{\lambda}) \neq a^\star(\boldsymbol{\nu})\}$, i.e. the bandit instances with a different optimal arm than $\boldsymbol{\nu}$. For two probability distributions $\mathbb{P}, \mathbb{Q}$ on the same measurable space $(\Omega, \mathcal{F})$, the Total Variation (TV) distance is defined as $\mathrm{TV}(\mathbb{P} \| \mathbb{Q}) \triangleq \sup_{A \in \mathcal{F}}\{\mathbb{P}(A) - \mathbb{Q}(A)\}$, while the KL divergence (or relative entropy) is $\mathrm{KL}(\mathbb{P} \| \mathbb{Q}) \triangleq \int \log\left(\frac{d\mathbb{P}}{d\mathbb{Q}}(\omega)\right) d\mathbb{P}(\omega)$, when $\mathbb{P} \ll \mathbb{Q}$, and $\infty$ otherwise. We denote the probability simplex by $\Sigma_K \triangleq \{\omega \in [0,1]^K : \sum_{a=1}^K \omega_a = 1\}$.

First, we derive an $\epsilon$-global DP variant of the 'transportation' lemma, i.e. Lemma 1 in [KCG16].

**Lemma 1** (Transportation lemma under $\epsilon$-global DP). *Let $\delta \in (0,1)$ and $\epsilon > 0$. Let $\boldsymbol{\nu}$ be a bandit instance and $\lambda \in \mathrm{Alt}(\boldsymbol{\nu})$. For any $\delta$-correct $\epsilon$-global DP BAI strategy, we have that*

$$6\epsilon \sum_{a=1}^K \mathbb{E}_{\boldsymbol{\nu}, \pi}[N_a(\tau)] \, \mathrm{TV}(\nu_a \| \lambda_a) \geq \mathrm{kl}(1-\delta, \delta),$$

*where $\mathrm{kl}(1-\delta, \delta) \triangleq x \log \frac{x}{y} + (1-x)\log\frac{1-x}{1-y}$ for $x, y \in (0,1)$.*

*Proof sketch.* We use Sequential Karwa-Vadhan Lemma [AB22, Lemma 2] with a data-processing inequality in the BAI canonical model. Extra care is needed *to deal with the stopping times in the coupling, compared to a fixed horizon $T$ in regret minimization*. The proof is deferred to Appendix B.

Leveraging Lemma 1, we derive a sample complexity lower bound for any $\epsilon$-DP-FC-BAI strategy.

**Theorem 2** (Sample complexity lower bound for $\epsilon$-DP-FC-BAI). *Let $\delta \in (0,1)$ and $\epsilon > 0$. For any $\delta$-correct $\epsilon$-global DP BAI strategy, we have that*

$$\mathbb{E}_{\boldsymbol{\nu}}[\tau] \geq T^\star(\boldsymbol{\nu}; \epsilon) \log(1/3\delta), \tag{3}$$

*where $(T^\star(\boldsymbol{\nu}; \epsilon))^{-1} \triangleq \sup_{\omega \in \Sigma_K} \inf_{\boldsymbol{\lambda} \in \mathrm{Alt}(\boldsymbol{\nu})} \min\left(\sum_{a=1}^K \omega_a \mathrm{KL}(\nu_a \| \lambda_a), 6\epsilon \sum_{a=1}^K \omega_a \mathrm{TV}(\nu_a \| \lambda_a)\right)$.*

**Comments on the lower bound.** Similar to the lower bound for the non-private BAI [GK16], the lower bound of Theorem 2 is the value of a two-player zero-sum game between a MIN player and MAX player. MIN plays an alternative instance $\lambda$ close to $\nu$ in order to confuse MAX. The latter plays an allocation $\omega \in \Sigma_K$ to explore the different arms, with the purpose of maximising the divergence between $\nu$ and the confusing instance $\lambda$ that MIN played. On top of the KL divergence present in the non-private lower bound, our bound features the TV distance that appears naturally when incorporating the $\epsilon$-global DP constraint. The proof is deferred to Appendix B. In order to compare the lower bound of an $\epsilon$-global BAI strategy with the non-private lower bound of [GK16], we relax Theorem 2 to further derive a simpler bound, as in Corollary 1.

**Corollary 1.** *For any $\delta$-correct $\epsilon$-global DP BAI strategy, we have that*

$$\mathbb{E}_{\boldsymbol{\nu}}[\tau] \geq \max\left(T^\star_{\mathrm{KL}}(\boldsymbol{\nu}), \frac{1}{6\epsilon} T^\star_{\mathrm{TV}}(\boldsymbol{\nu})\right) \log(1/3\delta),$$

*where $(T^\star_{\boldsymbol{d}}(\boldsymbol{\nu}))^{-1} \triangleq \sup_{\omega \in \Sigma_K} \inf_{\boldsymbol{\lambda} \in \mathrm{Alt}(\boldsymbol{\nu})} \sum_{a=1}^K \omega_a \boldsymbol{d}(\nu_a, \lambda_a)$, and $\boldsymbol{d}$ is either KL or TV.*

*Proof.* The proof is direct by observing that $T^\star(\boldsymbol{\nu}; \epsilon) \geq T^\star_{\mathrm{KL}}(\boldsymbol{\nu})$ and $T^\star(\boldsymbol{\nu}; \epsilon) \geq \frac{1}{6\epsilon} T^\star_{\mathrm{TV}}(\boldsymbol{\nu})$. □

**Comparison with the non-private lower bound.** $T^\star_{\mathrm{KL}}$ is the characteristic time in the non-private lower bound [GK16], and we refer to Section 2.2 of [GK16] for a detailed discussion on its properties. The sample complexity lower bound suggests the existence of *two hardness regimes depending on $\epsilon$*, $T^\star_{KL}$ and $T^\star_{\mathrm{TV}}$. (1) *Low-privacy regime*: When $\epsilon > T^\star_{\mathrm{TV}(\boldsymbol{\nu})}/(6T^\star_{\mathrm{KL}}(\boldsymbol{\nu}))$, the lower bound retrieves the non-private lower bound, i.e. $T^\star_{\mathrm{KL}}(\boldsymbol{\nu})$, and thus, **privacy can be achieved for free**. (2) *High-privacy regime:* When $\epsilon < T^\star_{\mathrm{TV}}(\boldsymbol{\nu})/(6T^\star_{\mathrm{KL}}(\boldsymbol{\nu}))$, the lower bound becomes $T^\star_{\mathrm{TV}}/(6\epsilon)$ and $\epsilon$-global DP $\delta$-BAI requires more samples than non-private ones.

In the following proposition, we characterise $T^\star_{\mathrm{TV}}$ for Bernoulli instances.

**Proposition 1** (TV characteristic time for Bernoulli instances)**.** *Let $\nu$ be a bandit instance, i.e. such that $\nu_a = Bernoulli(\mu_a)$ and $\mu_1 > \mu_2 \geq \cdots \geq \mu_K$. Let $\Delta_a \triangleq \mu_1 - \mu_a$ and $\Delta_{min} \triangleq \min_{a \neq 1} \Delta_a$. We have that*

$$T^\star_{\mathrm{TV}}(\boldsymbol{\nu}) = \frac{1}{\Delta_{min}} + \sum_{a=2}^{K} \frac{1}{\Delta_a}, \qquad \text{and} \qquad \frac{1}{\Delta_{min}} \leq T^\star_{\mathrm{TV}}(\boldsymbol{\nu}) \leq \frac{K}{\Delta_{min}}.$$

*Proof sketch.* The proof is direct by solving the optimisation problem defining $T^\star_{\mathrm{TV}}$ and using that $\mathrm{TV}\,(Bernoulli(p) \,\|\, Bernoulli(q)) = |p - q|$. We refer to Appendix B for details.

*Comment.* The aforementioned bound on TV characteristic time for Bernoulli instances is $\epsilon$-global DP parallel of the KL-characteristic time bound $T^\star_{\mathrm{KL}}(\boldsymbol{\nu}) \leq \sum_{a=1}^{K} \Delta_a^{-2}$ [GK16]. Using Pinsker's inequality, one can connect the TV and KL characteristic times by $T^\star_{\mathrm{TV}}(\boldsymbol{\nu}) \geq \sqrt{2T^\star_{\mathrm{KL}}(\boldsymbol{\nu})}$.

# 4 Algorithm design: Private Top Two with adaptive episodes (AdaP-TT)

In this section, we propose AdaP-TT, an $\epsilon$-global DP version of the TTUCB algorithm [JD22]. We show that AdaP-TT satisfies $\epsilon$-global DP, is $\delta$-correct, and has an asymptotic sample complexity that matches the high privacy lower bounds up to multiplicative constants.

**TTUCB** belongs to the family of Top Two algorithms [Rus16, SHM+20, JDB+22], which selects at each time two arms called leader and challenger, and sample among them. After initialisation, TTUCB uses a UCB-based leader and a Transportation Cost (TC) challenger, expressed by

$$B_n = \arg\max_{a \in [K]} \{\hat{\mu}_{n,a} + \sqrt{6 \log(n)/N_{n,a}}\}, \quad \text{and} \quad C_n = \arg\min_{a \neq B_n} \frac{\hat{\mu}_{n,B_n} - \hat{\mu}_{n,a}}{\sqrt{1/N_{n,B_n} + 1/N_{n,a}}}.$$

Here, $(\hat{\mu}_n, N_n)$ are the empirical means and counts on the whole history. The theoretical motivation behind the TC challenger comes from the theoretical lower bound in FC-BAI, which involves the KL-characteristic time $T^\star_{\mathrm{KL}}(\boldsymbol{\mu}) = \min_{\beta \in (0,1)} T^\star_{\mathrm{KL},\beta}(\boldsymbol{\mu})$. For Gaussian distributions, we have

$$2T^\star_{\mathrm{KL},\beta}(\boldsymbol{\mu})^{-1} = \max_{\omega \in \Sigma_K, \omega_{a^\star} = \beta} \frac{(\mu_{a^\star} - \mu_a)^2}{1/\beta + 1/\omega_a} \quad \text{and} \quad T^\star_{\mathrm{KL},1/2}(\boldsymbol{\mu}) \leq 2T^\star_{\mathrm{KL}}(\boldsymbol{\mu}),$$

The maximiser of the above equation is denoted by $\omega^\star_{\mathrm{KL},\beta}(\boldsymbol{\mu})$, and is further referred to as the $\beta$-optimal allocation as it is unique. Let $N^a_{n,b}$ denote the number of times arm $b$ was pulled when $a$ was the leader, and $L_{n,a}$ denotes the number of times arm $a$ was the leader. In order to select the next arm to sample $I_n$, TTUCB relies on $K$ tracking procedures, i.e. set $I_n = B_n$ if $N^{B_n}_{n,B_n} \leq \beta L_{n+1,B_n}$, else $I_n = C_n$. This ensures that $\max_{a \in [K], n > K} |N^a_{n,a} - \beta L_{n,a}| \leq 1$. Standing on this premise, we now describe how we design an $\epsilon$-global DP extension of TTUCB.

**Private algorithm design.** As illustrated in Algorithm 2, AdaP-TT relies on three ingredients: *adaptive episodes with doubling*, *forgetting*, and *adding calibrated Laplacian noise*. (1) AdaP-TT maintains $K$ episodes, i.e. one per arm. The private empirical estimate of the mean of an arm is only updated at the end of an episode, that means when the number of times that a particular arm was played doubles (Line 5). (2) For each arm $a$, AdaP-TT forgets rewards from previous phases of arm $a$, i.e. the private empirical estimate of arm $a$ is only computed using the rewards collected in the last phase of arm $a$ (Line 8). This assures that the means of each arm are estimated using a non-overlapping sequence of rewards. (3) Thanks to this *doubling* and *forgetting*, AdaP-TT is $\epsilon$-global DP as soon as each empirical mean (Line 9) is made $\epsilon$-DP, and thus, avoiding any use of privacy composition. This is achieved by adding Laplace noise. We formalise this intuition in Lemma 2 of Appendix C.

**Remark 2.** *The aforementioned generic wrapper can be used to construct a near-optimal differentially private version of any existing FC-BAI algorithm that deploys a sampling rule with the empirical means of rewards. In this work, we consider and rigorously analyse the Top Two algorithms since they demonstrate both good theoretical guarantees and empirical performance.*

**Theorem 3** (Privacy analysis)**.** *For rewards in $[0, 1]$,* AdaP-TT *satisfies $\epsilon$-global DP.*

*Proof sketch.* A change in one user *only affects* the empirical mean calculated at one episode of an arm, which is made private using the Laplace Mechanism and Lemma 2. Since the sampled actions, recommended action, and stopping time are computed only using the private empirical means, AdaP-TT satisfies $\epsilon$-global DP thanks to post-processing lemma. We refer to Appendix C for details.

**Private GLR stopping rule.** We consider the private GLR stopping rule based on the private means and on the pulling counts from the last phase (Line 12), and recommend the arm with the highest private mean (Line 11). Lemma 4 yields a threshold function ensuring that any sampling rule is $\delta$-correct, when using the private GLR stopping rule.

**Theorem 4** ($\delta$-correctness)**.** *Let $\delta \in (0, 1)$, $\epsilon > 0$. Let $s > 1$ and $\zeta$ be the Riemann $\zeta$ function. Let $c_k(n, m, \delta) = 2\mathcal{C}_G(\log((K-1)\zeta(s)k^s/\delta)/2) + 2\log(4 + \log n) + 2\log(4 + \log m)$ be the threshold without privacy. Given any sampling rule, the following threshold*

$$c_{\epsilon,k_1,k_2}(n, m, \delta) = 2c_{k_1 k_2}(n, m, \delta/2) + \frac{1}{n\epsilon^2} \log \left( \frac{2Kk_1^s \zeta(s)}{\delta} \right)^2 + \frac{1}{m\epsilon^2} \log \left( \frac{2Kk_2^s \zeta(s)}{\delta} \right)^2 \quad (4)$$

*with the GLR stopping rule yields a $\delta$-correct algorithm for sub-Gaussian distributions. The function $\mathcal{C}_G$ is defined in (15). It satisfies $\mathcal{C}_G(x) \approx x + \ln(x)$.*

*Remark.* We observe that approximately $c_{\epsilon,k_1,k_2}(n, m, \delta) \approx 2\log(1/\delta) + (1/n + 1/m)\log(1/\delta)^2/\epsilon^2$.

*Proof sketch.* Proving $\delta$-correctness of a GLR stopping rule is done by leveraging concentration results. Specifically, we start by decomposing the failure probability $\mathbb{P}_\mu (\tau_\delta < +\infty, \hat{a} \neq a^\star)$ into a non-private and a private part using the basic property of $\mathbb{P}(X + Y \geq a + b) \leq \mathbb{P}(X \geq a) + \mathbb{P}(Y \geq b)$. The two-factor in front of $c_{k_1 k_2}$ originates from the looseness of this decomposition. To remove it, we would need a tighter stopping threshold that jointly controls both the non-private and the private parts. We conclude using concentration results from sub-Gaussian random variables for the non-private part, and Laplace random variables for the private part.

**Theorem 5** (Asymptotic upper bound on expected sample complexity)**.** *Let $(\delta, \beta) \in (0, 1)^2$ and $\epsilon > 0$. Combined with the private GLR stopping rule using threshold as in (4),* AdaP-TT *is $\delta$-correct and satisfies that, for all $\mu \in \mathbb{R}^K$ such that $\min_{a \neq b} |\mu_a - \mu_b| > 0$,*

$$\limsup_{\delta \to 0} \frac{\mathbb{E}_\mu[\tau_\delta]}{\log(1/\delta)} \leq 4T^\star_{\mathrm{KL},\beta}(\boldsymbol{\mu}) \left( 1 + \sqrt{1 + \frac{\Delta_{\max}^2}{2\epsilon^2}} \right) \, .$$

*Proof sketch.* We adapt the asymptotic proof of the TTUCB algorithm, which is based on the unified analysis of Top Two algorithms from [JDB+22]. Below, we present high-level ideas of the proof and specify the effect of different elements of AdaP-TT on the expected sample complexity.

*Consequences of Theorem 5.* (1) The **non-private TTUCB algorithm** [JD22] achieves a sample complexity of $T^\star_{\mathrm{KL},\beta}(\boldsymbol{\mu})$ for sub-Gaussian random variables. The proof relies on showing that the empirical pulling counts are converging towards the $\beta$-optimal allocation $\omega^\star_{\mathrm{KL},\beta}(\boldsymbol{\mu})$. (2) The **effect of doubling and forgetting** is a multiplicative four-factor, i.e. $4T^\star_{\mathrm{KL},\beta}(\boldsymbol{\mu})$. The first two-factor is due to forgetting since we throw away half of the samples. The second two-factor is due to doubling since we have to wait for the end of an episode to evaluate the stopping condition. (3) The **Laplace noise** only affects the empirical estimate of the mean. Since the Laplace noise has no bias and a sub-exponential tail, the private means will still converge towards their true values. Therefore, the empirical counts of AdaP-TT will also converge to $\omega^\star_{\mathrm{KL},\beta}(\boldsymbol{\mu})$ asymptotically. (4) While the **Laplace noise has little effect on the sampling rule** itself, it **changes drastically the dependency in** $\log(1/\delta)$ **of the threshold** used in the GLR stopping rule. The private threshold $c_{\epsilon,k_1,k_2}$ has an extra factor $\mathcal{O}(\log^2(1/\delta))$ compared to the non-private one $c_k$. Using the convergence towards $\omega^\star_{\mathrm{KL},\beta}(\boldsymbol{\mu})$, the stopping condition is met as soon as $\frac{n}{T^\star_{\mathrm{KL},\beta}(\boldsymbol{\mu})} \lesssim 2\log(1/\delta) + \frac{\Delta_{\max}^2}{2\epsilon^2} \frac{T^\star_{\mathrm{KL},\beta}(\boldsymbol{\mu})}{n} \log^2(1/\delta)$. Solving the inequality for $n$ concludes the proof while adding a multiplicative four-factor.

**Algorithm 2** AdaP-TT. *Private statistics are in red. Changes due to privacy are in blue.*

---

1: **Input:** $\beta \in (0,1)$, risk $\delta \in (0,1)$, privacy budget $\epsilon$, thresholds $c_{\epsilon,k_1,k_2} : \mathbb{N}^2 \times (0,1) \to \mathbb{R}^+$
2: **Output:** Recommendation $\hat{a}$ and Stopping time $\tau$ satisfying $\epsilon$-global DP
3: **Initialization:** $\forall a \in [K]$, pull arm $a$, set $k_a = 1$, $T_1(a) = K + 1$, $L_{n,a} = 0$, $N_{n,a} = 1$, $n = K + 1$.
4: **for** $n > K$ **do**
5:     **if** there exists $a \in [K]$ such that $N_{n,a} \geq 2N_{T_{k_a}(a),a}$ **then**         $\triangleright$ Per-arm doubling
6:         Change phase $k_a \leftarrow k_a + 1$ for this arm $a$
7:         Set $T_{k_a}(a) = n$ and $\tilde{N}_{k_a,a} = N_{T_{k_a}(a),a} - N_{T_{k_a-1}(a),a}$     $\triangleright$ Pulls of $a$ in its last phase
8:         Set $\hat{\mu}_{k_a,a} = \tilde{N}_{k_a,a}^{-1} \sum_{s=T_{k_a-1}(a)}^{T_{k_a}(a)-1} X_s \mathbb{1}\{I_s = a\}$   $\triangleright$ Empirical mean of $a$ in its last phase
9:         Set $\tilde{\mu}_{k_a,a} = \hat{\mu}_{k_a,a} + Y_{k_a,a}$ where $Y_{k_a,a} \sim \mathrm{Lap}((\epsilon\tilde{N}_{k_a,a})^{-1})$      $\triangleright$ Make it private
10:     **end if**
11:     Set $\hat{a}_n = \arg\max_{b \in [K]} \tilde{\mu}_{k_b,b}$             $\triangleright$ Arm with highest private mean
12:     **if** $\frac{(\tilde{\mu}_{k_{\hat{a}_n},\hat{a}_n} - \tilde{\mu}_{k_b,b})^2}{1/\tilde{N}_{k_{\hat{a}_n},\hat{a}_n} + 1/\tilde{N}_{k_b,b}} \geq 2c_{\epsilon,k_{\hat{a}_n},k_b}(\tilde{N}_{k_{\hat{a}_n},\hat{a}_n}, \tilde{N}_{k_b,b}, \delta)$ for all $b \neq \hat{a}_n$ **then**
13:         **return** $(\hat{a}_n, n)$   $\triangleright$ If GLR stopping condition is met, recommend the empirical best arm
14:     **end if**
15:     Set $B_n = \arg\max_{a \in [K]}\{\tilde{\mu}_{k_a,a} + \sqrt{k_a/\tilde{N}_{k_a,a}} + k_a/(\epsilon\tilde{N}_{k_a,a})\}$      $\triangleright$ Private UCB leader
16:     Set $C_n = \arg\min_{a \neq B_n} \frac{\tilde{\mu}_{k_{B_n},B_n} - \tilde{\mu}_{k_a,a}}{\sqrt{1/N_{n,B_n} + 1/N_{n,a}}}$       $\triangleright$ Private TC challenger
17:     Set $I_n = B_n$ if $N_{n,B_n}^{B_n} \leq \beta L_{n+1,B_n}$, else $I_n = C_n$          $\triangleright$ Tracking
18:     Pull $I_n$ and observe $X_n \sim \nu_{I_n}$
19:     Set $N_{n+1,I_n} \leftarrow N_{n,I_n}+1$, $N_{n+1,I_n}^{B_n} \leftarrow N_{n,I_n}^{B_n}+1$ and $L_{n+1,B_n} \leftarrow L_{n,B_n}+1$. Set $n \leftarrow n+1$
20: **end for**

---

*Discussion.* For $\beta = 1/2$, it is well known that $T_{\mathrm{KL},1/2}^\star(\boldsymbol{\mu}) \leq 2T_{\mathrm{KL}}^\star(\boldsymbol{\mu}) \leq 8\sum_{a \neq a^\star} \Delta_a^{-2}$. We consider Bernoulli instances ($0 < \Delta_{\min} \leq \Delta_{\max} < 1$), where the gaps have the same order of magnitude, i.e. *Condition 1*: there exists a constant $C \geq 1$ such that $\Delta_{\max}/\Delta_{\min} \leq C$. For such instances, there exists a universal constant $c$, such that

$$\limsup_{\delta \to 0} \frac{\mathbb{E}_{\boldsymbol{\mu}}[\tau_\delta]}{\log(1/\delta)} \leq c \max\left\{ T_{\mathrm{KL},1/2}^\star(\boldsymbol{\mu}), C\epsilon^{-1}\sum_{a \neq a^\star} \Delta_a^{-1} \right\}.$$

Without privacy, i.e. $\epsilon \to +\infty$, AdaP-TT yields a multiplicative eight-factor. On top of the four-factor due to doubling and forgetting, another multiplicative two comes from $2c_{k_1k_2}$ in Equation (4).

**Comparison to the lower bound.** For Bernoulli bandits verifying *Condition 1*, the upper bound of Theorem 5 matches the $T_{\mathrm{TV}}^\star(\boldsymbol{\mu})/\epsilon$ lower bound of Corollary 1 up to constants in the high-privacy regime, i.e. when $\epsilon \preceq T_{\mathrm{TV}}^\star(\boldsymbol{\mu})/T_{\mathrm{KL}}^\star(\boldsymbol{\mu})$. In the low-privacy regime, the upper bound reduces to $T_{\mathrm{KL},1/2}^\star(\boldsymbol{\mu})$. In Appendix E.7, we discuss in-depth why this difference is necessary for private BAI algorithms based on the GLR stopping rule, which poses an interesting open problem.

**Comparison to DP-SE.** DP-SE is a private version of the successive-elimination algorithm studied in [SS19] for the regret minimisation setting. The algorithm samples active arms uniformly during phases of geometrically increasing length. Based on the private confidence bounds, DP-SE eliminates provably sub-optimal arms at the end of each phase. Due to its phased-elimination structure, DP-SE can be easily converted into an $\epsilon$-DP-FC-BAI algorithm, where we stop once there is only one active arm left. In particular, the proof of Theorem 4.3 of [SS19] shows that with high probability any sub-optimal arm $a \neq a^\star$ is sampled no more than $\mathcal{O}(\Delta_a^2 + (\epsilon\Delta_a)^{-1})$. From this result, it is straightforward to extract a sample complexity upper bound for DP-SE, i.e. $\mathcal{O}(\sum_{a \neq a^\star} \Delta_a^{-2} + \sum_{a \neq a^\star} (\epsilon\Delta_a)^{-1})$. This shows that DP-SE too achieves (ignoring constants) the high-privacy lower bound $T_{\mathrm{TV}}^\star(\boldsymbol{\mu})/\epsilon$ for Bernoulli instances. However, due to its uniform sampling within the phases, DP-SE is less adaptive than AdaP-TT. Inside a phase, DP-SE continues to sample arms that might already be known to be bad, while AdaP-TT adapts its sampling rule based on the transportation costs that reflect the amount of evidence collected in favour of the hypothesis that the leader is the best arm. Finally, AdaP-TT has the advantage of being anytime, i.e. its sampling strategy does not depend on the risk $\delta$.

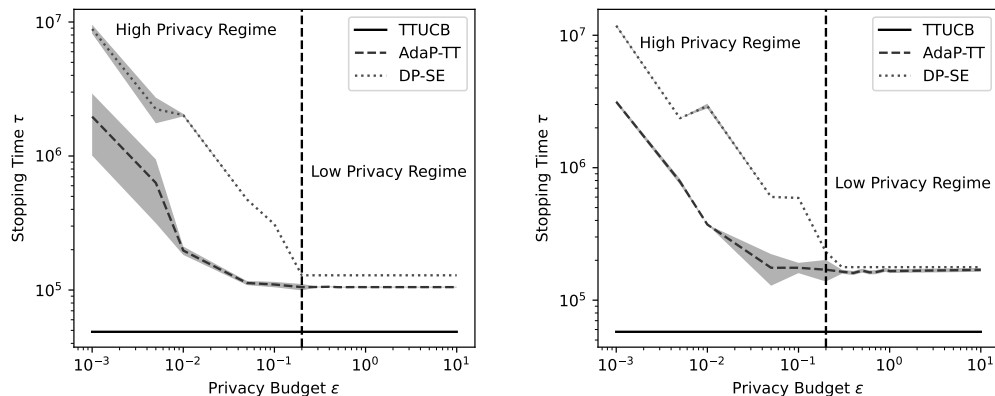

Figure 1: Evolution of the stopping time $\tau$ (mean $\pm$ std. over 100 runs) of AdaP-TT, DP-SE, and TTUCB with respect to the privacy budget $\epsilon$ for $\delta = 10^{-2}$ on Bernoulli instance $\mu_1$ (left) and $\mu_2$ (right). The shaded vertical line separates the two privacy regimes. AdaP-TT outperforms DP-SE.

## 5 Experimental analysis

We perform experiments to show that: (i) AdaP-TT has better empirical performance compared to DP-SE, and (ii) the transition between high and low-privacy regimes is reflected empirically.

**Experimental setup.** We compare the performances of AdaP-UCB and DP-SE for FC-BAI in different Bernoulli instances as in [SS19]. The first instance has means $\mu_1 = (0.95, 0.9, 0.9, 0.9, 0.5)$ and the second instance has means $\mu_2 = (0.75, 0.7, 0.7, 0.7, 0.7)$. As a benchmark, we also compare to the non-private TTUCB. We set the risk $\delta = 10^{-2}$. We implement all the algorithms in Python (version 3.8) and on an 8-core 64-bits Intel i5@1.6 GHz CPU. We run each algorithm 100 times, and plot corresponding average and standard deviations of stopping times in Figure 1. We also test the algorithms on other Bernoulli instances and report the results in Appendix F.

**Result analysis.** *a. Efficiency in performance.* AdaP-TT requires less samples than DP-SE to provide a $\delta$-correct answer. In the high privacy regime, i.e. small $\epsilon$, AdaP-TT outperforms DP-SE in all the instances tested. In the low privacy regimes, i.e. large $\epsilon$, both algorithms have similar performance that in the worst case is four times the samples required of TTUCB, as shown theoretically.
*b. Impact of privacy regimes.* As indicated by the theoretical sample complexity lower bounds and upper bounds, the experimental performance of AdaP-TT demonstrates two regimes: a high-privacy regime (for $\epsilon < 0.2$), where the stopping time of AdaP-TT depends on the privacy budget $\epsilon$, and a low privacy regime (for $\epsilon > 0.2$), where the performance of AdaP-TT does not depend on $\epsilon$.

## 6 Conclusion and future works

We study FC-BAI with $\epsilon$-global DP. We derive a sample complexity lower bound that quantifies the additional samples needed by a $\delta$-correct BAI strategy in order to ensure $\epsilon$-global DP. The lower bound further suggests the existence of two privacy regimes. In the *low-privacy regime*, no additional samples are needed, and *privacy can be achieved for free*. For the *high-privacy regime*, the lower bound reduces to $\Omega(\epsilon^{-1}T^\star_{\text{TV}})$, and *more samples are required*. We also propose AdaP-TT, an $\epsilon$-global DP variant of the Top Two algorithms, that runs in adaptive phases and adds Laplace noise. AdaP-TT achieves the high privacy regime lower bound up to multiplicative constants.

The upper bound matches the lower bound by a multiplicative constant in the high privacy regime, and is also loose in some instances in the low privacy regime, due to the mismatch between the KL divergence of Bernoulli distributions and that of Gaussian. It would be an interesting technical challenge to merge this gap. One possible direction to solve this issue is to use transportation costs tailored to Bernoulli for both the Top Two Sampling and the stopping. Another interesting direction would be to extend the proposed technique to other variants of pure DP, namely $(\epsilon, \delta)$-DP and Rényi-DP [Mir17], or other trust models, namely local DP [DJW13] and shuffle DP [Che21, GDD$^+$21].

## Acknowledgments and Disclosure of Funding

This work has been partially supported by the THIA ANR program "AI_PhD@Lille". A. Al-Marjani acknowledges the support of the Chaire SeqALO (ANR-20-CHIA-0020). D. Basu acknowledges the Inria-Kyoto University Associate Team "RELIANT" for supporting the project, and the ANR JCJC for the REPUBLIC project (ANR-22-CE23-0003-01). We thank Emilie Kaufmann and Aurélien Garivier for the interesting conversations. We also thank Philippe Preux for his support.

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

# Appendix

## Table of Contents

# A  Outline

The appendices are organized as follows:

- The proof of our lower bound is detailed in Appendix B (Theorem 2).
- The proof of the privacy of AdaP-TT is done in Appendix C (Theorem 3).
- The AdaP-TT algorithm is presented in more details in Appendix D, and we show the $\delta$-correctness of the non-private GLR stopping rule (Theorem 4).
- The asymptotic upper bound on the expected sample complexity of AdaP-TT is proven in Appendix E (Theorem 5).
- Extended experiments are presented in Appendix F.

# B  Lower bounds on sample complexity

In this section, we provide the proofs for the sample complexity lower bounds. First, we present the canonical model for BAI to introduce the relevant quantities. Then, we prove an $\epsilon$-global version of the transportation lemma, i.e. Lemma 1. Using this lemma, we prove Theorem 2. Finally, we prove the formula expressing the TV characteristic time for Bernoulli instances.

## B.1  Canonical model for BAI

Let $\boldsymbol{\nu} \triangleq \{\nu_a : a \in [K]\}$ be a bandit instance, consisting of $K$ arms with finite means $\{\mu_a\}_{a\in[K]}$. Now, we recall the interaction between a BAI strategy $\pi$ and the bandit instance $\nu$ in the Protocol 1. The BAI strategy $\pi$ halts at $\tau$, samples a sequence of actions $\underline{A}^\tau$, and recommends the action $\hat{A}$. Let $\mathbb{P}_{\boldsymbol{\nu},\pi}$ be the probability distribution over the triplets $(\tau, \underline{A}^\tau, \hat{A})$, when the BAI strategy $\pi$ interacts with the bandit instance $\nu$.

For a fixed $T > 1$, a sequence of actions $\underline{a}^T = (a_1, \ldots, a_T) \in [K]^T$ and a recommendation $\hat{a} \in [K]$, we define the event $E = \{\tau = T, \underline{A}^\tau = \underline{a}^T, \hat{A} = \hat{a}\}$. We have that

$$\mathbb{P}_{\boldsymbol{\nu},\pi}(E) = \int_{\underline{r}^T = (r_1, \ldots, r_T) \in \mathbb{R}^T} \pi(\underline{a}^T, \hat{a}, T \mid \underline{r}^T) \prod_{t=1}^T \mathrm{d}\nu_{a_t}(r_t) dr_t$$

where

$$\pi(\underline{a}^T, \hat{a}, T \mid \underline{r}^T) \triangleq \mathrm{Rec}_{T+1}\left(\hat{a} \mid \mathcal{H}_T\right) \mathrm{S}_{T+1}\left(\top \mid \mathcal{H}_T\right) \prod_{t=1}^T \mathrm{S}_t\left(a_t \mid \mathcal{H}_{t-1}\right)$$

and $\mathcal{H}_t = (a_1, r_1, \ldots, a_t, r_t)$.

**Remark on the bandit feedback.** Let $\pi$ be an $\epsilon$-DP BAI strategy. Let $T \geq 1$, $\underline{a}^T \in [K]^T$ a sequence sampled actions and $\hat{a} \in [K]$ a recommended actions. This time, let $\underline{r}^T = \{r_1, \ldots, r_T\} \in \mathbb{R}^T$ and $\underline{r'}^T \in \mathbb{R}^T$ two neighbouring sequence of rewards, i.e. $d_{\mathrm{Ham}}(\underline{r}^T, \underline{r'}^T) \triangleq \sum_{t=1}^T \mathbb{1}\{r_t \neq r'_t\} = 1$. Consider the table of rewards $\underline{d}^T$ consisting of concatenating $\underline{r}^T$ colon-wise $K$ times, i.e. $\underline{d}^T_{t,i} = \underline{r}^T_t$ for all $i \in [K]$ and all $t \in [T]$. Define $\underline{d'}^T$ similarly with respect to $\underline{r'}^T$.

In this case, by definition of $\pi$, $\underline{d}^T$ and $\underline{d'}^T$, it is direct that

$$\pi(\underline{a}^T, \hat{a}, T \mid \underline{r}^T) = \pi(\underline{a}^T, \hat{a}, T \mid \underline{d}^T)$$

and $d_{\mathrm{Ham}}(\underline{d}^T, \underline{d'}^T) = 1$.

Which means that

$$\pi(\underline{a}^T, \hat{a}, T \mid \underline{r}^T) \leq e^\epsilon \pi(\underline{a}^T, \hat{a}, T \mid \underline{r'}^T).$$

In other words, *if $\pi$ is $\epsilon$-pure DP for neighbouring table of rewards $\underline{d}^T$, then $\pi$ is also $\epsilon$-pure DP for neighbouring sequence of observed rewards $\underline{r}^T$.*

## B.2 Transportation lemma under $\epsilon$-global DP: Proof of Lemma 1

**Lemma 1** (Transportation lemma under $\epsilon$-global DP). *Let $\delta \in (0,1)$ and $\epsilon > 0$. Let $\boldsymbol{\nu}$ be a bandit instance and $\lambda \in \mathrm{Alt}(\boldsymbol{\nu})$. For any $\delta$-correct $\epsilon$-global DP BAI strategy, we have that*

$$6\epsilon \sum_{a=1}^{K} \mathbb{E}_{\boldsymbol{\nu},\pi}\left[N_a(\tau)\right] \mathrm{TV}\left(\nu_a \parallel \lambda_a\right) \geq \mathrm{kl}(1-\delta, \delta),$$

*where $\mathrm{kl}(x,y) \triangleq x \log \frac{x}{y} + (1-x) \log \frac{1-x}{1-y}$ for $x, y \in (0,1)$.*

*Proof.* **Step 1: Distinguishability due to $\delta$-correctness.** Let $\pi$ be a $\delta$-correct $\epsilon$-global DP BAI strategy. Let $\boldsymbol{\nu}$ be a bandit instance and $\lambda \in \mathrm{Alt}(\boldsymbol{\nu})$.

Let $\mathbb{P}_{\boldsymbol{\nu},\pi}$ denote the probability distribution of $(\underline{A}, \widehat{A}, \tau)$ when the BAI strategy $\pi$ interacts with $\nu$. For any alternative instance $\lambda \in \mathrm{Alt}(\boldsymbol{\nu})$, the data-processing inequality gives that

$$\mathrm{KL}\left(\mathbb{P}_{\boldsymbol{\nu},\pi} \parallel \mathbb{P}_{\boldsymbol{\lambda},\pi}\right) \geq \mathrm{kl}\left(\mathbb{P}_{\boldsymbol{\nu},\pi}\left(\widehat{A} = a^\star(\boldsymbol{\nu})\right), \mathbb{P}_{\boldsymbol{\lambda},\pi}\left(\widehat{A} = a^\star(\boldsymbol{\nu})\right)\right)$$
$$\geq \mathrm{kl}(1-\delta, \delta). \tag{5}$$

where the second inequality is because $\pi$ is $\delta$-correct i.e. $\mathbb{P}_{\boldsymbol{\nu},\pi}\left(\widehat{A} = a^\star(\boldsymbol{\nu})\right) \geq 1 - \delta$ and $\mathbb{P}_{\boldsymbol{\lambda},\pi}\left(\widehat{A} = a^\star(\boldsymbol{\nu})\right) \leq \delta$, and the monotonicity of the $\mathrm{kl}$.

**Step 2: Connecting KL and TV under $\epsilon$-global DP.** On the other hand, by the definition of the KL, we have that

$$\mathrm{KL}\left(\mathbb{P}_{\boldsymbol{\nu},\pi} \parallel \mathbb{P}_{\boldsymbol{\lambda},\pi}\right) = \mathbb{E}_{\tau, \underline{A}^\tau, \hat{A} \sim \mathbb{P}_{\boldsymbol{\nu},\pi}}\left[\log\left(\frac{\mathbb{P}_{\boldsymbol{\nu},\pi}(\tau, \underline{A}^\tau, \hat{A})}{\mathbb{P}_{\boldsymbol{\lambda},\pi}(\tau, \underline{A}^\tau, \hat{A})}\right)\right]$$

where

$$\mathbb{P}_{\boldsymbol{\nu},\pi}(\tau = T, \underline{A}^\tau = \underline{a}^T, \hat{A} = \hat{a}) = \int_{\underline{r} \in \mathbb{R}^T} \pi(\underline{a}^T, \hat{a}, T \mid \underline{r}) \prod_{t=1}^{T} \mathrm{d}\nu_{a_t}(r_t).$$

Since $\pi$ is $\epsilon$-global DP, using the sequential Karwa-Vadhan lemma [AB22, Lemma 2], we get that

$$\log\left(\frac{\mathbb{P}_{\boldsymbol{\nu},\pi}(\tau = T, \underline{A}^\tau = \underline{a}^T, \hat{A} = \hat{a})}{\mathbb{P}_{\boldsymbol{\lambda},\pi}(\tau = T, \underline{A}^\tau = \underline{a}^T, \hat{A} = \hat{a})}\right) \leq 6\epsilon \sum_{t=1}^{T} \mathrm{TV}\left(\nu_{a_t} \parallel \lambda_{a_t}\right)$$
$$= 6\epsilon \sum_{a=1}^{K} N_a(T) \mathrm{TV}\left(\nu_a \parallel \lambda_a\right)$$

Which gives that

$$\mathrm{KL}\left(\mathbb{P}_{\boldsymbol{\nu},\pi} \parallel \mathbb{P}_{\boldsymbol{\lambda},\pi}\right) \leq 6\epsilon \, \mathbb{E}_{\boldsymbol{\nu},\pi}\left[\sum_{a=1}^{K} N_a(\tau) \mathrm{TV}\left(\nu_a \parallel \lambda_a\right)\right]. \tag{6}$$

Combining Inequalities 5 and 6 concludes the proof. $\qquad\square$

## B.3 Lower bound on sample complexity: Proof of Theorem 2

**Theorem 2** (Sample complexity lower bound for $\epsilon$-DP-FC-BAI). *Let $\delta \in (0,1)$ and $\epsilon > 0$. For any $\delta$-correct $\epsilon$-global DP BAI strategy, we have that*

$$\mathbb{E}_{\boldsymbol{\nu}}[\tau] \geq T^\star(\boldsymbol{\nu}; \epsilon) \log(1/3\delta),$$

*where $(T^\star(\boldsymbol{\nu}; \epsilon))^{-1} \triangleq \sup_{\omega \in \Sigma_K} \inf_{\boldsymbol{\lambda} \in \mathrm{Alt}(\boldsymbol{\nu})} \min\left(\sum_{a=1}^{K} \omega_a \mathrm{KL}\left(\nu_a \parallel \lambda_a\right), 6\epsilon \sum_{a=1}^{K} \omega_a \mathrm{TV}\left(\nu_a \parallel \lambda_a\right)\right).$*

*Proof.* Let $\pi$ be a $\delta$-correct $\epsilon$-global DP BAI strategy. Let $\boldsymbol{\nu}$ be a bandit instance and $\boldsymbol{\lambda} \in \mathrm{Alt}(\boldsymbol{\nu})$.

Let $\mathbb{E}$ denote the expectation under $\mathbb{P}_{\boldsymbol{\nu},\pi}$, ie $\mathbb{E} \triangleq \mathbb{E}_{\boldsymbol{\nu},\pi}$.

By Lemma 1, we have that $6\epsilon \sum_{a=1}^{K} \mathbb{E}[N_a(\tau)] \mathrm{TV}(\nu_a \| \lambda_a) \geq \mathrm{kl}(1-\delta,\delta)$.

Lemma 1 from [KCG16] gives that $\sum_{a=1}^{K} \mathbb{E}[N_a(\tau)] \mathrm{KL}(\nu_a \| \lambda_a) \geq \mathrm{kl}(1-\delta,\delta)$.

Since these two inequalities hold for all $\boldsymbol{\lambda} \in \mathrm{Alt}(\boldsymbol{\nu})$, we get

$$
\begin{aligned}
\mathrm{kl}(1-\delta,\delta) &\leq \inf_{\boldsymbol{\lambda} \in \mathrm{Alt}(\boldsymbol{\nu})} \min\left( 6\epsilon \sum_{a=1}^{K} \mathbb{E}[N_a(\tau)] \mathrm{TV}(\nu_a \| \lambda_a), \sum_{a=1}^{K} \mathbb{E}[N_a(\tau)] \mathrm{KL}(\nu_a \| \lambda_a) \right) \\
&\overset{(a)}{=} \mathbb{E}[\tau] \inf_{\boldsymbol{\lambda} \in \mathrm{Alt}(\boldsymbol{\nu})} \min\left( 6\epsilon \sum_{a=1}^{K} \frac{\mathbb{E}[N_a(\tau)]}{\mathbb{E}[\tau]} \mathrm{TV}(\nu_a \| \lambda_a), \sum_{a=1}^{K} \frac{\mathbb{E}[N_a(\tau)]}{\mathbb{E}[\tau]} \mathrm{KL}(\nu_a \| \lambda_a) \right) \\
&\overset{(b)}{\leq} \mathbb{E}[\tau] \left( \sup_{\omega \in \Sigma_K} \inf_{\boldsymbol{\lambda} \in \mathrm{Alt}(\boldsymbol{\nu})} \min\left( 6\epsilon \sum_{a=1}^{K} \omega_a \mathrm{TV}(\nu_a \| \lambda_a), \sum_{a=1}^{K} \omega_a \mathrm{KL}(\nu_a \| \lambda_a) \right) \right).
\end{aligned}
$$

(a) is due to the fact that $\mathbb{E}[\tau]$ does not depend on $\boldsymbol{\lambda}$. (b) is obtained by noting that the vector $(\omega_a)_{a \in [K]} \triangleq \left( \frac{\mathbb{E}_{\nu,\pi}[N_a(\tau)]}{\mathbb{E}_{\nu,\pi}[\tau]} \right)_{a \in [K]}$ belongs to the simplex $\Sigma_K$.

The theorem follows by noting that for $\delta \in (0,1), \mathrm{kl}(1-\delta,\delta) \geq \log(1/3\delta)$. $\qquad\square$

### B.4 TV characteristic time for Bernoulli instances: Proof of Proposition 1

**Proposition 1** (TV characteristic time for Bernoulli instances)**.** *Let $\nu$ be a bandit instance, i.e. such that $\nu_a = Bernoulli(\mu_a)$ and $\mu_1 > \mu_2 \geq \cdots \geq \mu_K$. Let $\Delta_a \triangleq \mu_1 - \mu_a$ and $\Delta_{min} \triangleq \min_{a \neq 1} \Delta_a$. We have that*

$$
T_{\mathrm{TV}}^\star(\boldsymbol{\nu}) = \frac{1}{\Delta_{min}} + \sum_{a=2}^{K} \frac{1}{\Delta_a}, \qquad \text{and} \qquad \frac{1}{\Delta_{min}} \leq T_{\mathrm{TV}}^\star(\boldsymbol{\nu}) \leq \frac{K}{\Delta_{min}}.
$$

*Proof.* **Step 1:** Let $\nu$ be a bandit instance, i.e. such that $\nu_a \triangleq \mathrm{Bernoulli}(\mu_a)$ and $\mu_1 > \mu_2 \geq \cdots \geq \mu_K$.

For the alternative bandit instance $\boldsymbol{\lambda}$, we refer to the mean of arm $a$ as $\rho_a$, i.e. $\lambda_a \triangleq \mathrm{Bernoulli}(\rho_a)$.

By the definition of $T_{\mathbf{TV}}^\star$, we have that

$$
\begin{aligned}
(T_{\mathbf{TV}}^\star(\boldsymbol{\nu}))^{-1} &= \sup_{\omega \in \Sigma_K} \inf_{\boldsymbol{\lambda} \in \mathrm{Alt}(\boldsymbol{\nu})} \sum_{a=1}^{K} \omega_a \mathrm{TV}(\nu_a \| \lambda_a) \\
&\overset{(a)}{=} \sup_{\omega \in \Sigma_K} \min_{a \neq 1} \inf_{\boldsymbol{\lambda}: \rho_a > \rho_1} \omega_1 |\mu_1 - \rho_1| + \omega_a |\mu_a - \rho_a| \\
&\overset{(b)}{=} \sup_{\omega \in \Sigma_K} \min_{a \neq 1} \min(\omega_1, \omega_a) \Delta_a \\
&\overset{(c)}{=} \sup_{\omega \in \Sigma_K} \omega_1 \min_{a \neq 1} \min(1, \frac{\omega_a}{\omega_1}) \Delta_a \\
&\overset{(d)}{=} \sup_{(x_2,\ldots,x_K) \in (\mathbb{R}^+)^{K-1}} \frac{\min_{a \neq 1} g_a(x_a)}{1 + x_2 + \cdots + x_K},
\end{aligned}
$$

where $g_a(x_a) \triangleq \min(1, x_a) \Delta_a$.

Equality (a) is obtained due to the fact that $\mathrm{Alt}(\boldsymbol{\nu}) = \bigcup_{a \neq 1}\{\boldsymbol{\lambda} : \rho_a > \rho_1\}$, and for Bernoullis, $\mathrm{TV}(\nu_a \| \lambda_a) = |\mu_a - \rho_a|$.

Equality (b) is true, since $\inf_{\boldsymbol{\lambda}: \rho_a > \rho_1} \omega_1 |\mu_1 - \rho_1| + \omega_a |\mu_a - \rho_a| = \min(\omega_1, \omega_a) \Delta_a$.

Equality (c) holds true, since $\omega_1 \neq 1$ (if $\omega_1 = 0$, the value of the objective is 0).

Equality (d) is obtained by the change of variable $x_a \triangleq \frac{\omega_a}{\omega_1}$

**Step 2:** Let $(x_2, \ldots, x_K) \in (\mathbb{R}^+)^{K-1}$. By the definition of $g_a$, we have that

$$g_a(x_a) \leq x_a \Delta_a \quad \text{and} \quad g_a(x_a) \leq \Delta_a.$$

This leads to the inequalities

$$\min_{a \neq 1} g_a(x_a) \leq g_a(x_a) \leq x_a \Delta_a \quad \text{and} \quad \min_{a \neq 1} g_a(x_a) \leq \Delta_{\min}.$$

Thus,

$$\left( \min_{a \neq 1} g_a(x_a) \right) \left( \frac{1}{\Delta_{\min}} + \sum_{a=2}^{K} \frac{1}{\Delta_a} \right) = \frac{\min_{a \neq 1} g_a(x_a)}{\Delta_{\min}} + \sum_{a=2}^{K} \frac{\min_{a \neq 1} g_a(x_a)}{\Delta_a}$$

$$\leq 1 + \sum_{a=2}^{K} x_a.$$

This means that for every $(x_2, \ldots, x_K) \in (\mathbb{R}^+)^{K-1}$,

$$\frac{\min_{a \neq 1} g_a(x_a)}{1 + x_2 + \cdots + x_K} \leq \frac{1}{\frac{1}{\Delta_{\min}} + \sum_{a=2}^{K} \frac{1}{\Delta_a}}.$$

Here, the upper bound is achievable for $x_a^\star = \frac{\Delta_{\min}}{\Delta_a}$, since $g_a(x_a^\star) = \Delta_{\min}$ for all $a \neq 1$.

This concludes that

$$(T_{\mathbf{TV}}^\star(\boldsymbol{\nu}))^{-1} = \frac{1}{\frac{1}{\Delta_{\min}} + \sum_{a=2}^{K} \frac{1}{\Delta_a}} \qquad \Longrightarrow \qquad (T_{\mathbf{TV}}^\star(\boldsymbol{\nu})) = \Delta_{\min} + \sum_{a=2}^{K} \frac{1}{\Delta_a}.$$

**Step 3:** The lower and upper bounds on $(T_{\mathbf{TV}}^\star(\boldsymbol{\nu}))$ follow from the fact that $\frac{1}{\Delta_a} \geq 0$ for all $a$, and $\frac{1}{\Delta_a} \leq \frac{1}{\Delta_{\min}}$ for all $a \neq 1$.

Hence, we conclude the proof. $\qquad\qquad\square$

### B.5 On the total variation distance and the hardness of privacy

Our lower bound suggests that the hardness of the DP-FC-BAI problem is characterized by $T_{TV}^\star$, which is a total variation counterpart of the classic KL-based characteristic time $T_{KL}^\star$ in FC-BAI [GK16]. The total variation distance appears to be the natural measure to quantify the hardness of privacy in other settings such as regret minimization [AB22], Karwa-Vadhan lemma [KV17] and Differentially Private Assouad, Fano, and Le Cam [ASZ21]. The high-level intuition is that: Pure DP can be seen as a multiplicative stability constraint of $e^\epsilon$ when one data point changes. With group privacy, if two datasets differ in $d_{ham}$ points, then one incurs a factor $e^{d_{ham}\epsilon}$. Now, by sampling $n$ i.i.d points from a distribution $P$ and $n$ i.i.d points from a distribution $Q$, the Karwa-Vadhan lemma states that the incurred factor is $e^{(nTV(P,Q))\epsilon}$. This is proved by building a maximal coupling, which is the coupling that minimizes the Hamming distance in expectation. In brief, *the total variation naturally appears in lower bounds since it is the quantity that characterises the hardness of the optimal transport problem minimizing the hamming distance*, i.e $TV(P,Q) = \inf_{(X,Y)\sim(P,Q)} E(\mathbb{1}_{X \neq Y})$. However, it is possible that the problem can be characterized by other f-divergences. Finally, one can always go from TV to KL using Pinsker's inequality, though that would always be less tight than the TV-based lower bound.

**On the relation between $T_{TV}^\star$ and $T_{KL}^\star$** A direct application of Pinsker's inequality gives that $T_{TV}^\star(\nu) \geq \sqrt{2T_{KL}^\star(\nu)}$. For completeness, we present here the exact calculations:

For every alternative mean parameter $\lambda$ and every arm $a$, using Pinkser's inequality, we have that $d_{TV}(\mu_a, \lambda_a) \leq \sqrt{\frac{1}{2} d_{KL}(\mu_a, \lambda_a)}$. Therefore, for every allocation over arms $\omega$, we have

$$\sum_a \omega_a d_{TV}(\mu_a, \lambda_a) \leq \sum_a \omega_a \sqrt{\frac{1}{2} d_{KL}(\mu_a, \lambda_a)} \leq \sqrt{\frac{1}{2} \sum_a \omega_a d_{KL}(\mu_a, \lambda_a)} \,.$$

Taking the supremum over the simplex and the infimum over the set of alternative mean parameters yields $(T_{TV}^\star(\nu))^{-1} \leq \sqrt{\frac{1}{2}(T_{KL}^\star(\nu))^{-1}}$. This concludes the proof.

## C   Privacy analysis

In this section, we prove that AdaP-TT satisfies $\epsilon$-DP. We first provide the privacy lemma that justifies using doubling and forgetting in AdaP-TT. Using the privacy lemma and the post-processing property of DP, we conclude the privacy analysis of AdaP-TT.

### C.1   Privacy lemma for non-overlapping sequences

**Lemma 2** (Privacy of non-overlapping sequence of empirical means). *Let $\mathcal{M}$ be a mechanism that takes a **set** as input and outputs the private empirical mean, i.e.*

$$\mathcal{M}(\{r_i, \ldots, r_j\}) \triangleq \frac{1}{j-i} \sum_{t=i}^{j} r_t + Lap\left(\frac{1}{(j-i)\epsilon}\right). \tag{7}$$

*Let $\ell < T$ and $t_1, \ldots t_\ell, t_{\ell+1}$ be in $[1, T]$ such that $1 = t_1 < \cdots < t_\ell < t_{\ell+1} - 1 = T$. Let's define the following mechanism*

$$\mathcal{G} : \{r_1, \ldots, r_T\} \to \bigotimes_{i=1}^{\ell} \mathcal{M}_{\{r_{t_i}, \ldots, r_{t_{i+1}-1}\}} \tag{8}$$

*In other words, $\mathcal{G}$ is the mechanism we get by applying $\mathcal{M}$ to the non-overlapping partition of the sequence $\{r_1, \ldots, r_T\}$ according to $t_1 < \cdots < t_\ell < t_{\ell+1}$, i.e.*

$$\begin{pmatrix} r_1 \\ r_2 \\ \vdots \\ r_T \end{pmatrix} \xrightarrow{\mathcal{G}} \begin{pmatrix} \mu_1 \\ \vdots \\ \mu_\ell \end{pmatrix}$$

*where $\mu_i \sim \mathcal{M}_{\{r_{t_i}, \ldots, r_{t_{i+1}-1}\}}$.*

*For $r_t \in [0, 1]$, the mechanism $\mathcal{G}$ is $\epsilon$-DP.*

*Proof.* Let $r^T \triangleq (r_1, \ldots, r_T)$ and $r'^T \triangleq (r'_1, \ldots, r'_T)$ be two neighbouring reward sequences in [0,1]. This implies that $\exists j \in [1, T]$ such that $r_j \neq r'_j$ and $\forall t \neq j, r_t = r'_t$.

Let $\ell'$ be such that $t_{\ell'} \leq j \leq t_{\ell'+1} - 1$, and follows the convention that $t_0 = 1$ and $t_{\ell+1} = T + 1$.

Let $\mu \triangleq (\mu_1, \ldots, \mu_\ell)$ a fixed sequence of outcomes. Then,

$$\frac{\mathbb{P}(\mathcal{G}(r^T) = \mu)}{\mathbb{P}(\mathcal{G}(r'^T) = \mu)} = \frac{\mathbb{P}\left(\mathcal{M}(\{r_{t_{\ell'}}, \ldots, r_{t_{\ell'+1}-1}\}) = \mu_{\ell'}\right)}{\mathbb{P}\left(\mathcal{M}(\{r_{t_{\ell'}}, \ldots, r_{t_{\ell'+1}-1}\}) = \mu_{\ell'}\right)} \leq e^\epsilon,$$

where the last inequality holds true because $\mathcal{M}$ satisfies $\epsilon$-DP following Theorem 1.                    $\square$

## C.2 Privacy analysis of AdaP-TT: Proof of Theorem 3

**Theorem 3** (Privacy analysis). *For rewards in* $[0, 1]$, AdaP-TT *satisfies* $\epsilon$-*global DP.*

*Proof.* Let $T \geq 1$. Let $\underline{\mathbf{d}}^T = \{\mathbf{x}_1, \ldots, \mathbf{x}_T\}$ and $\underline{\mathbf{d}}'^T = \{\mathbf{d}'_1, \ldots, \mathbf{d}'_T\}$ two neighbouring reward tables in $(\mathbb{R}^K)^T$. Let $j \in [1, T]$ such that, for all $t \neq j$, $d_t = d'_t$.

We also fix a sequence of sampled actions $\underline{a}^T = \{a_1, \ldots, a_T\} \in [K]^T$ and a recommended action $\hat{a} \in K$.

We refer to AdaP-TT BAI strategy by $\pi$.

We want to show that: $\pi(\underline{a}^T, \hat{a}, T \mid \underline{\mathbf{d}}^T) \leq e^\epsilon \pi(\underline{a}^T, \hat{a}, T \mid \underline{\mathbf{d}}'^T)$.

The main idea is that the change of reward in the $j$-th reward only affects the empirical mean computed in one episode, which is made private using the Laplace Mechanism and Lemma 2.

**Step 1.** Sequential decomposition of the output probability

We observe that due to the sequential nature of the interaction, the output probability can be decomposed to a part that depends on $\underline{\mathbf{d}}^{j-1} \triangleq \{\mathbf{x}_1, \ldots, \mathbf{x}_{j-1}\}$, which is identical for both $\underline{\mathbf{d}}^T$ and $\underline{\mathbf{d}}'^T$ and a second conditional part on the history.

Specifically, we have that

$$\pi(\underline{a}^T, \hat{a}, T \mid \underline{\mathbf{d}}^T) \triangleq \operatorname{Rec}_{T+1}(\hat{a} \mid \mathcal{H}_T) \operatorname{S}_{T+1}(\top \mid \mathcal{H}_T) \prod_{t=1}^T \operatorname{S}_t(a_t \mid \mathcal{H}_{t-1})$$

$$\triangleq \mathcal{P}^\pi_{\underline{\mathbf{d}}^{j-1}}(\underline{a}^j) \mathcal{P}^\pi_{\underline{\mathbf{d}}}(a_{>j}, \hat{a}, T \mid \underline{a}^j)$$

where

- $a_{>j} \triangleq (a_{j+1}, \ldots, a_T)$

- $\mathcal{P}^\pi_{\underline{\mathbf{d}}^{j-1}}(\underline{a}^j) \triangleq \prod_{t=1}^j \operatorname{S}_t(a_t \mid \mathcal{H}_{t-1})$

- $\mathcal{P}^\pi_{\underline{\mathbf{d}}}(a_{>j}, \hat{a}, T \mid \underline{a}^j) \triangleq \operatorname{Rec}_{T+1}(\hat{a} \mid \mathcal{H}_T) \operatorname{S}_{T+1}(\top \mid \mathcal{H}_T) \prod_{t=j+1}^T \operatorname{S}_t(a_t \mid \mathcal{H}_{t-1})$

Similarly

$$\pi(\underline{a}^T, \hat{a}, T \mid \underline{\mathbf{d}}'^T) \triangleq \mathcal{P}^\pi_{\underline{\mathbf{d}}^{j-1}}(\underline{a}^j) \mathcal{P}^\pi_{\underline{\mathbf{d}}'}(a_{>j}, \hat{a}, T \mid \underline{a}^j)$$

since $\underline{\mathbf{d}}'^{j-1} = \underline{\mathbf{d}}^{j-1}$.

Which means that

$$\frac{\pi(\underline{a}^T, \hat{a}, T \mid \underline{\mathbf{d}}^T)}{\pi(\underline{a}^T, \hat{a}, T \mid \underline{\mathbf{d}}'^T)} = \frac{\mathcal{P}^\pi_{\underline{\mathbf{d}}}(a_{>j}, \hat{a}, T \mid \underline{a}^j)}{\mathcal{P}^\pi_{\underline{\mathbf{d}}'}(a_{>j}, \hat{a}, T \mid \underline{a}^j)} \tag{9}$$

**Step 2.** The adaptive episodes are the same, before step $j$

Let $\ell$ such that $t_\ell \leq j < t_{\ell+1}$ when $\pi$ interacts with $\underline{\mathbf{d}}^T$. Let us call it $\psi^\pi_{\underline{\mathbf{d}}^T}(j) \triangleq \ell$.

Similarly, let $\ell'$ such that $t_{\ell'} \leq j < t_{\ell'+1}$ when $\pi$ interacts with $\underline{\mathbf{d}}'^T$. Let us call it $\psi^\pi_{\underline{\mathbf{d}}'^T}(j) \triangleq \ell'$.

Since $\psi^\pi_{\underline{\mathbf{d}}^T}(j)$ only depends on $\underline{\mathbf{d}}^{j-1}$, which is identical for $\underline{\mathbf{d}}^T$ and $\underline{\mathbf{d}}'^T$, we have that $\psi^\pi_{\underline{\mathbf{d}}^T}(j) = \psi^\pi_{\underline{\mathbf{d}}'^T}(j)$ with probability 1.

We call $\xi_j$ the last **time-step** of the episode $\psi^\pi_{\underline{\mathbf{d}}^T}(j)$, i.e $\xi_j \triangleq t_{\psi^\pi_{\underline{\mathbf{d}}^T}(j)+1} - 1$.

**Step 3.** Private sufficient statistics

Let $r_t \triangleq \underline{\mathbf{d}}^T_{t, a_t}$, be the reward corresponding to the action $a_t$ in the table $\underline{\mathbf{d}}^T$. Similarly, $r'_t \triangleq \underline{\mathbf{d}}'^T_{t, a_t}$ for $\underline{\mathbf{d}}'^T$.

Let us define $L_j \triangleq \mathcal{G}_{\{r_1,\dots,r_{\xi_j}\}}$ and $L'_j \triangleq \mathcal{G}_{\{r'_1,\dots,r'_{\xi_j}\}}$, where $\mathcal{G}$ is defined as in Eq. 8, using the same episodes for $d$ and $d'$. In other words, $L_j$ is the list of private empirical means computed on a non-overlapping sequence of rewards before step $\xi_j$.

Using the forgetting structure of AdaP-TT, there exists a randomised mapping $f_{\underline{\mathbf{d}}_{>\xi_j}}$ such that $\mathcal{P}^{\pi}_{\underline{\mathbf{d}}}(.\mid \underline{a}^j) = f_{\underline{\mathbf{d}}_{>\xi_j}} \circ L_j$ and $\mathcal{P}^{\pi}_{\underline{\mathbf{d}}'}(.\mid \underline{a}^j) = f_{\underline{\mathbf{d}}_{>\xi_j}} \circ L'_j$.

In other words, the interaction of $\pi$ with $\underline{\mathbf{d}}$ and $\underline{\mathbf{d}}'$ from step $\xi_j + 1$ until $T$ only depends on the sufficient statistics $L_j$, which summarises what happened before $\xi_j$, and the new inputs $\underline{\mathbf{d}}_{>\xi_j}$, which are the same for $\underline{\mathbf{d}}$ and $\underline{\mathbf{d}}'$.

**Step 4.** Concluding with Lemma 2 and the post-processing lemma

Since rewards are in $[0,1]$, using Lemma 2, we have that $\mathcal{G}$ is $\epsilon$-DP.

Since $\mathcal{P}^{\pi}_{\underline{\mathbf{d}}}(.\mid \underline{a}^j)$ is just a post-processing of the output of $\mathcal{G}$, we have that

$$\frac{\mathcal{P}^{\pi}_{\underline{\mathbf{d}}}(a_{>j}, \hat{a}, T \mid \underline{a}^j)}{\mathcal{P}^{\pi}_{\underline{\mathbf{d}}'}(a_{>j}, \hat{a}, T \mid \underline{a}^j)} \le e^{\epsilon} \, ,$$

and Eq. (9) concludes the proof. $\qquad\square$

# D  AdaP-TT: A private Top Two algorithm with adaptive episodes

We propose a generic wrapper to adapt existing BAI algorithms to tackle private BAI (Appendix D.1). Then, we show how to instantiate our wrapper with an instance of Top Two algorithm (Appendix D.2), namely TTUCB [JD22].

## D.1  Generic wrapper on existing BAI algorithms

We propose a generic wrapper to adapt existing BAI algorithms to tackle the $\epsilon$-DP-FC-BAI problem.

1. It uses *adaptive episodes with doubling and forgetting* (Appendix D.1.1). This builds on the idea used to make the UCB algorithm private for regret minimisation [AB22].

2. It relies on a *private GLR (generalised likelihood ratio) stopping rule* (Appendix D.1.2). The GLR stopping rule [GK16] is widely used in the BAI literature, since it ensures $\delta$-correctness of the stopping rule regardless of the sampling rule.

For each arm $a \in [K]$, we maintain a phase at time $n$ whose index will be denoted by $k_{n,a} \in \mathbb{N}$. We switch phase as soon as the number of times that the arm was played is doubled. We only evaluate the stopping condition when we switch phase for an arm.

### D.1.1  Adaptive episodes with doubling and forgetting

As initialisation, we start by pulling each arm once, and set $k_{n,a} = 1$, $T_1(a) = K + 1$ and $N_{n,a} = 1$ for all $a \in [K]$. In the following, we will consider $n > K$ and we denote the *global pulling count* of arm $a$ before time $n$ by $N_{n,a} \triangleq \sum_{t \in [n-1]} \mathbb{1}(I_t = a)$. For each arm $a \in [K]$, the random stopping time denoting the end of $k_a - 1$ and the beginning of phase $k_a > 1$ is denoted by

$$T_{k_a}(a) \triangleq \inf\left\{ n \in \mathbb{N} \mid N_{n,a} \ge 2 N_{T_{k_a-1}(a),n} \right\} \, . \tag{10}$$

At the beginning of phase $k_a$ for an arm $a \in [K]$, we update the empirical mean based on the observations on arm $a \in [K]$ collected during this last phase, i.e.

$$\hat{\mu}_{k_a,a} \triangleq \frac{1}{\tilde{N}_{k_a,a}} \sum_{s=T_{k_a-1}(a)}^{T_{k_a}(a)-1} X_s \mathbb{1}\{I_s = a\} \, , \tag{11}$$

where the *local pulling count* pf arm $a$ is denoted by $\tilde{N}_{k_a,a} \triangleq N_{T_{k_a}(a),a} - N_{T_{k_a-1}(a),a}$, meaning it is the number of collected samples during the phase $k_a - 1$. To ensure privacy, we add a Laplace

noise to define the private empirical mean, i.e.

$$\tilde{\mu}_{k_a,a} \triangleq \hat{\mu}_{k_a,a} + Y_{k_a,a} \quad \text{where} \quad Y_{k_a,a} \sim \text{Lap}\left(\frac{1}{\epsilon \tilde{N}_{k_a,a}}\right). \tag{12}$$

We emphasise that only the private version the estimator $\hat{\mu}_{k_a,a}$, i.e. $\tilde{\mu}_{k_a,a}$, is used by the algorithm until the end of phase $k_a$ for arm $a$. Since $(k_{n,a})_{a \in [K]}$ denotes the current phases at time $n$, our algorithm relies on $(\hat{\mu}_{k_{n,a},a}, \tilde{\mu}_{k_a,a}, \tilde{N}_{k_{n,a},a}, N_{n,a})_{a \in [K]}$.

Due to the doubling, the growth of the global and local pulling counts is exponential (Lemma 3).

**Lemma 3.** *For all $a \in [K]$ and all $k \in \mathbb{N}$ such that $\mathbb{E}_{\boldsymbol{\nu}}[T_k(a)] < +\infty$, we have*

$$N_{T_k(a),a} = 2^{k-1} \quad and \quad \tilde{N}_{k,a} = 2^{k-2}.$$

*Proof.* Let $a \in [K]$. After initialisation, we have $k = 1$, $T_1(a) = K + 1$ and $N_{T_1(a),a} = 1$. Using the definition of the adaptive phase switch (Equation (10)), it is direct to see that $N_{T_2(a),a} = 2$ and $\tilde{N}_{2,a} = 1$ when $\mathbb{E}_{\boldsymbol{\nu}}[T_2(a)] < +\infty$.

Now, we proceed by recurrence. Suppose that $N_{T_k(a),a} = 2^{k-1}$ and $\tilde{N}_{k,a} = 2^{k-2}$ when $\mathbb{E}_{\boldsymbol{\nu}}[T_k(a)] < +\infty$. If $\mathbb{E}_{\boldsymbol{\nu}}[T_{k+1}(a)] < +\infty$, then it means that the phase $k$ ends for arm $a$ almost surely. Since we sample only one arm at each round, at the beginning of phase $k + 1$ for arm $a$, we have $N_{T_{k+1}(a),a} = 2N_{T_k(a),a} = 2^k$ by using the definition of the adaptive phase switch (10). Then, we have directly that $\tilde{N}_{k+1,a} = N_{T_{k+1}(a),a} - N_{T_k(a),a} = 2^k - 2^{k-1} = 2^{k-1}$. □

### D.1.2   GLR stopping rule

**Non-private GLR stopping rule with phases.**   Given a set of non-private threshold functions $(c_k)_{k \in \mathbb{N}}$, such that $c_k : \mathbb{N} \times \mathbb{N} \times (0,1) \to \mathbb{R}_+$ for all $k \in \mathbb{N}$, the non-private GLR stopping rule can be evaluated at the beginning of each phase for each arm, namely

$$\tau_\delta^{\text{NP}} = \inf\left\{ n \in \mathbb{N} \mid \forall b \neq \hat{a}_n^{\text{NP}}, \ \frac{(\hat{\mu}_{k_{n,\hat{a}_n^{\text{NP}}},\hat{a}_n^{\text{NP}}} - \hat{\mu}_{k_{n,b},b})^2}{1/\tilde{N}_{k_{n,\hat{a}_n^{\text{NP}}},\hat{a}_n^{\text{NP}}} + 1/\tilde{N}_{k_{n,b},b}} \geq 2c_{k_{n,\hat{a}_n^{\text{NP}}}k_{n,b}}(\tilde{N}_{k_{n,\hat{a}_n^{\text{NP}}},\hat{a}_n^{\text{NP}}}, \tilde{N}_{k_{n,b},b}, \delta) \right\}, \tag{13}$$

where $\hat{a}_n^{\text{NP}} = \arg\max_{a \in [K]} \hat{\mu}_{k_{n,a},a}$ is the non-private candidate answer until we switch phase again (for any arm). We emphasise that this *stopping condition is only evaluated at the beginning of each phase for each arm since it involves quantities that are fixed until we switch phase again.*

Lemma 4 gives non-private threshold functions ensuring that the non-private GLR stopping rule is $\delta$-correct for all $\delta \in (0,1)$, independently of the sampling rule.

**Lemma 4.** *Let $\delta \in (0,1)$. Let $s > 1$ and $\zeta$ be the Riemann $\zeta$ function. Given any sampling rule, the non-private GLR stopping rule (Equation (13)) with non-private threshold functions*

$$c_k(n,m,\delta) = 2\mathcal{C}_G\left(\frac{1}{2}\log\left(\frac{(K-1)\zeta(s)^2 k^s}{\delta}\right)\right) + 2\log(4 + \log n) + 2\log(4 + \log m) \tag{14}$$

*ensures $\delta$-correctness for 1-sub-Gaussian distributions. The function $\mathcal{C}_G$ is defined in Equation (15). It satisfies $\mathcal{C}_G \approx x + \log x$.*

*Proof.* The non-private GLR stopping rule matches the one used for Gaussian bandits. Proving $\delta$-correctness of a GLR stopping rule is done by leveraging concentration results. In particular, we build upon Theorem 9 of [KK21], which is restated below.

**Lemma 5.** *Consider sub-Gaussian bandits with means $\boldsymbol{\mu} \in \mathbb{R}^K$. Let $S \subseteq [K]$ and $x > 0$.*

$$\mathbb{P}_{\boldsymbol{\nu}}\left[\exists n \in \mathbb{N}, \sum_{k \in S} \frac{N_{n,k}}{2}(\mu_{n,k} - \mu_k)^2 > \sum_{k \in S} 2\log(4 + \log(N_{n,k})) + |S|\mathcal{C}_G\left(\frac{x}{|S|}\right)\right] \leq e^{-x},$$

*where $\mathcal{C}_G$ is defined in [KK21] as*

$$\mathcal{C}_G(x) \triangleq \min_{\lambda \in ]1/2,1]} \frac{g_G(\lambda) + x}{\lambda} \quad and \quad g_G(\lambda) \triangleq 2\lambda - 2\lambda \log(4\lambda) + \log \zeta(2\lambda) - \frac{1}{2}\log(1-\lambda) \ .$$

(15)

*Here, $\zeta$ is the Riemann $\zeta$ function and $\mathcal{C}_G(x) \approx x + \log(x)$.*

Let $\hat{a} = \hat{a}_n^{\mathrm{NP}} = \arg\max_{b \in [K]} \hat{\mu}_{k_{n,a},a}$. Standard manipulations yield that for all $b \neq \hat{a}$

$$\frac{(\hat{\mu}_{k_{n,\hat{a}},\hat{a}} - \hat{\mu}_{k_{n,b},b})^2}{1/\tilde{N}_{k_{n,\hat{a}},\hat{a}} + 1/\tilde{N}_{k_{n,b},b}} = \inf_{y \geq x} \left\{ \tilde{N}_{k_{n,\hat{a}},\hat{a}}(\hat{\mu}_{k_{n,\hat{a}},\hat{a}} - x)^2 + \tilde{N}_{k_{n,b},b}(\hat{\mu}_{k_{n,b},b} - y)^2 \right\} \ .$$

Using the non-private GLR stopping rule (13) with non-private threshold functions $(c_k)_{k \in \mathbb{N}}$ and the above manipulations, we obtain

$$\mathbb{P}_{\boldsymbol{\nu}}\left(\tau_\delta^{\mathrm{NP}} < +\infty, \hat{a} \neq a^\star\right)$$

$$\leq \mathbb{P}_{\boldsymbol{\nu}}\left(\exists n \in \mathbb{N}, \exists a \neq a^\star, \ a = \arg\max_{b \in [K]} \hat{\mu}_{k_{n,b},b}, \ \forall b \neq a, \right.$$

$$\left. \inf_{y \geq x}\left\{ \tilde{N}_{k_{n,a},a}(\hat{\mu}_{k_{n,a},a} - x)^2 + \tilde{N}_{k_{n,b},b}(\hat{\mu}_{k_{n,b},b} - y)^2 \right\} \geq 2c_{k_{n,a}k_{n,b}}(\tilde{N}_{k_{n,a},a}, \tilde{N}_{k_{n,b},b}, \delta) \right)$$

$$\underset{(a)}{\leq} \mathbb{P}_{\boldsymbol{\nu}}\left(\exists n \in \mathbb{N}, \exists a \neq a^\star, \ a = \arg\max_{b \in [K]} \hat{\mu}_{k_{n,a},a}, \right.$$

$$\left. \tilde{N}_{k_{n,a},a}(\hat{\mu}_{k_{n,a},a} - \mu_a)^2 + \tilde{N}_{k_{n,a^\star},a^\star}(\hat{\mu}_{k_{n,a^\star},a^\star} - \mu_{a^\star})^2 \geq 2c_{k_{n,a}k_{n,a^\star}}(\tilde{N}_{k_{n,a},a}, \tilde{N}_{k_{n,a^\star},a^\star}, \delta) \right)$$

$$\underset{(b)}{\leq} \mathbb{P}_{\boldsymbol{\nu}}\left(\exists a \neq a^\star, \exists(k_a, k_{a^\star}) \in \mathbb{N}^2, \right.$$

$$\left. \tilde{N}_{k_a,a}(\hat{\mu}_{k_a,a} - \mu_a)^2 + \tilde{N}_{k_{a^\star},a^\star}(\hat{\mu}_{k_{a^\star},a^\star} - \mu_{a^\star})^2 \geq 2c_{k_a k_{a^\star}}(\tilde{N}_{k_a,a}, \tilde{N}_{k_{a^\star},a^\star}, \delta) \right)$$

$$\underset{(c)}{\leq} \sum_{a \neq a^\star} \sum_{(k_a, k_{a^\star}) \in \mathbb{N}^2} \mathbb{P}_{\boldsymbol{\nu}}\left( \tilde{N}_{k_a,a}(\hat{\mu}_{k_a,a} - \mu_a)^2 + \tilde{N}_{k_{a^\star},a^\star}(\hat{\mu}_{k_{a^\star},a^\star} - \mu_{a^\star})^2 \right.$$

$$\left. \geq 2c_{k_a k_{a^\star}}(\tilde{N}_{k_a,a}, \tilde{N}_{k_{a^\star},a^\star}, \delta) \right) \ ,$$

The inequality (a) is obtained with $(b, x, y) = (a^\star, \mu_a, \mu_{a^\star})$. The inequality (b) drops the condition $a = \arg\max_{b \in [K]} \hat{\mu}_{k_b,b}$, hence we can restrict to $(k_a, k_{a^\star}) \in \mathbb{N}^2$ since it doesn't depend on other phase indices. The inequality (c) relies on a direct union bound. For all $a \neq a^\star$ and all $(k_a, k_{a^\star}) \in \mathbb{N}^2$, the estimators $\hat{\mu}_{k_a,a}$ (resp. $\hat{\mu}_{k_{a^\star},a^\star}$) are based solely on the observations collected for arm $a$ (resp. arm $a^\star$) between times $n \in \{T_{k_a-1}(a), \cdots, T_{k_a}(a) - 1\}$ (resp. $n \in \{T_{k_{a^\star}-1}(a^\star), \cdots, T_{k_{a^\star}}(a^\star) - 1\}$) with local counts $\tilde{N}_{k_a,a}$ (resp. $\tilde{N}_{k_{a^\star},a^\star}$), i.e. dropping past observations. Using Lemma 5 for all $a \neq a^\star$ and all $(k_a, k_{a^\star}) \in \mathbb{N}^2$, we obtain

$$\mathbb{P}_{\boldsymbol{\nu}}\left(\tau_\delta < +\infty, \hat{i} = i^\star\right) \leq \frac{\delta}{K-1}\frac{1}{\zeta(s)^2} \sum_{a \neq a^\star} \sum_{(k_a, k_{a^\star}) \in \mathbb{N}^2} \frac{1}{(k_a k_{a^\star})^s} = \delta \ .$$

$\square$

**Private GLR stopping rule with phases.** Since we want to ensure privacy, the non-private GLR stopping rule cannot be used since it relies on the empirical means $\hat{\mu}_{k_{n,a},a}$, which are not private. To alleviate this problem, we propose the private GLR stopping rule, which is based on the private empirical means $\tilde{\mu}_{k_{n,a},a}$.

Given a set of private threshold functions $(c_{\epsilon,k_1,k_2})_{(\epsilon,k_1,k_2) \in \mathbb{R}_+^\star \times \mathbb{N}^2}$, such that $c_{\epsilon,k_1,k_2} : \mathbb{N} \times \mathbb{N} \times (0,1) \to \mathbb{R}_+$ for all $(\epsilon, k_1, k_2) \in \mathbb{R}_+^\star \times \mathbb{N}^2$, the non-private GLR stopping rule is evaluated at the beginning of each phase for each arm, namely

$$\tau_\delta = \inf\left\{ n \in \mathbb{N} \mid \forall b \neq \hat{a}_n, \ \frac{(\tilde{\mu}_{k_{n,\hat{a}_n},\hat{a}_n} - \tilde{\mu}_{k_{n,b},b})^2}{1/\tilde{N}_{k_{n,\hat{a}_n},\hat{a}_n} + 1/\tilde{N}_{k_{n,b},b}} \geq 2c_{\epsilon,k_{n,\hat{a}_n},k_{n,b}}(\tilde{N}_{k_{n,\hat{a}_n},\hat{a}_n}, \tilde{N}_{k_{n,b},b}, \delta) \right\} \ ,$$

(16)

where $\hat{a}_n = \arg\max_{a \in [K]} \tilde{\mu}_{k_{n,a},a}$ is the private candidate answer until we switch phase again (for any arm). We emphasise that this stopping condition is only evaluated at the beginning of each phase for each arm since it involves quantities that are fixed until we switch phase again.

Theorem 4 gives private threshold functions ensuring that the private GLR stopping rule is $\delta$-correct for all $\delta \in (0,1)$ and all $\epsilon \in \mathbb{R}^\star_+$, independently of the sampling rule.

**Theorem 4** ($\delta$-correctness of the private GLR stopping rule). *Let $\delta \in (0,1)$ and $\epsilon \in \mathbb{R}^\star_+$. Let $s > 1$ and $\zeta$ be the Riemann $\zeta$ function and non-private threshold function $(c_k)_{k \in \mathbb{N}}$ as in (14). Given any sampling rule, the private GLR stopping rule (Equation (16)) with private threshold functions*

$$c_{\epsilon,k_1,k_2}(n,m,\delta) = 2c_{k_1 k_2}(n,m,\delta/2) + \frac{1}{n\epsilon^2}\log\left(\frac{2Kk_1^s\zeta(s)}{\delta}\right)^2 + \frac{1}{m\epsilon^2}\log\left(\frac{2Kk_2^s\zeta(s)}{\delta}\right)^2 \quad (17)$$

*ensures $\delta$-correctness for 1-sub-Gaussian distributions.*

*Proof.* Let $\epsilon \in \mathbb{R}^\star_+$. Since $Y_{k_{n,a},a} \sim \mathrm{Lap}\left((\epsilon \tilde{N}_{k_{n,a},a})^{-1}\right)$, we have that $\tilde{N}_{k_{n,a},a}|Y_{k_{n,a},a}| \sim \mathcal{E}(\epsilon)$ for all $a \in [K]$ and all $n \in \mathbb{N}$, where $\mathcal{E}(\cdot)$ denotes the exponential distribution. Using concentration results for exponential distribution, a direct union bound yields that

$$\mathbb{P}\left(\exists n \in \mathbb{N}, \exists a \in [K], \tilde{N}_{k_{n,a},a}|Y_{k_{n,a},a}| \geq \frac{1}{\epsilon}\log\left(\frac{Kk_{n,a}^s\zeta(s)}{\delta}\right)\right) \leq \delta. \quad (18)$$

Let us denote $\tilde{c}_{\epsilon,k_1,k_2}(n,m,\delta)$ the threshold associated to the Laplace noise, i.e.

$$\tilde{c}_{\epsilon,k_1,k_2}(n,m,\delta) = \frac{1}{n\epsilon^2}\log\left(\frac{Kk_1^s\zeta(s)}{\delta}\right)^2 + \frac{1}{m\epsilon^2}\log\left(\frac{Kk_2^s\zeta(s)}{\delta}\right)^2.$$

Using the private GLR stopping rule (Equation (16)) with private threshold functions $(c_{\epsilon,k_1,k_2})_{(\epsilon,k_1,k_2) \in \mathbb{R}^\star_+ \times \mathbb{N}^2}$, similar manipulations as above yields

$\mathbb{P}_{\boldsymbol{\nu}}(\tau_\delta < +\infty, \hat{a} \neq a^\star)$

$\leq \mathbb{P}_{\boldsymbol{\nu}}\left(\exists n \in \mathbb{N}, \exists a \neq a^\star, \tilde{N}_{k_{n,a},a}(\hat{\mu}_{k_{n,a},a} - \mu_a + Y_{k_{n,a},a})^2 + \tilde{N}_{k_{n,a^\star},a^\star}(\hat{\mu}_{k_{n,a^\star},a^\star} - \mu_{a^\star} + Y_{k_{n,a^\star},a^\star})^2\right.$

$\qquad\qquad \left. \geq 4c_{k_{n,a}k_{n,a^\star}}(\tilde{N}_{k_{n,a},a}, \tilde{N}_{k_{n,a^\star},a^\star}, \delta/2) + 2\tilde{c}_{\epsilon,k_{n,a},k_{n,a^\star}}(\tilde{N}_{k_{n,a},a}, \tilde{N}_{k_{n,a^\star},a^\star}, \delta/2)\right)$

$\underset{(a)}{\leq} \mathbb{P}_{\boldsymbol{\nu}}\left(\exists a \neq a^\star, \exists(k_a, k_{a^\star}) \in \mathbb{N}^2,\right.$

$\qquad\qquad \left. \tilde{N}_{k_a,a}(\hat{\mu}_{k_a,a} - \mu_a)^2 + \tilde{N}_{k_{a^\star},a^\star}(\hat{\mu}_{k_{a^\star},a^\star} - \mu_{a^\star})^2 \geq 2c_{k_a k_{a^\star}}(\tilde{N}_{k_a,a}, \tilde{N}_{k_{a^\star},a^\star}, \delta/2)\right)$

$\qquad + \mathbb{P}_{\boldsymbol{\nu}}\left(\exists n \in \mathbb{N}, \exists a \neq a^\star, \tilde{N}_{k_{n,a},a}Y_{k_{n,a},a}^2 + \tilde{N}_{k_{n,a^\star},a^\star}Y_{k_{n,a^\star},a^\star}^2\right.$

$\qquad\qquad \left. \geq \tilde{c}_{\epsilon,k_{n,a},k_{n,a^\star}}(\tilde{N}_{k_{n,a},a}, \tilde{N}_{k_{n,a^\star},a^\star}, \delta/2)\right)$

$\underset{(b)}{\leq} \delta/2 + \mathbb{P}\left(\exists n \in \mathbb{N}, \exists a \in [K], \tilde{N}_{k_{n,a},a}|Y_{k_{n,a},a}| \geq \frac{1}{\epsilon}\log\left(\frac{2Kk_{n,a}^s\zeta(s)}{\delta}\right)\right)$

$\underset{(c)}{\leq} \delta/2 + \delta/2 = \delta.$

The inequality (a) uses that $\mathbb{P}(X + Y \geq a + b) \leq \mathbb{P}(X \geq a) + \mathbb{P}(Y \geq b)$ and $(x - y)^2 \leq 2x^2 + 2y^2$. The inequality (c) leverages Lemma 4 and a direct inclusion of event. The inequality (c) deploys Equation (18) to conclude. $\qquad\square$

## D.2 AdaP-TT: Instantiating our wrapper with a Top Two sampling rule

**A blueprint of Top Two algorithm design.** At time $n > K$, a Top Two sampling rule defines a leader $B_n$ and a challenger $C_n$. Then, it selects among them the next arm to sample $I_n$. Given a proportion $\beta \in (0,1)$ fixed beforehand, the choice of $I_n \in \{B_n, C_n\}$ should ensure that the leader is sampled close to $\beta$ of the times where it was chosen as leader. In early works on Top Two algorithms, this choice is randomised. Following [JD22], we use $K$ independent tracking procedures.

We denote by $N_{n,b}^a \triangleq \sum_{t \in [n-1]} \mathbb{1}(B_t = a, I_t = C_t = b)$ the number of times the arm $b$ was pulled while the arm $a$ was the leader, and by $L_{n,a} \triangleq \sum_{t \in [n-1]} \mathbb{1}(B_t = a)$ the number of times arm $a$ was the leader. At time $n > K$, the next arm to be pulled $I_n$ is defined as

$$I_n = B_n \quad \text{if } N_{n,B_n}^{B_n} \le \beta L_{n+1,B_n} \text{ , otherwise} \quad I_n = C_n \text{ .} \tag{19}$$

In other words, we sample the leader if we have not yet sampled it a fraction $\beta$ of the times it was leader. Those $K$ independent tracking procedure satisfy the desired property (Lemma 6).

**Lemma 6** (Lemma 2.2 in [JD22]). *For all $n > K$ and all $a \in [K]$, we have*

$$-1/2 \le N_{n,a}^a - \beta L_{n,a} \le 1 \text{ .}$$

To finish specifying a Top Two algorithm, we simply need to specify the choice of the *leader/challenger pair*. Intuitively, a good choice of the leader/challenger pair should ensure (1) *sufficient exploration*, (2) *convergence of the leader towards the best arm $a^\star$*, and (3) *convergence of the global pulling proportions to the $\beta$-optimal allocation* $\omega_{\mathrm{KL},\beta}(\mu)$, which is defined for Gaussian distributions as

$$\omega_{\mathrm{KL},\beta}^\star(\boldsymbol{\mu}) \triangleq \operatorname*{arg\,max}_{\omega \in \Sigma_K, \omega_{a^\star} = \beta} \min_{a \ne a^\star} \frac{\Delta_a^2}{1/\beta + 1/\omega_a} \text{ .}$$

While we consider TTUCB algorithm [JD22] in AdaP-TT, we emphasise that other Top Two algorithms could be used with the same type of guarantees. TTUCB is a Top Two algorithm which combines a UCB-based leader and a Transportation Cost (TC) challenger. Its key novelty lies in the use of $K$ tracking procedures. Since it is deterministic, the analysis is less cumbersome.

**Non-private leader/challenger pair.** The non-private leader/challenger pair is inspired by the TTUCB algorithm [JD22]. At time $n > K$, the non-private UCB leader is defined as

$$B_n^{\mathrm{NP}} \triangleq \operatorname*{arg\,max}_{a \in [K]} \left\{ \hat{\mu}_{k_{n,a},a} + \sqrt{\frac{k_{n,a}}{\tilde{N}_{k_{n,a},a}}} \right\} \text{ ,} \tag{20}$$

where $\sqrt{k_{n,a}/\tilde{N}_{k_{n,a},a}}$ is a bonus to cope for the uncertainty which depends on the current phase $k_{n,a}$. Later, we show that $\log(n)/k_{n,a} = \Theta(1)$ for all $a \in [K]$. Hence, it has the same scaling as the bonus used in standard UCB. Since it depends solely on the local counts $(\tilde{N}_{k_{n,a},a})_{a \in [K]}$ and the empirical means $(\hat{\mu}_{k_{n,a},a})_{a \in [K]}$, the non-private UCB leader is fixed until we switch phase again.

At time $n > K$, the non-private TC challenger is defined as

$$C_n^{\mathrm{NP}} \triangleq \operatorname*{arg\,min}_{a \ne B_n^{\mathrm{NP}}} \frac{\hat{\mu}_{k_{n,B_n^{\mathrm{NP}}},B_n^{\mathrm{NP}}} - \hat{\mu}_{k_{n,a},a}}{\sqrt{1/N_{n,B_n^{\mathrm{NP}}} + 1/N_{n,a}}} \text{ .} \tag{21}$$

While it depends on the empirical means $(\hat{\mu}_{k_{n,a},a})_{a \in [K]}$ that are fixed till we switch phase again, it also depends on the global counts $(N_{n,a})_{a \in [K]}$. Therefore, the non-private TC challenger is chosen in an adaptive manner. This is the key to obtain guarantees on the expected sample complexity of the non-private algorithm.

We derive upper bounds on the expected sample complexity of the non-private AdaP-TT algorithm in the asymptotic regime of $\delta \to 0$ (Theorem 6). In particular, it shows that the cost of doubling and forgetting is multiplicative four-factor compared to the TTUCB algorithm, which achieves $T_{\mathrm{KL},\beta}^\star(\mu)$ (see Theorem 2.3 in [JD22]). We defer its proof to Appendix E.8.

**Theorem 6** (Asymptotic upper bound on expected sample complexity of non-private AdaP-TT). *Let $(\delta, \beta) \in (0,1)^2$. Combined with the non-private GLR stopping rule (Equation (13)) using non-private threshold functions as in Equation (14), the non-private AdaP-TT algorithm is $\delta$-correct and satisfies that, for all 1-sub-Gaussian distributions $\nu$ with means $\boldsymbol{\mu} \in \mathbb{R}^K$ such that $\min_{a \ne b} |\mu_a - \mu_b| > 0$,*

$$\limsup_{\delta \to 0} \frac{\mathbb{E}_{\boldsymbol{\nu}}[\tau_\delta^{NP}]}{\log(1/\delta)} \le 4 T_{\mathrm{KL},\beta}^\star(\mu) \text{ ,}$$

*where $T_{\mathrm{KL},\beta}^\star(\mu)$ is the $\beta$-characteristic time for Gaussian distributions, such that*

$$2 T_{\mathrm{KL},\beta}^\star(\boldsymbol{\mu})^{-1} \triangleq \max_{\omega \in \Sigma_K, \omega_{a^\star} = \beta} \frac{(\mu_{a^\star} - \mu_a)^2}{1/\beta + 1/\omega_a} \text{ .}$$

**Private leader/challenger pair.** Since we want to ensure privacy, the non-private leader/challenger pair cannot be used since it relies on the empirical means $\hat{\mu}_{k_{n,a},a}$, which are not private. To alleviate this problem, we propose a private leader/challenger pair which is based on the private empirical means $\tilde{\mu}_{k_{n,a},a}$.

At time $n > K$, the private UCB leader is defined as

$$B_n \triangleq \arg\max_{a \in [K]} \left\{ \tilde{\mu}_{k_{n,a},a} + \sqrt{\frac{k_{n,a}}{\tilde{N}_{k_{n,a},a}}} + \frac{k_{n,a}}{\epsilon \tilde{N}_{k_{n,a},a}} \right\} , \tag{22}$$

where $k_{n,a}/(\epsilon \tilde{N}_{k_{n,a},a})$ is a bonus to cope for the uncertainty due to the Laplace noise. It also depends on the current phase $k_{n,a}$ and has the same scaling as the private UCB indices. Likewise, the private UCB leader is fixed until we switch phase again.

At time $n > K$, the private Transportation Cost (TC) challenger is defined as

$$C_n \triangleq \arg\min_{a \neq B_n} \frac{\tilde{\mu}_{k_{n,B_n},B_n} - \tilde{\mu}_{k_{n,a},a}}{\sqrt{1/N_{n,B_n} + 1/N_{n,a}}} . \tag{23}$$

Likewise, the private TC challenger is chosen in an adaptive manner, which is key to obtain guarantees on the expected sample complexity of the private algorithm.

We derive upper bounds on the expected sample complexity of the private AdaP-TT algorithm in the asymptotic regime of $\delta \to 0$ (Theorem 5). We defer its proof to Appendix E.

**Theorem 5** (Asymptotic upper bound on expected sample complexity of AdaP-TT). *Let $(\delta, \beta) \in (0,1)^2$. Combined with the non-private GLR stopping rule (Equation* (13)*) using non-private threshold functions as in Equation* (14)*, the non-private* AdaP-TT *algorithm is $\delta$-correct and satisfies that, for all bandit instances $\nu$ with 1-sub-Gaussian distributions and means $\mu \in \mathbb{R}^K$ such that $\min_{a \neq b} |\mu_a - \mu_b| > 0$,*

$$\limsup_{\delta \to 0} \frac{\mathbb{E}_\nu[\tau_\delta]}{\log(1/\delta)} \leq 4 T^\star_{\mathrm{KL},\beta}(\mu) \left( 1 + \sqrt{1 + \frac{\Delta^2_{\max}}{2\epsilon^2}} \right) ,$$

*where $T^\star_{\mathrm{KL},\beta}(\mu)$ is the $\beta$-characteristic time for Gaussian distributions.*

# E    Analysis of AdaP-TT: Proof of Theorem 5

Let $\beta \in (0,1)$, $\epsilon \in \mathbb{R}^\star_+$, and $\nu$ be a bandit instance consisting of $K$, 1-sub-Gaussian distributions with distinct means $\mu \in \mathbb{R}^K$, i.e. $\min_{a \neq b} |\mu_a - \mu_b| > 0$. For conciseness, we denote $\Delta_a \triangleq \mu_{a^\star} - \mu_a$, $\Delta_{\min} \triangleq \min_{a \neq a^\star} \Delta_a$, and $\Delta_{\max} \triangleq \max_{a \neq a^\star} \Delta_a$.

For Gaussian distributions, the unique $\beta$-optimal allocation $\omega^\star_{\mathrm{KL},\beta}(\mu) = \{\omega^\star_{\beta,a}\}$ is defined as

$$\omega^\star_{\mathrm{KL},\beta}(\mu) \triangleq \arg\max_{\omega \in \Sigma_K, \omega_{a^\star} = \beta} \min_{a \neq a^\star} \frac{\Delta^2_a}{1/\beta + 1/\omega_a} . \tag{24}$$

At equilibrium, we have equality of the transportation costs (see [JD22] for example), namely

$$\forall a \neq a^\star, \quad \frac{\Delta^2_a}{1/\beta + 1/\omega^\star_{\beta,a}} = 2 T^\star_{\mathrm{KL},\beta}(\mu)^{-1} . \tag{25}$$

Our proof follows the unified sample complexity analysis of Top Two algorithms from [JDB+22].

Let $\gamma > 0$. We denote by $T_{\mu,\gamma}$ the *convergence time* towards $\omega^\star_\beta$, which is a random variable quantifies the number of samples required for the global empirical allocations $N_n/(n-1)$ to be $\gamma$-close to $\omega^\star_\beta$ for any subsequent time, namely

$$T_{\mu,\gamma} \triangleq \inf \left\{ T \geq 1 \mid \forall n \geq T, \left\| \frac{N_n}{n-1} - \omega^\star_\beta \right\|_\infty \leq \gamma \right\} . \tag{26}$$

The rest of Appendix E is organised as follows. First, we prove that AdaP-TT ensures sufficient exploration (Appendix E.2) Second, we prove that there is convergence towards the $\beta$-optimal

allocation (Appendix E.3) in finite time. Finally, we conclude the proof of Theorem 5 (Appendix E.5). In Appendix E.6, we compare our asymptotic upper bound with our asymptotic lower bound on the expected sample complexity (Theorem 2). In Appendix E.7, we discuss the limitation of our result and pose an open problem. Appendix E.8 will detail the slight modification that needs to be made in order to obtain Theorem 6.

## E.1   Technical results

Before delving into the proofs, we first recall some useful technical results extracted from the literature.

**Concentration results.**   In order to control the randomness of $(\tilde{\mu}_{k_a,a})_{a\in[K]}$, we use a standard concentration result on the empirical mean of sub-Gaussian random variables and on sub-exponential observations (Lemma 7). Since Bernoulli distributions are $1/2$-sub-Gaussian and the absolute value of a Laplace is an exponential distribution, Lemma 7 applies to our setting.

**Lemma 7.** *There exists a sub-Gaussian random variable $W_\mu$ such that, almost surely,*

$$\forall a \in [K], \ \forall k_a \in \mathbb{N}, \quad |\hat{\mu}_{k_a,a} - \mu_a| \leq W_\mu \sqrt{\frac{\log(e + \tilde{N}_{k_a,a})}{\tilde{N}_{k_a,a}}} \ .$$

*There exists a sub-exponential random variable $W_\epsilon$ such that, almost surely,*

$$\forall a \in [K], \ \forall k_a \in \mathbb{N}, \quad |Y_{k_a,a}| \leq W_\epsilon \frac{\log(e + k_a)}{\tilde{N}_{k_a,a}} \ .$$

*In particular, any random variable which is polynomial in $(W_\epsilon, W_\mu)$ has a finite expectation.*

*Proof.* The first part is a known result, e.g. Appendix E.2 in [JDB$^+$22]. Let us define

$$W_\epsilon \triangleq \sup_{a\in[K]} \sup_{k_a\in\mathbb{N}} \frac{\tilde{N}_{k_a,a}|Y_{k_a,a}|}{\log(e + k_a)} \ .$$

By definition, we have that, almost surely,

$$\forall a \in [K], \ \forall k_a \in \mathbb{N}, \quad |Y_{k_a,a}| \leq W_\epsilon \frac{\log(e + k_a)}{\tilde{N}_{k_a,a}} \ .$$

Since $\tilde{N}_{k,i}|Y_{k,i}| \sim \mathcal{E}(\epsilon)$, Lemma 72 in [JDB$^+$22] yields that $W_\epsilon$ is a sub-exponential random variable. Since $W_\mu$ is sub-Gaussian and $W_\epsilon$ is a sub-exponential, any random variable which is polynomial in $(W_\epsilon, W_\mu)$ has a finite expectation. $\square$

**Inversion results.**   Lemma 8 gathers properties on the function $\overline{W}_{-1}$, which is used in the literature to obtain concentration results.

**Lemma 8** ([JDK23]). *Let $\overline{W}_{-1}(x) \triangleq -W_{-1}(-e^{-x})$ for all $x \geq 1$, where $W_{-1}$ is the negative branch of the Lambert W function. The function $\overline{W}_{-1}$ is increasing on $(1, +\infty)$ and strictly concave on $(1, +\infty)$. In particular, $\overline{W}'_{-1}(x) = \left(1 - \frac{1}{\overline{W}_{-1}(x)}\right)^{-1}$ for all $x > 1$. Then, for all $y \geq 1$ and $x \geq 1$,*

$$\overline{W}_{-1}(y) \leq x \quad \Longleftrightarrow \quad y \leq x - \log(x) \ .$$

*Moreover, for all $x > 1$,*

$$x + \log(x) \leq \overline{W}_{-1}(x) \leq x + \log(x) + \min\left\{\frac{1}{2}, \frac{1}{\sqrt{x}}\right\} \ .$$

Lemma 9 is an inversion result to upper bound a time, which is implicitly defined. It is a direct consequence of Lemma 8.

**Lemma 9.** *Let $\overline{W}_{-1}$ defined in Lemma 8. Let $A > 0$, $B > 0$ such that $B/A + \log A > 1$ and*

$$C(A, B) = \sup\{x \mid x < A\log x + B\} \ .$$

*Then, $C(A, B) < h_1(A, B)$ with $h_1(z, y) = z\overline{W}_{-1}(y/z + \log z)$.*

*Proof.* Since $B/A + \log A > 1$, we have $C(A, B) \geq A$, hence

$$C(A, B) = \sup \{x \mid x < A\log(x) + B\} = \sup \{x \geq A \mid x < A\log(x) + B\} \ .$$

Using Lemma 8 yields that

$$x \geq A \log x + B \iff \frac{x}{A} - \log\left(\frac{x}{A}\right) \geq \frac{B}{A} + \log A \iff x \geq A\overline{W}_{-1}\left(\frac{B}{A} + \log A\right) \ .$$

□

### E.2 Sufficient exploration

The first step of in the generic analysis of Top Two algorithms [JDB+22] is to show that AdaP-TT ensures sufficient exploration. The main idea is to show that, if there are still undersampled arms, either the leader or the challenger will be among them. Therefore, after a long enough time, no arm can still be undersampled. We emphasise that there are multiple ways to select the leader/challenger pair in order to ensure sufficient exploration. Therefore, while we conduct the proof for AdaP-TT, other choices of leader/challenger pair would yield similar results.

Given an arbitrary phase $p \in \mathbb{N}$, we define the sampled enough set, i.e. the arms having reached phase $p$, and the arm with highest mean in this set (when not empty) as

$$S_n^p = \{a \in [K] \mid N_{n,a} \geq 2^{p-1}\} \quad \text{and} \quad a_n^\star = \arg\max_{a \in S_n^p} \mu_a \ . \tag{27}$$

Since $\min_{a \neq b} |\mu_a - \mu_b| > 0$, $a_n^\star$ is unique.

Let $p \in \mathbb{N}$ such that $(p-1)/4 \in \mathbb{N}$. We define the highly and the mildly under-sampled sets as

$$U_n^p \triangleq \{a \in [K] \mid N_{n,a} < 2^{(p-1)/2}\} \quad \text{and} \quad V_n^p \triangleq \{a \in [K] \mid N_{n,a} < 2^{3(p-1)/4}\} \ . \tag{28}$$

They correspond to the arms having not reached phase $(p-1)/2$ and phase $3(p-1)/4$, respectively.

**Lemma 10 shows that, when the leader is sampled enough, it is the arm with highest true mean among the sampled enough arms.**

**Lemma 10.** *Let $S_n^p$ and $a_n^\star$ as in (27). There exists $p_0$ with $\mathbb{E}_{\boldsymbol{\nu}}[\exp(\alpha p_0)] < +\infty$ for all $\alpha > 0$ such that if $p \geq p_0$, for all $n$ such that $S_n^p \neq \emptyset$, $B_n \in S_n^p$ implies that $B_n = a_n^\star = \arg\max_{a \in S_n^p} \tilde{\mu}_{k_{n,a},a}$.*

*Proof.* Let $p_0$ to be specified later. Let $p \geq p_0$. Let $n \in \mathbb{N}$ such that $S_n^p \neq \emptyset$, where $S_n^p$ and $a_n^\star$ as in Equation (27). Let $(k_{n,a})_{a \in [K]}$ be the phases indices for all arms. Since $N_{n,a} \geq 2^{p-1}$ for all $a \in S_n^p$, we have $k_{n,a} \geq p$ and $\tilde{N}_{k_{n,a},a} \geq 2^{p-2}$ by using Lemma 3. Using Lemma 7, we obtain that

$$\tilde{\mu}_{k_{n,a_n^\star},a_n^\star} \geq \mu_{a_n^\star} - W_\mu\sqrt{\frac{\log(e + 2^{p-2})}{2^{p-2}}} - W_\epsilon\frac{\log(e+p)}{2^{p-2}} \ ,$$

$$\tilde{\mu}_{k_{n,a},a} \leq \mu_a + W_\mu\sqrt{\frac{\log(e + 2^{p-2})}{2^{p-2}}} + W_\epsilon\frac{\log(e+p)}{2^{p-2}} \ , \quad \forall a \in S_n^p \setminus \{a_n^\star\}.$$

Here, we use that $x \to \log(e + x)/x$ is decreasing.

Let $\overline{\Delta}_{\min} = \min_{a \neq b} |\mu_a - \mu_b|$. By assumption on the considered instances, we know that $\overline{\Delta}_{\min} > 0$. Let $p_1 = \lceil \log_2(X_1 - e) \rceil + 2$ and $p_2 = \lceil \log_2((X_2 - e - 2)\log 2 + 1) \rceil + 2$ with

$$X_1 = \sup\left\{x > 1 \mid x \leq 64\overline{\Delta}_{\min}^{-2}W_\mu^2 \log x + e\right\} \leq h_1(64\overline{\Delta}_{\min}^{-2}W_\mu^2, e) \ ,$$

$$X_2 = \sup\left\{x > 1 \mid x \leq \frac{8}{\log 2}\overline{\Delta}_{\min}^{-1}W_\epsilon \log x + e + 2 - 1/\log 2\right\} \leq h_1(8\overline{\Delta}_{\min}^{-1}W_\epsilon/\log 2, 4) \ ,$$

where we used Lemma 9, and $h_1$ defined therein. Then, for all $p \in \mathbb{N}$ such that $p \geq \max\{p_1, p_2\} + 1$ and all $n \in \mathbb{N}$ such that $S_n^p \neq \emptyset$, we have $\tilde{\mu}_{k_{n,a_n^\star},a_n^\star} \geq \mu_{a_n^\star} - \overline{\Delta}_{\min}/4$ and $\tilde{\mu}_{k_{n,a},a} \leq \mu_a + \overline{\Delta}_{\min}/4$ for all $a \in S_n^p \setminus \{a_n^\star\}$, hence $a_n^\star = \arg\max_{a \in [K]} \tilde{\mu}_{k_{n,a},a}$.

We have, for all $\alpha \in \mathbb{R}_+$,

$$\exp(\alpha p_1) \leq e^{3\alpha}(X_1 - e)^{\alpha/\log 2} \quad \text{hence} \quad \mathbb{E}_{\boldsymbol{\nu}}[\exp(\alpha p_1)] < +\infty \ ,$$

where we used Lemma 7 and $h_1(x, e) \sim_{x \to +\infty} x \log x$ to obtain that $\exp(\alpha p_1)$ is at most polynomial in $W_\mu$. Likewise, we obtain that $\mathbb{E}_\nu[\exp(\alpha p_2)] < +\infty$ for all $\alpha \in \mathbb{R}_+$.

Let us define the UCB indices by $I_{k_{n,a},a} = \tilde{\mu}_{k_{n,a},a} + \sqrt{k_{n,a}/\tilde{N}_{k_{n,a},a}} + k_{n,a}/(\epsilon \tilde{N}_{k_{n,a},a})$. Using the above, we have

$$I_{k_{n,a_n^\star},a_n^\star} \geq \mu_{a_n^\star} - W_\mu \sqrt{\frac{\log(e + 2^{p-2})}{2^{p-2}}} - W_\epsilon \frac{\log(e + p)}{2^{p-2}},$$

$$\forall a \in S_n^p \setminus \{a_n^\star\}, \quad I_{k_{n,a},a} \leq \mu_a + W_\mu \sqrt{\frac{\log(e + 2^{p-2})}{2^{p-2}}} + W_\epsilon \frac{\log(e + p)}{2^{p-2}} + \sqrt{\frac{p}{2^{p-2}}} + \frac{p}{\epsilon 2^{p-2}},$$

where we used Lemma 3 and the fact that $x \to \log(e + x)/x$ and $x \to x2^{2-x}$ are decreasing function for $x \geq 2$. Let $p_3 = \lceil \log_2 X_3 \rceil + 2$ and $p_4 = \lceil \log_2 X_4 \rceil + 2$ with

$$X_3 = \sup \left\{ x > 1 \mid x \leq 64\overline{\Delta}_{\min}^{-2}(\log_2 x + 2) \right\} \leq h_1(64\overline{\Delta}_{\min}^{-2}/\log 2, \ 128\overline{\Delta}_{\min}^{-2}),$$

$$X_4 = \sup \left\{ x > 1 \mid x \leq 8\epsilon^{-1}\overline{\Delta}_{\min}^{-1}(\log_2 x + 2) \right\} \leq h_1(8\epsilon^{-1}\overline{\Delta}_{\min}^{-1}/\log 2, \ 16\overline{\Delta}_{\min}^{-1}\epsilon^{-1}),$$

where we used Lemma 9, and $h_1$ defined therein. We highlight that $(p_3, p_4)$ are deterministic values, hence their expectation is finite. Then, for all $p \in \mathbb{N}$ such that $p \geq p_0 = \max\{p_1, p_2, p_3, p_4\} + 1$ and all $n \in \mathbb{N}$ such that $S_n^p \neq \emptyset$, we have $I_{k_{n,a_n^\star},a_n^\star} \geq \mu_{a_n^\star} - \overline{\Delta}_{\min}/4$ and $I_{k_{n,a},a} \leq \mu_a + \overline{\Delta}_{\min}/2$ for all $a \in S_n^p \setminus \{a_n^\star\}$, hence $a_n^\star = B_n$ since we have $B_n = \arg\max_{a \in [K]} I_{k_{n,a},a}$.

Since we have $\mathbb{E}_\nu[\exp(\alpha p_0)] < +\infty$ for all $\alpha \in \mathbb{R}_+$, this concludes the proof. $\square$

**Lemma 11 shows that the transportation costs between the sampled enough arms with largest true means and the other sampled enough arms are increasing fast enough.**

**Lemma 11.** *Let $S_n^p$ and $a_n^\star$ are as in Equation (27). There exists $p_1$ with $\mathbb{E}_\nu[\exp(\alpha p_1)] < +\infty$ for all $\alpha > 0$ such that if $p \geq p_1$, for all $n$ such that $S_n^p \neq \emptyset$, for all $b \in S_n^p \setminus \{a_n^\star\}$, we have*

$$\frac{\tilde{\mu}_{k_{n,a_n^\star},a_n^\star} - \tilde{\mu}_{k_{n,b},b}}{\sqrt{1/\tilde{N}_{k_{n,a_n^\star},a_n^\star} + 1/\tilde{N}_{k_{n,b},b}}} \geq 2^{p/2} C_\mu,$$

*where $C_\mu > 0$ is a problem dependent constant.*

*Proof.* Let $p_1$ to be specified later. Let $p \geq p_1$. Let $n \in \mathbb{N}$ such that $S_n^p \neq \emptyset$, where $S_n^p$ and $a_n^\star$ as in Equation (27). Let $(k_{n,a})_{a \in [K]}$ be the phases indices for all arms. Since $N_{n,a} \geq 2^{p-1}$ for all $a \in S_n^p$, we have $k_{n,a} \geq p$ and $\tilde{N}_{k_{n,a},a} \geq 2^{p-2}$ by using Lemma 3. Let $\overline{\Delta}_{\min} = \min_{a \neq b} |\mu_a - \mu_b|$, which satisfies $\overline{\Delta}_{\min} > 0$ by assumption on the instance considered.

Using Lemma 7, for all $b \in S_n^p \setminus \{a_n^\star\}$, we obtain

$$\tilde{\mu}_{k_{n,a_n^\star},a_n^\star} - \tilde{\mu}_{k_{n,b},b} \geq \overline{\Delta}_{\min} - W_\mu \sqrt{\frac{\log(e + 2^{p-2})}{2^{p-4}}} - W_\epsilon \frac{\log(e + p)}{2^{p-3}}.$$

Let $p_3 = \lceil \log_2((X_3 - e)/4) \rceil + 4$ and $p_2 = \lceil \log_2((X_2 - e - 3)\log 2 + 1) \rceil + 3$ with

$$X_3 = \sup \left\{ x > 1 \mid x \leq 64\overline{\Delta}_{\min}^{-2} W_\mu^2 \log x + e \right\} \leq h_1(64\overline{\Delta}_{\min}^{-2} W_\mu^2, \ e),$$

$$X_2 = \sup \left\{ x > 1 \mid x \leq 4\overline{\Delta}_{\min}^{-1} W_\epsilon \log x + e + 3 - 1/\log 2 \right\} \leq h_1(4\overline{\Delta}_{\min}^{-1} W_\epsilon, \ 5),$$

where we used Lemma 9, and $h_1$ defined therein. Then, for all $p \in \mathbb{N}$ such that $p \geq p_1 = \max\{p_3, p_2\} + 1$ and all $n \in \mathbb{N}$ such that $S_n^p \neq \emptyset$, we have, for all $b \in S_n^p \setminus \{a_n^\star\}$,

$$\tilde{\mu}_{k_{n,a_n^\star},a_n^\star} - \tilde{\mu}_{k_{n,b},b} \geq \overline{\Delta}_{\min}/2.$$

As in the proof of Lemma 10, we obtain that $\mathbb{E}_\nu[\exp(\alpha p_1)] < +\infty$ for all $\alpha \in \mathbb{R}_+$.

Then, for all $b \in S_n^p \setminus \{a_n^\star\}$, we have

$$\frac{\tilde{\mu}_{k_{n,a_n^\star},a_n^\star} - \tilde{\mu}_{k_{n,b},b}}{\sqrt{1/\tilde{N}_{k_{n,a_n^\star},a_n^\star} + 1/\tilde{N}_{k_{n,b},b}}} \geq 2^{p/2} \frac{\overline{\Delta}_{\min}}{2^{5/2}},$$

where we used that $\min\{\tilde{N}_{k_{n,a_n^\star}}, \tilde{N}_{k_{n,b},b}\} \geq 2^{p-2}$. Setting $C_\mu = \overline{\Delta}_{\min}/2^{5/2}$ yields the result. $\square$

**Lemma 12 shows that the transportation costs between sampled enough arms and undersampled arms are not increasing too fast.**

**Lemma 12.** *Let $S_n^p$ be as in Equation (27). For all $p \geq 1$ and all $n$ such that $S_n^p \neq \emptyset$*

$$\forall a \in S_n^p, \forall b \notin S_n^p, \quad \frac{\tilde{\mu}_{k_{n,a},a} - \tilde{\mu}_{k_{n,b},b}}{\sqrt{1/\tilde{N}_{k_{n,a},a} + 1/\tilde{N}_{k_{n,b},b}}} \leq 2^{p/2} D_\mu + 2W_\mu \sqrt{\log(e + 2^{p-2})} + 2W_\epsilon \log(e+p),$$

*where $D_\mu > 0$ is a problem dependent constant and $(W_\mu, W_\epsilon)$ are the random variables defined in Lemma 7.*

*Proof.* Let $p \geq 1$. Let $n \in \mathbb{N}$ such that $S_n^p \neq \emptyset$, where $S_n^p$ as in Equation (27). Let $(k_{n,a})_{a \in [K]}$ be the phases indices for all arms. Since $N_{n,a} \geq 2^{p-1}$ for all $a \in S_n^p$, we have $k_{n,a} \geq p$ and $\tilde{N}_{k_{n,a},a} \geq 2^{p-2}$ by using Lemma 3. Likewise, $N_{n,a} < 2^{p-1}$ for all $a \notin S_n^p$, we have $k_{n,a} < p$ and $\tilde{N}_{k_{n,a},a} < 2^{p-2}$. Let $\overline{\Delta}_{\max} = \min_{a \neq b} |\mu_a - \mu_b|$, which satisfies $\overline{\Delta}_{\max} > 0$ by assumption on the instance considered. Using Lemma 7, for all $a \in S_n^p$ and $b \notin S_n^p$, we obtain

$$\frac{\tilde{\mu}_{k_{n,a},a} - \tilde{\mu}_{k_{n,b},b}}{\sqrt{1/\tilde{N}_{k_{n,a},a} + 1/\tilde{N}_{k_{n,b},b}}} \leq \sqrt{\tilde{N}_{k_{n,b},b}} (\tilde{\mu}_{k_{n,a},a} - \tilde{\mu}_{k_{n,b},b})$$

$$\leq \sqrt{\tilde{N}_{k_{n,b},b}} (\mu_a - \mu_b) + 2W_\mu \sqrt{\log(e + \tilde{N}_{k_{n,b},b})} + 2W_\epsilon \frac{\log(e + k_{n,b})}{\sqrt{\tilde{N}_{k_{n,b},b}}}$$

$$\leq 2^{(p-2)/2} \overline{\Delta}_{\max} + 2W_\mu \sqrt{\log(e + 2^{p-2})} + 2W_\epsilon \log(e + p)$$

where we used that $\tilde{N}_{k_{n,b},b} \geq 1$, $k_{n,b} < p$, $\tilde{N}_{k_{n,b},b} < 2^{p-2} \leq \tilde{N}_{k_{n,a},a}$ and $x \to \log(e + x)/x$ is decreasing. Taking $D_\mu = \overline{\Delta}_{\max}/2$ yields the result. $\square$

**Lemma 13 shows that the challenger is mildly undersampled if the leader is not mildly undersampled.**

**Lemma 13.** *Let $V_n^p$ be as in Equation (28). There exists $p_2$ with $\mathbb{E}_{\boldsymbol{\nu}}[\exp(\alpha p_2)] < +\infty$ for all $\alpha > 0$ such that if $p \geq p_2$, for all $n$ such that $U_n^p \neq \emptyset$, $B_n \notin V_n^p$ implies $C_n \in V_n^p$.*

*Proof.* Let $p_2$ to be specified later. Let $p \geq p_2$. Let $n \in \mathbb{N}$ such that $U_n^p \neq \emptyset$ and $V_n^p \neq [K]$, where $U_n^p \subseteq V_n^p$ are defined in Equation (28). In the following, we suppose that $B_n \notin V_n^p$.

Let $(k_{n,a})_{a \in [K]}$ be the phases indices for all arms. Let $p_0$ as in Lemma 10. Let $b_n^\star = \arg\max_{b \notin V_n^p} \mu_b$. Then, for all $p \geq 4p_0/3 - 1/3$ and all $n$ such that $B_n \notin V_n^p$, Lemma 10 yields that $B_n = b_n^\star = \arg\max_{a \notin V_n^p} \tilde{\mu}_{k_{n,a},a}$.

Let $p_1$ and $C_\mu$ as in Lemma 11, and $D_\mu$ as in Lemma 12. Then, for all $p \geq 4\max\{p_0, p_1\}/3 - 1/3$ and all $n$ such that $B_n \notin V_n^p$, we have $B_n = b_n^\star$ and

$$\forall b \notin V_n^p, \quad \frac{\tilde{\mu}_{k_{n,b_n^\star},b_n^\star} - \tilde{\mu}_{k_{n,b},b}}{\sqrt{1/\tilde{N}_{k_{n,b_n^\star},b_n^\star} + 1/\tilde{N}_{k_{n,b},b}}} \geq 2^{(3p+1)/8} C_\mu ,$$

$$\forall b \in U_n^p, \quad \frac{\tilde{\mu}_{k_{n,b_n^\star},b_n^\star} - \tilde{\mu}_{k_{n,b},b}}{\sqrt{1/\tilde{N}_{k_{n,b_n^\star},b_n^\star} + 1/\tilde{N}_{k_{n,b},b}}} \leq 2^{(p+1)/4} D_\mu + 2W_\mu \sqrt{\log(e + 2^{(p+1)/2 - 2})}$$

$$+ 2W_\epsilon \log(e + (p+1)/2) ,$$

where we used Lemmas 11 and 12. Let $p_3 = 16\lceil \log_2(2D_\mu/C_\mu) \rceil + 1$, then we have $2^{(p-1)/16} > \frac{D_\mu}{C_\mu}$ for all $p \geq p_3$. Let $p_4 = \frac{16}{9}\lceil \log_2 X_4 \rceil + 25$ and $p_5 = \frac{32}{9}\lceil \log_2 X_5 \rceil + 7$ where

$$X_4 = \sup\left\{ x > 1 \mid x \leq \frac{W_\mu^2}{C_\mu^2} \log(e + x^{8/9} 2^{25/18 - 3/4}) \right\} ,$$

$$X_5 = \sup\left\{ x > 1 \mid x \leq \frac{2W_\epsilon}{C_\mu} \log(e + 4 + 32\log_2(x)/18) \right\} .$$

As in the proof of Lemma 10, using Lemma 7 yields that $\mathbb{E}_{\boldsymbol{\nu}}[\exp(\alpha p_4)] < +\infty$ and $\mathbb{E}_{\boldsymbol{\nu}}[\exp(\alpha p_5)] < +\infty$ for all $\alpha \in \mathbb{R}_+$. Let $p_2 = \max\{p_3, p_4, p_5, 4\max\{p_0, p_1\}/3 - 1/3\} + 1$. Then, we have shown that for all $p \geq p_2$, for all $n$ such that $B_n \notin V_n^p$, we have $B_n = b_n^\star$ and

$$
\min_{b \notin V_n^p} \frac{\tilde{\mu}_{k_{n,b_n^\star},b_n^\star} - \tilde{\mu}_{k_{n,b},b}}{\sqrt{1/\tilde{N}_{k_{n,b_n^\star},b_n^\star} + 1/\tilde{N}_{k_{n,b},b}}} > \max_{b \in U_n^p} \frac{\tilde{\mu}_{k_{n,b_n^\star},b_n^\star} - \tilde{\mu}_{k_{n,b},b}}{\sqrt{1/\tilde{N}_{k_{n,b_n^\star},b_n^\star} + 1/\tilde{N}_{k_{n,b},b}}} \, ,
$$

Therefore, by definition of the TC challenger $C_n = \arg\min_{b \neq b_n^\star} \frac{\tilde{\mu}_{k_{n,b_n^\star},b_n^\star} - \tilde{\mu}_{k_{n,b},b}}{\sqrt{1/\tilde{N}_{k_{n,b_n^\star},b_n^\star} + 1/\tilde{N}_{k_{n,b},b}}}$, we obtain

that $C_n \in V_n^p$. Otherwise, there would be a contradiction given that we assumed that $U_n^p \neq \emptyset$. Given all the condition exhibited above, it is direct to see that $\mathbb{E}_{\boldsymbol{\nu}}[\exp(\alpha p_2)] < +\infty$ for all $\alpha > 0$. This concludes the proof. $\qquad\square$

**Lemma 14 shows that all the arms are sufficient explored for large enough $n$.**

**Lemma 14.** *There exists $N_0$ with $\mathbb{E}_{\boldsymbol{\nu}}[N_0] < +\infty$ such that for all $n \geq N_0$ and all $a \in [K]$,*

$$
N_{n,a} \geq \sqrt{n/K} \quad \text{and} \quad k_{n,a} \geq \frac{\log(n/K)}{2\log 2} + 1 \, .
$$

*Proof.* Let $p_0$ and $p_2$ as in Lemmas 10 and 13. Combining Lemmas 10 and 13 yields that, for all $p \geq p_3 = \max\{p_2, 4p_0/3 - 1/3\}$ and all $n$ such that $U_n^p \neq \emptyset$, we have $B_n \in V_n^p$ or $C_n \in V_n^p$. We have $\mathbb{E}_{\boldsymbol{\nu}}[2^{p_2}] < +\infty$. We have $2^{p-1} \geq K2^{3(p-1)/4}$ for all $p \geq p_4 = 4\lceil\log_2 K\rceil + 1$. Let $p \geq \max\{p_3, p_4\}$.

Suppose towards contradiction that $U_{K2^{p-1}}^p$ is not empty. Then, for any $1 \leq t \leq K2^{p-1}$, $U_t^p$ and $V_t^p$ are non empty as well. Using the pigeonhole principle, there exists some $a \in [K]$ such that $N_{2^{p-1},a} \geq 2^{3(p-1)/4}$. Thus, we have $\left|V_{2^{p-1}}^p\right| \leq K - 1$. Our goal is to show that $|V_{2^p}^p| \leq K - 2$. A sufficient condition is that one arm in $V_{2^{p-1}}^p$ is pulled at least $2^{3(p-1)/4}$ times between $2^{p-1}$ and $2^p - 1$.

**Case 1.** Suppose there exists $a \in V_{2^{p-1}}^p$ such that $L_{2^p,a} - L_{2^{p-1},a} \geq \frac{2^{3(p-1)/4}}{\beta} + 3/(2\beta)$. Using Lemma 6, we obtain

$$
N_{2^p,a}^a - N_{2^{p-1},a}^a \geq \beta(L_{2^p,a} - L_{2^{p-1},a}) - 3/2 \geq 2^{3(p-1)/4} \, ,
$$

hence $a$ is sampled $2^{3(p-1)/4}$ times between $2^{p-1}$ and $2^p - 1$.

**Case 2.** Suppose that for all $a \in V_{2^{p-1}}^p$, we have $L_{2^p,a} - L_{2^{p-1},a} < 2^{3(p-1)/4}/\beta + 3/(2\beta)$. Then,

$$
\sum_{a \notin V_{2^{p-1}}^p} (L_{2^p,a} - L_{2^{p-1},a}) \geq 2^{p-1} - K\left(2^{3(p-1)/4}/\beta + 3/(2\beta)\right)
$$

Using Lemma 6, we obtain

$$
\left| \sum_{a \notin V_{2^{p-1}}^p} (N_{2^p,a}^a - N_{2^{p-1},a}^a) - \beta \sum_{a \notin V_{2^{p-1}}^p} (L_{2^p,a} - L_{2^{p-1},a}) \right| \leq 3(K-1)/2 \, .
$$

Combining all the above, we obtain

$$
\sum_{a \notin V_{2^{p-1}}^p} (L_{2^p,a} - L_{2^{p-1},a}) - \sum_{a \notin V_{2^{p-1}}^p} (N_{2^p,a}^a - N_{2^{p-1},a}^a)
$$
$$
\geq (1-\beta) \sum_{a \notin V_{2^{p-1}}^p} (L_{2^p,a} - L_{2^{p-1},a}) - 3(K-1)/2
$$
$$
\geq (1-\beta)\left(2^{p-1} - K\left(2^{3(p-1)/4}/\beta + 3/(2\beta)\right)\right) - 3(K-1)/2 \geq K2^{3(p-1)/4} \, ,
$$

where the last inequality is obtained for $p \geq p_5$ with

$$
p_5 = \sup\left\{p \in \mathbb{N} \mid (1-\beta)\left(2^{p-1} - K\left(2^{3(p-1)/4}/\beta + 3/(2\beta)\right)\right) - 3(K-1)/2 < K2^{3(p-1)/4}\right\}.
$$

The LHS summation is exactly the number of times where an arm $a \notin V_{2^{p-1}}^p$ was leader but wasn't sampled, hence

$$\sum_{t=2^{p-1}}^{2^p-1} \mathbb{1}\left(B_t \notin V_{2^{p-1}}^p,\ I_t = C_t\right) \geq K2^{3(p-1)/4}$$

For any $2^{p-1} \leq t \leq 2^p - 1$, $U_t^p$ is non-empty, hence we have $B_t \notin V_{2^{p-1}}^p$ (hence $B_t \notin V_t^p$) implies $C_t \in V_t^p \subseteq V_{2^{p-1}}^p$. Therefore, we have shown that

$$\sum_{t=2^{p-1}}^{2^p-1} \mathbb{1}\left(I_t \in V_{2^{p-1}}^p\right) \geq \sum_{t=2^{p-1}}^{2^p-1} \mathbb{1}\left(B_t \notin V_{2^{p-1}}^p,\ I_t = C_t\right) \geq K2^{3(p-1)/4}\ .$$

Therefore, there is at least one arm in $V_{2^{p-1}}^p$ that is sampled $2^{3(p-1)/4}$ times between $2^{p-1}$ and $2^p - 1$.

In summary, we have shown $|V_{2^p}^p| \leq K - 2$ for all $p \geq p_6 = \max\{p_3, p_4, p_5\}$. By induction, for any $1 \leq k \leq K$, we have $\left|V_{k2^{p-1}}^p\right| \leq K - k$, and finally $U_{K2^{p-1}}^p = \emptyset$ for all $p \geq p_6$. Defining $N_0 = K2^{p_6-1}$, we have $\mathbb{E}_{\boldsymbol{\nu}}[N_0] < +\infty$ by using Lemmas 10 and 13 for $p_3 = \max\{p_2, 4p_0/3 - 1/3\}$ and $p_4$ and $p_5$ are deterministic. For all $n \geq N_0$, we let $2^{p-1} = \frac{n}{K}$. Then, by applying the above, we have $U_{K2^{p-1}}^p = U_n^{\log_2(n/K)+1}$ is empty, which shows that $N_{n,a} \geq \sqrt{n/K}$ for all $a \in [K]$. Using Lemma 3, we obtain that $k_{n,a} \geq \frac{\log(n/K)}{2\log 2} + 1$ for all $a \in [K]$. This concludes the proof. $\qquad\square$

### E.3 Convergence towards $\beta$-optimal allocation

The second step of in the generic analysis of Top Two algorithms [JDB$^+$22] is to show that AdaP-TT ensures convergence of its empirical proportions towards the $\beta$-optimal allocation. First, we show that the leader coincides with the best arm. Hence, the tracking procedure will ensure that the empirical proportion of time we sample it is exactly $\beta$. Second, we show that a sub-optimal arm whose empirical proportion overshoots its $\beta$-optimal allocation will not be sampled next as challenger. Therefore, this "overshoots implies not sampled" mechanism will ensure the convergence towards the $\beta$-optimal allocation. We emphasise that there are multiple ways to select the leader/challenger pair in order to ensure convergence towards the $\beta$-optimal allocation. Therefore, while we conduct the proof for AdaP-TT, other choices of leader/challenger pair would yield similar results. Note that our results heavily rely on having obtained sufficient exploration first.

**Convergence for the best arm.** Lemma 15 exhibits a random phase which ensures that the leader and the candidate answer are equal to the best arm for large enough $n$.

**Lemma 15.** *Let $N_0$ be as in Lemma 14. There exists $N_1 \geq N_0$ with $\mathbb{E}_{\boldsymbol{\nu}}[N_1] < +\infty$ such that, for all $n \geq N_1$, we have $\hat{a}_n = B_n = a^\star$.*

*Proof.* Let $k \geq 1$ and $(T_k(a))_{a \in [K]}$ as in Equation (10). Suppose that $\mathbb{E}_{\boldsymbol{\nu}}[\max_{a \in [K]} T_k(a)] < +\infty$. Then, Lemma 3 yields that $N_{T_k(a),a} = 2^{k-1}$ and $\tilde{N}_{k,a} = 2^{k-2}$. Using Lemma 7, we obtain that

$$\tilde{\mu}_{k,a^\star} \geq \mu_{a^\star} - W_\mu \sqrt{\frac{\log(e + 2^{k-2})}{2^{k-2}}} - W_\epsilon \frac{\log(e + k)}{2^{k-2}}\ ,$$

$$\forall a \neq a^\star,\quad \tilde{\mu}_{k,a} \leq \mu_a + W_\mu \sqrt{\frac{\log(e + 2^{k-2})}{2^{k-2}}} + W_\epsilon \frac{\log(e + k)}{2^{k-2}}\ .$$

Let $p_1 = \lceil \log_2(X_1 - e) \rceil + 2$ and $p_2 = \lceil \log_2(X_2 - e - 1) \rceil + 2$ with

$$X_1 = \sup\left\{x > 1 \mid x \leq 64\Delta_{\min}^{-2} W_\mu^2 \log x + e\right\} \leq h_1(64\Delta_{\min}^{-2} W_\mu^2,\ e)\ ,$$

$$X_2 = \sup\left\{x > 1 \mid x \leq 8\Delta_{\min}^{-1} W_\epsilon \log x + e + 1\right\} \leq h_1(8\Delta_{\min}^{-1} W_\epsilon,\ e + 1)\ ,$$

$$X_2 \geq \sup\left\{x > 1 \mid x \leq 8\Delta_{\min}^{-1} W_\epsilon \log(e + 2 + \log x)\right\}\ ,$$

where we used Lemma 9, and $h_1$ defined therein. Then, for all $k \in \mathbb{N}^K$ such that $\min_{a \in [K]} k_a > p_0 = \max\{p_1, p_2\}$ such that $\mathbb{E}_{\boldsymbol{\nu}}[\max_{a \in [K]} T_{k_a}(a)] < +\infty$, we have $\tilde{\mu}_{k,a^\star} \geq \mu_{a^\star} - \Delta_{\min}/4$ and $\tilde{\mu}_{k,a} \leq \mu_a + \Delta_{\min}/4$ for all $a \neq a^\star$, hence $a^\star = \arg\max_{a \in [K]} \tilde{\mu}_{k,a}$. We have, for all $\alpha \in \mathbb{R}_+$,

$$\exp(\alpha p_1) \leq e^{3\alpha}(X_1 - e)^{\alpha/\log 2} \quad \text{hence} \quad \mathbb{E}_{\boldsymbol{\nu}}[\exp(\alpha p_1)] < +\infty\ ,$$

where we used Lemma 7 and $h_1(x, e) \sim_{x \to +\infty} x \log x$ to obtain that $\exp(\alpha p_1)$ is at most polynomial in $W_\mu$. Likewise, we obtain that $\mathbb{E}_\nu[\exp(\alpha p_2)] < +\infty$ for all $\alpha \in \mathbb{R}_+$. Therefore, we have $\mathbb{E}_\nu[\exp(\alpha p_0)] < +\infty$ for all $\alpha \in \mathbb{R}_+$.

Let us define the UCB indices by $I_{k,a} = \tilde{\mu}_{k,a} + \sqrt{k/\tilde{N}_{k,a}} + k/(\epsilon \tilde{N}_{k,a})$. Using the above, we have

$$I_{k,a^\star} \geq \mu_{a^\star} - W_\mu \sqrt{\frac{\log(e + 2^{k-2})}{2^{k-2}}} - W_\epsilon \frac{\log(e + k)}{2^{k-2}} + \frac{k}{\epsilon 2^{k-2}},$$

$$\forall a \neq a^\star, \quad I_{k,a} \leq \mu_a + W_\mu \sqrt{\frac{\log(e + 2^{k-2})}{2^{k-2}}} + W_\epsilon \frac{\log(e + k)}{2^{k-2}} + \frac{k}{\epsilon 2^{k-2}}.$$

Therefore, we have $a^\star = \arg\max_{a \in [K]} I_{k,a}$ for all $k \in \mathbb{N}^K$ such that $\min_{a \in [K]} k_a > \max\{p_1, p_2\}$ such that $\mathbb{E}_\nu[\max_{a \in [K]} T_{k_a}(a)] < +\infty$.

Let $N_0$ as in Lemma 14. Using Lemma 14, we obtain that, for all $n \geq N_0$ and all $a \in [K]$, $k_{n,a} \geq \log_2(n/K)/2 + 1$. Therefore, we obtain $\min_{a \in [K]} k_{n,a} > \max\{p_1, p_2\}$ is implied by $n \geq N_1 = \max\{K4^{\max\{p_1, p_2\}}, N_0\}$. Using the above, we conclude that $\mathbb{E}_\nu[N_1] < +\infty$ and $\hat{a}_n = B_n = a^\star$ for all $n \geq N_1$. $\qquad\square$

**Lemma 16 shows that that the pulling proportion of the best arm converges towards $\beta$, provided the phase defined in Lemma 15 is reached in finite time for all arms.**

**Lemma 16.** *Let $\gamma > 0$, and $N_1$ be as in Lemma 15. There exists a deterministic constant $C_0 \geq 1$ such that, for all $n \geq C_0 N_1$,*

$$\left| \frac{N_{n,a^\star}}{n-1} - \beta \right| \leq \gamma.$$

*Proof.* Let $\gamma > 0$. Let $N_1$ as in Lemma 15. Let $M \geq N_1$. Using Lemma 15, we obtain $B_n = a^\star$ for all $n \geq M$. Therefore, we obtain $L_{n,a^\star} \geq n - M$ and $\sum_{a \neq a^\star} N_{n,a^\star}^a \leq M$ for all $n \geq M$. Using Lemma 6 yields that

$$\left| \frac{N_{n,a^\star}}{n-1} - \beta \right| \leq \frac{|N_{n,a^\star}^{a^\star} - \beta L_{n,a^\star}|}{n-1} + \beta \left| \frac{L_{n,a^\star}}{n-1} - 1 \right| + \frac{1}{n-1} \sum_{a \neq a^\star} N_{n,a^\star}^a$$

$$\leq \frac{1}{2(n-1)} + \beta \frac{2(M-1)}{n-1} \leq \gamma,$$

where the last inequality is obtained by taking $n \geq \max\{M, (1/2 + 2\beta(M-1))/\gamma + 1\}$. $\qquad\square$

**Convergence for the sub-optimal arms**    **Lemma 17 exhibits a random phase which ensures that if a sub-optimal arm overshoots its $\beta$-optimal allocation then it cannot be selected as challenger for large enough $n$.**

**Lemma 17.** *Let $\gamma > 0$. Let $N_1$ and $C_0$ be as in Lemma 15 and 16. There exists $N_2 \geq C_0 N_1$ with $\mathbb{E}_\nu[N_2] < +\infty$ such that, for all $n \geq N_2$,*

$$\exists a \neq a^\star, \quad \frac{N_{n,a}}{n-1} \geq \omega_{\beta,a}^\star + \gamma \quad \Longrightarrow \quad C_n \neq a.$$

*Proof.* Let $\gamma > 0$ and $\tilde{\gamma} > 0$. Let $N_1$ as in Lemma 15 and $C_0$ as in Lemma 16 for $\tilde{\gamma}$. Let $n \geq C_0 N_1$.

Let $a \neq a^\star$ such that $\frac{N_{n,a}}{n-1} \geq \omega_{\beta,a}^\star + \gamma$. Suppose towards contradiction that $\frac{N_{n,b}}{n-1} > \omega_{\beta,a}^\star$ for all $b \notin \{a^\star, a\}$. Then, for all $n \geq C_0 N_1$, we have

$$1 - \beta + \tilde{\gamma} \geq 1 - \frac{N_{n,a^\star}}{n-1} = \sum_{b \neq a^\star} \frac{N_{n,b}}{n-1} > \gamma + \sum_{b \neq a^\star} \omega_{\beta,b}^\star = 1 - \beta + \gamma,$$

which yields a contradiction for $\tilde{\gamma} \leq \gamma$. Therefore, for all $n \geq C_0 N_1$, we have

$$\exists a \neq a^\star, \quad \frac{N_{n,a}}{n-1} \geq \omega_{\beta,a}^\star + \gamma \quad \Longrightarrow \quad \exists b \notin \{a^\star, a\}, \quad \frac{N_{n,b}}{n-1} \leq \omega_{\beta,b}^\star.$$

Then, we have

$$\sqrt{\frac{1 + N_{n,a^\star}/N_{n,b}}{1 + N_{n,a^\star}/N_{n,a}}} \geq \sqrt{\frac{1 + (\beta - \tilde{\gamma})/\omega^\star_{\beta,b}}{1 + (\beta + \tilde{\gamma})/(\omega^\star_{\beta,a} + \gamma)}} \ .$$

In the following, we use Lemma 7 and similar manipulations as in the proof of Lemma 15. Therefore, we obtain that, for all $c \neq a^\star$,

$$\left| \tilde{\mu}_{k_{n,a^\star},a^\star} - \tilde{\mu}_{k_{n,c},c} - \Delta_c \right| \leq W_\mu \left( \sqrt{\frac{\log(e + 2^{k_{n,a^\star}-2})}{2^{k_{n,a^\star}-2}}} + \sqrt{\frac{\log(e + 2^{k_{n,c}-2})}{2^{k_{n,c}-2}}} \right)$$

$$+ W_\epsilon \left( \frac{\log(e + k_{n,a^\star})}{2^{k_{n,a^\star}-2}} + \frac{\log(e + k_{n,c})}{2^{k_{n,c}-2}} \right) \ .$$

Let $p_3 = \lceil \log_2(X_1 - e) \rceil + 2$ and $p_2 = \lceil \log_2(X_2 - e - 1) \rceil + 2$ with

$$X_3 = \sup \left\{ x > 1 \mid x \leq 16\eta^{-2} W_\mu^2 \log x + e \right\} \leq h_1(16\eta^{-2} W_\mu^2, \ e) \ ,$$
$$X_2 = \sup \left\{ x > 1 \mid x \leq 4\eta^{-1} W_\epsilon \log x + e + 1 \right\} \leq h_1(4\eta^{-1} W_\epsilon, \ e + 1) \ ,$$
$$X_2 \geq \sup \left\{ x > 1 \mid x \leq 4\eta^{-1} W_\epsilon \log(e + 2 + \log x) \right\} \ ,$$

where we used Lemma 9, and $h_1$ defined therein. We have, for all $\alpha \in \mathbb{R}_+$,

$$\exp(\alpha p_3) \leq e^{3\alpha}(X_3 - e)^{\alpha/\log 2} \quad \text{hence} \quad \mathbb{E}_{\boldsymbol{\nu}}[\exp(\alpha p_3)] < +\infty \ ,$$

where we used Lemma 7 and $h_1(x, e) \sim_{x \to +\infty} x \log x$ to obtain that $\exp(\alpha p_3)$ is at most polynomial in $W_\mu$. Likewise, we obtain that $\mathbb{E}_{\boldsymbol{\nu}}[\exp(\alpha p_2)] < +\infty$ for all $\alpha \in \mathbb{R}_+$.

Using Lemma 14 (with $C_0 N_1 \geq N_1 \geq N_0$), we obtain that, for all $n \geq C_0 N_1$ and all $a \in [K]$, $k_{n,a} \geq \log_2(n/K)/2 + 1$. Therefore, we obtain $\min_{a \in [K]} k_{n,a} > \max\{p_2, p_3\}$ is implied by $n \geq N_2 = \max\{K 4^{\max\{p_3, p_2\}}, C_0 N_1\}$. Using the above, we conclude that $\mathbb{E}_{\boldsymbol{\nu}}[N_2] < +\infty$ and $\max_{c \neq a^\star} |\tilde{\mu}_{k_{n,a^\star},a^\star} - \tilde{\mu}_{k_{n,c},c} - \Delta_c| \leq \eta$ for all $n \geq N_2$.

Then, for all $n \geq N_2$, we have $B_n = a^\star$ and

$$\frac{\tilde{\mu}_{k_{n,a^\star},a^\star} - \tilde{\mu}_{k_{n,a},a}}{\tilde{\mu}_{k_{n,a^\star},a^\star} - \tilde{\mu}_{k_{n,b},b}} \sqrt{\frac{1 + N_{n,a^\star}/N_{n,b}}{1 + N_{n,a^\star}/N_{n,a}}} \geq \frac{\Delta_a - \eta}{\Delta_b + \eta} \sqrt{\frac{1 + (\beta - \tilde{\gamma})/\omega^\star_{\beta,b}}{1 + (\beta + \tilde{\gamma})/(\omega^\star_{\beta,a} + \gamma)}} > 1 \ ,$$

where the last inequality is obtained by taking $\eta$ and $\tilde{\gamma}$ sufficiently small and by using Equation (25), i.e.

$$\frac{\Delta_a}{\Delta_b} \sqrt{\frac{1 + \beta/\omega^\star_{\beta,b}}{1 + \beta/\omega^\star_{\beta,a}}} = 1 \ .$$

Therefore, we have shown that $B_n = a^\star$ and

$$\frac{\tilde{\mu}_{k_{n,a^\star},a^\star} - \tilde{\mu}_{k_{n,a},a}}{\sqrt{1/N_{n,a^\star} + 1/N_{n,a}}} > \frac{\tilde{\mu}_{k_{n,a^\star},a^\star} - \tilde{\mu}_{k_{n,b},b}}{\sqrt{1/N_{n,a^\star} + 1/N_{n,b}}} \quad \text{hence} \quad C_n \neq a \ .$$

This concludes the proof. $\qquad \square$

**Lemma 18 shows that that the pulling proportion of the best arm converges towards $\beta$ for large enough $n$.**

**Lemma 18.** *Let $\gamma > 0$ and $T_{\boldsymbol{\mu},\gamma}$ be as in Equation (26). Then, we have $\mathbb{E}_{\boldsymbol{\nu}}[T_{\boldsymbol{\mu},\gamma}] < +\infty$.*

*Proof.* Let $\gamma > 0$ and $\tilde{\gamma} > 0$. Let $N_2$ as in Lemma 17 for $\tilde{\gamma}$. Let $M \geq N_2$. Using Lemmas 15, 16 and 17 for all $n \geq M$, we obtain that $B_n = a^\star$, $\left| \frac{N_{n,a^\star}}{n-1} - \beta \right| \leq \tilde{\gamma}$ and

$$\exists a \neq a^\star, \quad \frac{N_{n,a}}{n-1} \geq \omega^\star_{\beta,a} + \tilde{\gamma} \quad \implies \quad C_n \neq a \ .$$

For all $a \neq a^\star$, let us define $t_{n,a}(\tilde{\gamma}) = \max \left\{ t \mid M \leq t \leq n, \ N_{t,a}/(n-1) < \omega^\star_{\beta,a} + \tilde{\gamma} \right\}$. Since $N_{t,a}/(n-1) \leq N_{t,a}/(t-1)$ for $t \leq n$, we have

$$\frac{N_{n,a}}{n-1} \leq \frac{M-1}{n-1} + \frac{1}{n-1} \sum_{t=M}^{n} \mathbb{1}\left( I_t = C_t = a \right)$$

$$\leq \frac{M-1}{n-1} + \frac{1}{n-1}\sum_{t=M}^{n}\mathbb{1}\left(\frac{N_{t,a}}{n-1} < \omega_{\beta,a}^{\star} + \tilde{\gamma}, \; I_t = C_t = a\right)$$

$$\leq \frac{M-1}{n-1} + \frac{N_{t_{n,a}(\tilde{\gamma}),a}}{n-1} < \frac{M-1}{n-1} + \omega_{\beta,a}^{\star} + \tilde{\gamma}\,.$$

The second inequality uses Lemma 17, and the two last inequalities use the definition of $t_{n,a}(\tilde{\gamma})$. Using that $\sum_{a\in[K]}\frac{N_{n,a}}{n-1} = \sum_{a\in[K]}\omega_{\beta,a}^{\star} = 1$, we obtain

$$\frac{N_{n,a}}{n-1} = 1 - \sum_{b\neq a}\frac{N_{n,a}}{n-1} \geq 1 - \sum_{b\neq a}\left(\omega_{\beta,b}^{\star} + \tilde{\gamma} + \frac{M-1}{n-1}\right) = \omega_{\beta,a}^{\star} - (K-1)\left(\tilde{\gamma} + \frac{M-1}{n-1}\right)\,.$$

Taking $\tilde{\gamma} \leq \gamma/(2(K-1))$ and $n \geq \max\{M, 2(K-1)(M-1)/\gamma + 1\}$ yields that

$$\left\|\frac{N_n}{n-1} - \omega_{\beta}^{\star}\right\|_{\infty} \leq \gamma\,.$$

Let $T_{\boldsymbol{\mu},\gamma}$ as in 26. Then, we showed that $T_{\boldsymbol{\mu},\gamma} \leq \max\{M, 2(K-1)(M-1)/\gamma+1\}$. Therefore, we have

$$\mathbb{E}_{\boldsymbol{\nu}}[T_{\boldsymbol{\mu},\gamma}] \leq \mathbb{E}_{\boldsymbol{\nu}}[\max\{M, 2(K-1)(M-1)/\gamma+1\}] < +\infty\,,$$

which concludes the proof. $\qquad\square$

### E.4 Cost of doubling and forgetting

Compared to the generic analysis of Top Two algorithms [JDB$^+$22], for AdaP-TT, we need to control the sample complexity cost of doubling and forgetting. Due to this reason, we have to pay a multiplicative four-factor: one two-factor due to doubling, and another two-factor due to forgetting. It is possible to show that this cost exists when adapting any "reasonable" BAI algorithm, meaning for any BAI algorithm in which the empirical proportions are converging towards an allocation $\omega$ such that $\min_a \omega_a > 0$. Those BAI algorithms are "reasonable" because the asymptotic lower bound stipulates that all arms have to be sampled linearly in order to be near optimal.

Let $\omega \in \Sigma_K$ be any allocation over arms such that $\min_a \omega_a > 0$. Let $\gamma > 0$. We denote by $T_{\boldsymbol{\mu},\gamma}(\omega)$ the *convergence time* towards $\omega$, which is a random variable quantifying the number of samples required for the global empirical allocations $N_n/(n-1)$ to be $\gamma$-close to $\omega$ for any subsequent time, namely

$$T_{\boldsymbol{\mu},\gamma}(\omega) \triangleq \inf\left\{T \geq 1 \mid \forall n \geq T, \; \left\|\frac{N_n}{n-1} - \omega\right\|_{\infty} \leq \gamma\right\}\,. \tag{29}$$

**Lemma 19 shows that the phase switches of the arms happen in a round-robin fashion, which means that an arm switches phase for a second time after all other arms first switch their own phases.**

**Lemma 19.** *Let $\omega \in \Sigma_K$ such that $\min_a \omega_a > 0$. Assume that there exists $\gamma_{\boldsymbol{\mu}} > 0$ such that for $\mathbb{E}_{\boldsymbol{\nu}}[T_{\boldsymbol{\mu},\gamma}(\omega)] < +\infty$ for all $\gamma \in (0,\gamma_{\boldsymbol{\mu}})$, where $T_{\boldsymbol{\mu},\gamma}(\omega)$ is defined in Equation (29). Let $\eta > 0$. There exists $\tilde{\gamma}_{\boldsymbol{\mu}} \in (0,\gamma_{\boldsymbol{\mu}})$ such that, for all $\gamma \in (0,\tilde{\gamma}_{\boldsymbol{\mu}})$, there exists $N_3 \geq T_{\boldsymbol{\mu},\gamma}(\omega)$ with $\mathbb{E}_{\boldsymbol{\nu}}[N_3] < +\infty$ which satisfies*

$$\forall n \geq N_3, \quad \frac{\max_{a\in[K]}T_{k_{n,a}}(a)-1}{\min_{a\in[K]}T_{k_{n,a}}(a)-1} \leq 2 + \eta\,.$$

*Proof.* Let $\eta > 0$. Let $\tilde{\gamma}_{\boldsymbol{\mu}} \in (0,\gamma_{\boldsymbol{\mu}})$ such that $2\max_{a\in[K]}(\omega_a + \gamma)/(\omega_a - \gamma) \leq 2 + \eta$, which is possible since $\min_a \omega_a > 0$. Let $\gamma \in (0,\tilde{\gamma}_{\boldsymbol{\mu}})$. By assumption, we have $\mathbb{E}_{\boldsymbol{\nu}}[T_{\boldsymbol{\mu},\gamma}(\omega)] < +\infty$. Then, for all $n \geq T_{\boldsymbol{\mu},\gamma}(\omega)$,

$$\left\|\frac{N_n}{n-1} - \omega\right\|_{\infty} \leq \gamma\,.$$

Let $M \geq T_{\boldsymbol{\mu},\gamma}(\omega)$. Let use denote by $k_M = (k_{M,a})_{a\in[K]}$ the current phases for all arms $a \in [K]$ at time $M$. Then, for all $n \geq M$ and all $a \in [K]$, we have $N_{n,a} \geq (n-1)(\omega_a - \gamma)$. Therefore, taking

$n \geq \max_{a \in [K]} 2^{k_{M,a}}(\omega_a - \gamma)^{-1} + 1$, we obtain that $N_{n,a} \geq 2^{k_{M,a}}$ for all $a \in [K]$, hence we have $\max_{a \in [K]} T_{k_{M,a}+1}(a) \leq n$. Since $\min_{a \in [K]} T_{k_{M,a}+1}(a) \geq M$, we have

$$\max_{a \in [K]} \left| \frac{N_{T_{k_{M,a}+1}(a),a}}{n - 1} - \omega_a \right| \leq \gamma \,.$$

Likewise, taking $n \geq \max_{a \in [K]} 2^{k_{M,a}+1}(\omega_a - \gamma)^{-1} + 1$, we obtain that $N_{n,a} \geq 2^{k_{M,a}+1}$ for all $a \in [K]$, hence we have $\max_{a \in [K]} T_{k_{M,a}+2}(a) \leq n$. Let $a_1 = \arg\min_{a \in [K]} T_{k_{M,a}+2}(a)$. By definition and using Lemma 3, we have

$$2^{k_{M,a_1}+1} = N_{T_{k_{M,a_1}+2}(a_1),a_1} \leq (T_{k_{M,a_1}+2}(a_1) - 1)(\omega_{a_1} + \gamma) \,,$$

$$\forall a \neq a_1, \quad 2^{k_{M,a}} \leq N_{T_{k_{M,a_1}+2}(a_1),a} \leq (T_{k_{M,a_1}+2}(a_1) - 1)(\omega_a + \gamma) \,.$$

Let $a_2 = \arg\max_{a \in [K]} T_{k_{M,a}+2}(a)$. By definition and using Lemma 3, we have

$$2^{k_{M,a_2}+1} = N_{T_{k_{M,a_2}+2}(a_2),a_2} \geq (T_{k_{M,a_2}+2}(a_2) - 1)(\omega_{a_2} - \gamma) \,,$$

Therefore, combining the above yields

$$(T_{k_{M,a_2}+2}(a_2) - 1) \leq (T_{k_{M,a_1}+2}(a_1) - 1)2\frac{\omega_{a_2} + \gamma}{\omega_{a_2} - \gamma} \leq (T_{k_{M,a_2}+2}(a_2) - 1)(2 + \eta) \,,$$

where the last inequality uses that $\gamma \in (0, \tilde{\gamma}_{\boldsymbol{\mu}})$ and $\tilde{\gamma}_{\boldsymbol{\mu}} \in (0, \gamma_{\boldsymbol{\mu}})$ is such that $2\max_{a \in [K]}(\omega_a + \gamma)/(\omega_a - \gamma) \leq 2 + \eta$. We take $n \geq N_3 = \max_{a \in [K]} T_{k_{M,a}+2}(a)$, hence we have $k_{n,a} \geq k_{M,a} + 2$ for all $a \in [K]$. Since $\mathbb{E}_{\boldsymbol{\nu}}[T_{\boldsymbol{\mu},\gamma}(\omega)] < +\infty$ (i.e. arms are sampled linearly), it is direct to see that $\mathbb{E}_{\boldsymbol{\nu}}[\max_{a \in [K]} T_{k_{M,a}+2}(a)] < +\infty$. This concludes the proof. $\qquad\square$

Note that $T_{\boldsymbol{\mu},\gamma}$ defined in 26 is such that $T_{\boldsymbol{\mu},\gamma} = T_{\boldsymbol{\mu},\gamma}(\omega^{\star}_{\mathrm{KL},\beta})$ where $T_{\boldsymbol{\mu},\gamma}(\omega)$ as in Equation (29). Lemma 18 showed that $\mathbb{E}_{\boldsymbol{\nu}}[T_{\boldsymbol{\mu},\gamma}] < +\infty$ for all $\gamma > 0$. Therefore, the condition of Lemma 19 are fulfilled by AdaP-TT.

### E.5 Asymptotic expected sample complexity

The final step of the generic analysis of Top Two algorithms [JDB+22] is to invert the private GLR stopping rule by leveraging the convergence of the empirical proportions towards the $\beta$-optimal allocation. Compared to the non-private GLR stopping rule, the private threshold in the private GLR stopping rule involves an additive term in $\mathcal{O}(\log(1/\delta)^2)$. This difference is the largest price that we pay to obtain a private BAI algorithm. We defer the reader to Appendix E.7 for a more detailed discussion on it.

**Asymptotically $\beta$-optimal $\epsilon$-DP-FC-BAI algorithm.** The inversion of the GLR stopping rule by leveraging the convergence of the empirical proportions towards the ($\beta$-)optimal allocation is a generic method used in the BAI literature. Provided this convergence is shown, it only depends on the threshold that ensures $\delta$-correctness. More precisely, it only depends on its asymptotic dependence in $\log(1/\delta)$. In addition to the multiplicative four-factor, the price of privacy for asymptotically $\beta$-optimal BAI algorithms when combined with the non-private GLR stopping rule is a problem dependent multiplicative factor $1 + \sqrt{1 + \Delta^2_{\max}/(2\epsilon^2)}$ (Lemma 20).

**Lemma 20.** *Let $(\delta, \beta) \in (0,1)^2$. Assume that there exists $\gamma_{\boldsymbol{\mu}} > 0$ such that for $\mathbb{E}_{\boldsymbol{\nu}}[T_{\boldsymbol{\mu},\gamma}] < +\infty$ for all $\gamma \in (0, \gamma_{\boldsymbol{\mu}})$, where $T_{\boldsymbol{\mu},\gamma}$ is defined in Equation* (26). *Combining the private GLR stopping rule (Equation* (16)) *with private threshold (Equation* (4)) *yields a $\delta$-correct algorithm which satisfies that, for all $\boldsymbol{\nu}$ with mean $\boldsymbol{\mu}$ such that $|a^{\star}(\boldsymbol{\mu})| = 1$,*

$$\limsup_{\delta \to 0} \frac{\mathbb{E}_{\boldsymbol{\nu}}[\tau_\delta]}{\log(1/\delta)} \leq 4T^{\star}_{\mathrm{KL},\beta}(\boldsymbol{\mu}) \left( 1 + \sqrt{1 + \frac{\Delta^2_{\max}}{2\epsilon^2}} \right) \,.$$

*Proof.* Theorem 4 yields the $\delta$-correctness.

Let $a^\star$ be the unique best arm, i.e. $a^\star(\boldsymbol{\mu}) = \{a^\star\}$. Let $\zeta > 0$. Using Equation (25) and the continuity of

$$(\boldsymbol{\mu}, w) \mapsto \min_{a \neq a^\star(\boldsymbol{\mu})} \frac{(\mu_{a^\star(\boldsymbol{\mu})} - \mu_a)^2}{2(1/w_{a^\star(\boldsymbol{\mu})} + 1/w_a)}$$

yields that there exists $\gamma_\zeta > 0$ such that $\left\| \frac{N_n}{n-1} - \omega_\beta^\star \right\|_\infty \leq \gamma_\zeta$ and $\max_{a \in [K]} |\tilde{\mu}_{k_{n,a}+1,a} - \mu_a| \leq \gamma_\zeta$ implies that

$$\forall a \neq a^\star, \quad \frac{(\tilde{\mu}_{k_{n,a^\star}+1,a^\star} - \tilde{\mu}_{k_{n,a}+1,a})^2}{(n-1)/N_{n,a^\star} + (n-1)/N_{n,a}} \geq \frac{2(1-\zeta)}{T_{\mathrm{KL},\beta}^\star(\boldsymbol{\mu})}$$

$$\frac{n-1}{N_{n,a^\star}} + \frac{n-1}{N_{n,a}} \leq \frac{\Delta_a^2}{2}(1+\zeta)T_{\mathrm{KL},\beta}^\star(\boldsymbol{\mu}) \,.$$

We choose such a $\gamma_\zeta$. Let $\gamma_{\boldsymbol{\mu}} > 0$ be such that for $\mathbb{E}_\nu[T_{\boldsymbol{\mu},\gamma}] < +\infty$ for all $\gamma \in (0, \gamma_{\boldsymbol{\mu}})$, where $T_{\boldsymbol{\mu},\gamma}$ is defined in Equation (26). Let $\eta > 0$. Let $\tilde{\gamma}_{\boldsymbol{\mu}} \in (0, \gamma_{\boldsymbol{\mu}})$ as in Lemma 19 for this $\eta$. In the following, let us consider $\gamma \in (0, \min\{\tilde{\gamma}_{\boldsymbol{\mu}}, \gamma_\zeta, \beta/4, \Delta_{\min}/4\})$.

Let $N_3 \geq T_{\boldsymbol{\mu},\gamma}$ with $\mathbb{E}_\nu[N_3] < +\infty$ as Lemma 19 for those $(\gamma, \eta)$. Then, we have $\mathbb{E}_\nu[T_{\boldsymbol{\mu},\gamma}] < +\infty$ and

$$\forall n \geq N_3, \quad \frac{\max_{a \in [K]} T_{k_{n,a}}(a) - 1}{\min_{a \in [K]} T_{k_{n,a}}(a) - 1} \leq 2 + \eta \,.$$

Since arms are sampled linearly, it is direct to construct $N_4 \geq N_3$ with $\mathbb{E}_\nu[N_4] < +\infty$ such that, for all $n \geq N_4$, we have $\max_{a \in [K]} \max_{k \in \{k_{n,a}, k_{n,a}+1\}} |\tilde{\mu}_{k,a} - \mu_a| \leq \gamma$, Therefore, we have $\hat{a}_n = a^\star$.

Let $\kappa \in (0,1)$. Let $n \geq N_4/\kappa$ and $(k_{n,a})_{a \in [K]}$ be the current phases at time $n$. Combining the above, we have $\hat{a}_n = a^\star$ and

$$\max_{a \in [K]} |\tilde{\mu}_{k_{n,a}+1,a} - \mu_a| \leq \gamma \quad, \quad \left\| \frac{N_n}{n-1} - \omega_\beta^\star \right\|_\infty \leq \gamma \quad \text{and} \quad \frac{\max_{a \in [K]} T_{k_{n,a}}(a) - 1}{\min_{a \in [K]} T_{k_{n,a}}(a) - 1} \leq 2 + \eta \,.$$

Let $a_1 = \arg\min_{a \in [K]} T_{k_{n,a}}(a)$ and $a_2 = \arg\max_{a \in [K]} T_{k_{n,a}}(a)$. Therefore, we obtain, for all $a \neq a^\star$,

$$\frac{(\tilde{\mu}_{k_{n,\hat{a}_n}+1,\hat{a}_n} - \tilde{\mu}_{k_{n,a}+1,a})^2}{1/\tilde{N}_{k_{n,\hat{a}_n}+1,\hat{a}_n} + 1/\tilde{N}_{k_{n,a}+1,a}} = \frac{(\tilde{\mu}_{k_{n,a^\star}+1,a^\star} - \tilde{\mu}_{k_{n,a}+1,a})^2}{1/N_{T_{k_{n,a^\star}}(a^\star),a^\star} + 1/N_{T_{k_{n,a}}(a),a}}$$

$$\geq \frac{(\tilde{\mu}_{k_{n,a^\star}+1,a^\star} - \tilde{\mu}_{k_{n,a}+1,a})^2}{1/N_{T_{k_{n,a_1}}(a_1),a^\star} + 1/N_{T_{k_{n,a_1}}(a_1),a}}$$

$$\geq \big( \min_{a \in [K]} T_{k_{n,a}}(a) - 1 \big) \frac{2(1-\zeta)}{T_{\mathrm{KL},\beta}^\star(\boldsymbol{\mu})} \,.$$

Similarly, we can show that, for all $a \neq a^\star$,

$$\frac{1}{\tilde{N}_{k_{n,a^\star}+1,a^\star}} + \frac{1}{\tilde{N}_{k_{n,a}+1,a}} = \frac{1}{N_{T_{k_{n,a^\star}}(a^\star),a^\star}} + \frac{1}{N_{T_{k_{n,a}}(a),a}}$$

$$\leq \frac{1}{N_{T_{k_{n,a_1}}(a_1),a^\star}} + \frac{1}{N_{T_{k_{n,a_1}}(a_1),a}}$$

$$\leq \frac{1}{\min_{a \in [K]} T_{k_{n,a}}(a) - 1} \frac{\Delta_a^2}{2}(1+\zeta)T_{\mathrm{KL},\beta}^\star(\boldsymbol{\mu})$$

$$\leq \frac{1}{\min_{a \in [K]} T_{k_{n,a}}(a) - 1} \frac{\Delta_{\max}^2}{2}(1+\zeta)T_{\mathrm{KL},\beta}^\star(\boldsymbol{\mu}) \,.$$

Let $(c_k)_{k \in \mathbb{N}}$ as in Equation (14). Using Lemma 3, we obtain, for all $a \neq a^\star$,

$$2c_{(k_{n,a^\star}+1)(k_{n,a}+1)}(\tilde{N}_{k_{n,a^\star}+1,a^\star}, \tilde{N}_{k_{n,a}+1,a}, \delta/2) \leq 8\log(4 + (\max_{b \in [K]} k_{n,b} - 1)\log 2)$$

$$+ 4\mathcal{C}_G \left( \log(1/\delta)/2 + s\log(\max_{b \in [K]} k_{n,b} - 1) + \log(2(K-1)\zeta(s)^2)/2 \right)$$

Likewise, we obtain, for all $a \in [K]$,

$$\frac{1}{\tilde{N}_{k_{n,a^\star}+1,a^\star}\epsilon^2} \log\left(\frac{2K(k_{n,a^\star}+1)^s\zeta(s)}{\delta}\right)^2 + \frac{1}{\tilde{N}_{k_{n,a}+1,a}\epsilon^2}\log\left(\frac{2K(k_{n,a}+1)^s\zeta(s)}{\delta}\right)^2$$

$$\leq \frac{\Delta_{\max}^2}{2\epsilon^2}\frac{(1+\zeta)T_{\mathrm{KL},\beta}^\star(\boldsymbol{\mu})}{\min_{b\in[K]}T_{k_{n,b}}(b)-1}\left(\log(1/\delta) + s\log(\max_{b\in[K]}k_{n,b}+1) + \log(2K\zeta(s))\right)^2$$

Let us denote by $T_{k_n+1}^+ = \max_{b\in[K]}T_{k_{n,b}+1}(b)$, $T_{k_n+2}^+ = \max_{b\in[K]}T_{k_{n,b}+2}(b)$, $T_{k_n+1}^- = \min_{b\in[K]}T_{k_{n,b}+1}(b)$, $T_{k_n}^- = \min_{b\in[K]}T_{k_{n,b}}(b)$. Let $T$ be a time such that $T \geq T_{k_n+1}^+ \geq \kappa T$. Using Lemmas 3 and 19, we have

$$(k_{n,b}-1)\log 2 = \log N_{T_{k_{n,b}}(b),b} \leq \log T_{k_{n,b}}(b) \leq \log T_{k_n}^+ \leq \log T_{k_n}^- + \log(2+\eta)\,.$$

Using the private GLR stopping rule (Equation (16)), we have

$$\min\{\tau_\delta, T\} - \kappa T \leq \sum_{T\geq T_{k_n}^+\geq \kappa T}(T_{k_n+2}^+ - T_{k_n+1}^+)\mathbb{1}\left(\tau_\delta > T_{k_n+1}^+\right)$$

$$\leq \sum_{T\geq T_{k_n}^+\geq \kappa T}(T_{k_n+2}^+ - T_{k_n+1}^+)\mathbb{1}\left(\exists a \neq a^\star, \frac{(\tilde{\mu}_{k_{n,a^\star}+1,a^\star} - \tilde{\mu}_{k_{n,a}+1,a})^2}{1/\tilde{N}_{k_{n,a^\star}+1,a^\star} + 1/\tilde{N}_{k_{n,a}+1,a}}\right.$$

$$\left. < 2c_{\epsilon,k_{n,a^\star}+1,k_{n,a}+1}(\tilde{N}_{k_{n,a^\star}+1,a^\star}, \tilde{N}_{k_{n,a}+1,a}, \delta)\right)$$

$$\leq \sum_{T\geq T_{k_n}^+\geq \kappa T}(T_{k_n+2}^+ - T_{k_n+1}^+)\mathbb{1}\left((T_{k_n}^- - 1)\frac{1-\zeta}{T_{\mathrm{KL},\beta}^\star(\boldsymbol{\mu})} < 8\log(4 + \log T_{k_n}^- + \log(2+\eta))\right.$$

$$+ 4\mathcal{C}_G\left(\log(1/\delta)/2 + s\log(2 + \log_2 T_{k_n}^- + \log_2(2+\eta)) + \log(2(K-1)\zeta(s)^2)/2\right)$$

$$\left. + \frac{\Delta_{\max}^2}{2\epsilon^2}\frac{(1+\zeta)T_{\mathrm{KL},\beta}^\star(\boldsymbol{\mu})}{T_{k_n}^- - 1}\left(\log(1/\delta) + s\log(2 + \log_2 T_{k_n}^- + \log_2(2+\eta)) + \log(2K\zeta(s))\right)^2\right),$$

Let $T_\zeta(\delta)$ defined as the largest deterministic time such that the above condition is satisfied when replacing $T_{k_n}^-$ by $(1-\kappa)T$. Let $k_\delta$ be the largest random vector of phases such that that $T_{k_\delta+1}^+ \leq T_\zeta(\delta)$ almost surely, hence $T_{k_\delta+2}^+ > T_\zeta(\delta)$ almost surely. Then, using the above yields that $\tau_\delta \leq T_{k_\delta+2}^+$ almost surely, hence

$$\limsup_{\delta\to 0}\frac{\mathbb{E}_{\boldsymbol{\nu}}[\tau_\delta]}{\log(1/\delta)} \leq \limsup_{\delta\to 0}\frac{\mathbb{E}_{\boldsymbol{\nu}}[T_{k_\delta+2}^+]}{\log(1/\delta)} \leq (2+\eta)^2\limsup_{\delta\to 0}\frac{\mathbb{E}_{\boldsymbol{\nu}}[T_{k_\delta+1}^+]}{\log(1/\delta)} \leq (2+\eta)^2\limsup_{\delta\to 0}\frac{T_\zeta(\delta)}{\log(1/\delta)},$$

where the second inequality uses Lemma 19 twice, i.e. $T_{k_\delta+2}^+ \leq (2+\eta)T_{k_\delta+2}^- \leq (2+\eta)^2 T_{k_\delta+1}^+$, and the last one used the definition of $k_\delta$ and that $T_\zeta(\delta)$ is deterministic.

Since we are only interested in upper bounding $\limsup_{\delta\to 0}\frac{T_\zeta(\delta)}{\log(1/\delta)}$, we can safely drop the second orders terms in $T$ and $\log(1/\delta)$. This allows us to remove the terms in $\mathcal{O}(\log\log T)$ and in $\mathcal{O}(\log\log(1/\delta))$. Using that $\mathcal{C}_G(x) = x + \mathcal{O}(\log x)$, tedious manipulations yields that

$$\limsup_{\delta\to 0}\frac{T_\zeta(\delta)}{\log(1/\delta)} \leq \frac{T_{\mathrm{KL},\beta}^\star(\boldsymbol{\mu})}{1-\kappa}D_\zeta(\mu,\epsilon)\,,$$

where

$$D_\zeta(\mu,\epsilon) = \sup\left\{x \mid x^2 < \frac{2}{1-\zeta}x + \frac{1+\zeta}{1-\zeta}\frac{\Delta_{\max}^2}{2\epsilon^2}\right\} \leq \frac{1}{1-\zeta}\left(1 + \sqrt{1 + (1-\zeta^2)\frac{\Delta_{\max}^2}{2\epsilon^2}}\right)\,.$$

The last inequality uses that $x^2 - 2bx - c < 0$ for all $x \in [0, b(1 + \sqrt{1 + c/b^2}))$. Therefore, we have shown that

$$\limsup_{\delta\to 0}\frac{\mathbb{E}_{\boldsymbol{\nu}}[\tau_\delta]}{\log(1/\delta)} \leq (2+\eta)^2\frac{T_{\mathrm{KL},\beta}^\star(\boldsymbol{\mu})}{(1-\kappa)(1-\zeta)}\left(1 + \sqrt{1 + (1-\zeta^2)\frac{\Delta_{\max}^2}{2\epsilon^2}}\right)\,.$$

Letting $\kappa$, $\eta$ and $\zeta$ goes to zero yields that

$$\limsup_{\delta \to 0} \frac{\mathbb{E}_{\boldsymbol{\nu}}[\tau_\delta]}{\log(1/\delta)} \leq 4T^\star_{\mathrm{KL},\beta}(\boldsymbol{\mu}) \left( 1 + \sqrt{1 + \frac{\Delta^2_{\max}}{2\epsilon^2}} \right) \ .$$

$\square$

**Concluding the proof of Theorem 5.**  Combining Lemmas 14, 18, 19 and 20 concludes the proof of Theorem 5. We restrict the result to instances such that $\min_{a \neq b} |\mu_a - \mu_b| > 0$ in order for Lemma 14 to hold. Note that this is an artifact of the asymptotic proof which could be alleviated with more careful considerations.

**Asymptotically optimal $\epsilon$-DP-FC-BAI algorithm.**  While Lemma 20 is derived for BAI algorithms converging towards the $\beta$-optimal allocation $\omega^\star_{\mathrm{KL},\beta}(\boldsymbol{\mu})$, it is direct to see that a similar inversion results can be obtained for BAI algorithms that converge towards the unique optimal allocation $\omega^\star_{\mathrm{KL}}(\boldsymbol{\mu}) = \{w^\star\}$ defined as

$$\omega^\star_{\mathrm{KL}}(\boldsymbol{\mu}) \triangleq \arg\max_{\omega \in \Sigma_K} \min_{a \neq a^\star} \frac{\Delta^2_a}{1/\omega_{a^\star} + 1/\omega_a} \ . \tag{30}$$

At equilibrium, we have equality of the transportation costs (see [JD22] for example), namely

$$\forall a \neq a^\star, \quad \frac{\Delta^2_a}{1/\omega^\star_{a^\star} + 1/\omega^\star_a} = 2T^\star_{\mathrm{KL}}(\boldsymbol{\mu})^{-1} \ . \tag{31}$$

In addition to the multiplicative four-factor, the price of privacy for asymptotically optimal BAI algorithms when combined with the non-private GLR stopping rule is a problem dependent multiplicative factor $1 + \sqrt{1 + \Delta^2_{\max}/(2\epsilon^2)}$ (Lemma 21). We omit the proof since it is the same as the one of Lemma 20.

**Lemma 21.** *Let $\delta \in (0,1)$ Assume that there exists $\gamma_{\boldsymbol{\mu}} > 0$ such that for $\mathbb{E}_{\boldsymbol{\nu}}[T_{\boldsymbol{\mu},\gamma}(\omega^\star)] < +\infty$ for all $\gamma \in (0, \gamma_{\boldsymbol{\mu}})$, where $T_{\boldsymbol{\mu},\gamma}(w)$ is defined in Equation (29) and $\omega^\star$ is defined in Equation (30). Combining the private GLR stopping rule (Equation (16)) with private threshold (Equation (4)) yields a $\delta$-correct algorithm which satisfies that, for all $\boldsymbol{\nu}$ with mean $\mu$ such that $|a^\star(\boldsymbol{\mu})| = 1$,*

$$\limsup_{\delta \to 0} \frac{\mathbb{E}_{\boldsymbol{\nu}}[\tau_\delta]}{\log(1/\delta)} \leq 4T^\star_{\mathrm{KL}}(\boldsymbol{\mu}) \left( 1 + \sqrt{1 + \frac{\Delta^2_{\max}}{2\epsilon^2}} \right) \ .$$

### E.6  Connection to the lower bound

In this section, we compare the sample complexity lower bound of Corollary 1 with the sample complexity upper bound of Theorem 5.

**Simplification of the upper bound.**  The asymptotic expected sample complexity of AdaP-TT (Theorem 5) is upper bounded by

$$\limsup_{\delta \to 0} \frac{\mathbb{E}_{\boldsymbol{\nu}}[\tau_\delta]}{\log(1/\delta)} \leq 4T^\star_{\mathrm{KL},\beta}(\boldsymbol{\mu}) \left( 1 + \sqrt{1 + \frac{\Delta^2_{\max}}{2\epsilon^2}} \right)$$

$$\underset{(a)}{\leq} 4T^\star_{\mathrm{KL},\beta}(\boldsymbol{\mu}) \left( 2 + \frac{\Delta_{\max}}{\sqrt{2}\epsilon} \right)$$

where $T^\star_{\mathrm{KL},\beta}(\boldsymbol{\mu})$ is the $\beta$-characteristic time for Gaussian bandits, and (a) is due to the sub-additivity of the square root.

For $\beta = 1/2$, [Rus16] showed that $T^\star_{\mathrm{KL},1/2}(\boldsymbol{\mu}) \leq 2T^\star_{\mathrm{KL}}(\boldsymbol{\mu})$.

On the other hand, [GK16] showed that $H(\boldsymbol{\mu}) \leq T^\star_{\mathrm{KL}}(\boldsymbol{\mu}) \leq 2H(\boldsymbol{\mu})$, where $H(\boldsymbol{\mu}) \triangleq \sum_{a \in [K]} 2\Delta_a^{-2}$ with $\Delta_{a^\star} = \Delta_{\min}$.

Plugging these two inequalities in the upper bound of Theorem 5 with $\beta = 1/2$ gives that

$$\limsup_{\delta \to 0} \frac{\mathbb{E}_{\boldsymbol{\nu}}[\tau_\delta]}{\log(1/\delta)} \leq 8T^\star_{\mathrm{KL},1/2}(\boldsymbol{\mu}) + 16H(\boldsymbol{\mu})\frac{\Delta_{\max}}{\sqrt{2}\epsilon}$$

Since we consider Bernoulli distributions, we know that $0 < \Delta_{\min} \leq \Delta_{\max} < 1$. If we restrict ourselves to instances such that all the gaps have the same order of magnitude (Condition 1): there exists a constant $C \geq 1$ such that $\Delta_{\max} \leq C\Delta_{\min}$.

For such instances, we obtain

$$\limsup_{\delta \to 0} \frac{\mathbb{E}_{\boldsymbol{\nu}}[\tau_\delta]}{\log(1/\delta)} \leq 8T^\star_{\mathrm{KL},1/2}(\boldsymbol{\mu}) + 16H(\boldsymbol{\mu})\frac{C\Delta_{\min}}{\sqrt{2}\epsilon}$$

$$\leq 8T^\star_{\mathrm{KL},1/2}(\boldsymbol{\mu}) + 16\sqrt{2}\frac{C}{\epsilon}\left(\frac{1}{\Delta_{\min}} + \sum_{a=2}^{K}\frac{1}{\Delta_a}\right)$$

where the last inequality is due to $H(\boldsymbol{\mu})\Delta_{\min} \leq \frac{2}{\Delta_{\min}} + \sum_{a=2}^{K}\frac{2}{\Delta_a}$.

Finally using that $a + b \leq 2\max(a,b)$, we get that

$$\limsup_{\delta \to 0} \frac{\mathbb{E}_{\boldsymbol{\nu}}[\tau_\delta]}{\log(1/\delta)} \leq c\max\left\{T^\star_{\mathrm{KL},1/2}(\boldsymbol{\mu}), \frac{C}{\epsilon}\left(\frac{1}{\Delta_{\min}} + \sum_{a=2}^{K}\frac{1}{\Delta_a}\right)\right\}$$

for the universal constant $c = 45.26$.

**The lower bound for Bernoulli instances.**  For Bernoulli instances, Corollary 1 gives that the lower bound of the expected sample complexity of any $\delta$-correct $\epsilon$-global DP BAI strategy is

$$\limsup_{\delta \to 0} \frac{\mathbb{E}_{\boldsymbol{\nu}}[\tau_\delta]}{\log(1/\delta)} \geq \max\left\{T^\star_{\mathrm{KL}}(\boldsymbol{\nu}), \frac{1}{6\epsilon}\left(\frac{1}{\Delta_{\min}} + \sum_{a=2}^{K}\frac{1}{\Delta_a}\right)\right\} .$$

where we use Proposition 1 to replace $T^\star_{\mathrm{TV}}(\boldsymbol{\nu})$ and $T^\star_{\mathrm{KL}}(\boldsymbol{\nu})$ is the characteristic time for **Bernoulli bandits**.

**Upper-lower bound discussion for the two privacy regimes.**  In the low privacy regime, our upper bound retrieves $T^\star_{\mathrm{KL},1/2}(\boldsymbol{\mu})$ for Gaussian distributions. Since the rewards in the analysis are supposed Bernoulli, the mismatch from **exact** optimality is coming from the mismatch between the KL divergence of Bernoulli distributions and that of Gaussian, which is generally controllable in most instances where the means are far from the borders, i.e. 0, and 1. This is in essence, similar to the mismatch between UCB and KL-UCB in the regret-minimization literature (Chapter 10 in [LS20]). To overcome this mismatch, it is necessary to adapt the transportation costs to the family of distributions considered. In our setting, this can be done by using the Bernoulli KL rather than Gaussian KL in lines 12 and 16 of the algorithm. While the Top Two algorithms for Bernoulli distributions have been studied in [JDB+22], the analysis is more involved. Therefore, it would obfuscate where and how privacy is impacting the expected sample complexity.

In the high privacy regime, and for instances verifying Condition 1, our upper bound matches the lower bound $\epsilon^{-1}T^\star_{\mathrm{TV}}(\boldsymbol{\nu})$ up to a constant. Having matching upper and lower bounds *only* for high privacy regimes is an interesting phenomenon that appears in different settings of differential privacy literature, such as regret minimization [AB22], parameter estimation [CWZ21] and hidden probabilistic graphical models [NKS19]. We speculate two facets of this phenomenon:

1. Explicit bounds in high and low privacy regimes: We have matching bounds only in the high or low privacy regimes because the lower bounds are generally harder to explicit and understand in transitional phases. Thus, it is harder to claim optimality in those phases.

2. Information-theoretic roots: There might be a more profound information-theoretic reason in relation to [NKS19]. Indeed, there seems to be a link between the privacy budget $\epsilon$ and the information thresholds introduced in [NKS19]. Specifically, if the randomized mapping $\mathcal{F}$ in [3] satisfies DP, then the noisy information threshold and noiseless information threshold can be similarly written as a function of $\epsilon$ and the total variation. Finding a rigorous link between these two quantities is an interesting question to explore.

### E.7 Limitation and open problem

In the previous section (Appendix E.6), we argue that the upper and lower bounds match up to multiplicative constants in both privacy regimes, provided we restrict ourselves to instances satisfying Condition 1 (i.e. there exists $C \geq 1$ such that $\Delta_{\max} \leq C\Delta_{\min}$).

While this holds for numerous instances, it does not account for instances in which the gaps have different orders of magnitude. One example would be the regime where $\Delta_{\min} \to 0$ while $\Delta_{\max}$ is fixed, hence yielding $T^\star_{\mathrm{KL}}(\boldsymbol{\mu}) \to +\infty$.

In Appendix E.5 (see Lemma 20 and 21), we show that combining the private GLR stopping rule (Equation (16)) with any BAI algorithms whose empirical proportions converge towards the $\beta$-optimal allocation $\omega^\star_{\mathrm{KL},\beta}(\boldsymbol{\mu})$ for Gaussian bandits will incur a problem dependent multiplicative cost

$$1 + \sqrt{1 + \frac{\Delta^2_{\max}}{2\epsilon^2}} \tag{32}$$

in addition to the multiplicative four-factor due to doubling and forgetting.

In order to have matching upper and lower bounds, we would need to have $\Delta_{\min}$ instead of $\Delta_{\max}$ in Equation (32). Unfortunately, our results show that it is not possible when using the private GLR stopping rule (Equation (16)) and a sampling rule that is tailored to asymptotic ($\beta$-)optimality for Gaussian bandits. Therefore, this impossibility result holds for a large class of BAI sampling rules when adapting them to tackle private BAI by using the private GLR stopping rule (Equation (16)).

**Origin of this limitation.** The term $\Delta_{\max}$ appears due to the private stopping threshold that ensures $\delta$-correctness of the private GLR stopping rule (Equation (16)).

For asymptotically ($\beta$-)optimal algorithms, the additive term due to privacy in the threshold is of the order

$$\frac{1}{\epsilon^2}\left(\frac{1}{N_{n,a}} + \frac{1}{N_{n,a^\star}}\right)\log(1/\delta)^2 \approx_{n\to+\infty} \frac{T^\star_{\mathrm{KL}}(\boldsymbol{\mu})}{n}\frac{\Delta^2_a}{2\epsilon^2}\log(1/\delta)^2 .$$

Therefore, the private GLR stopping rule (Equation (16)) will stop when, for all $a \neq a^\star$,

$$\frac{n}{T^\star_{\mathrm{KL}}(\boldsymbol{\mu})} \geq 2\log(1/\delta) + \frac{T^\star_{\mathrm{KL}}(\boldsymbol{\mu})}{n}\frac{\Delta^2_a}{2\epsilon^2}\log(1/\delta)^2 ,$$

which yields the problem-dependent multiplicative cost $1 + \sqrt{1 + \frac{\Delta^2_{\max}}{2\epsilon^2}}$.

**Open problem.** This impossibility result is specific to the way we derive an $\epsilon$-DP version of the GLR stopping rule. Therefore, a natural question is whether it is possible to derive a better private GLR stopping rule to match the lower bound for all Bernoulli instances.

**A two-phase algorithm.** AdaP-TT tracks the non-private lower bound (i.e. $T^\star_{KL}$) as a non-private algorithm would do, and the additional cost tracking that privately is shown to be $T^\star_{TV}/\epsilon$ (up to constants for bandits verifying Condition 1), where the additional cost comes from the added Laplace noise. Another approach would be to first perform a test to determine in which privacy regime the policy resides, and then track (privately) the corresponding characteristic time. Such an algorithm would empirically estimate both $T^\star_{TV}$ and $T^\star_{KL}$ in the first phase. If the privacy budget $\epsilon$ is bigger than the empirical estimate of $T^\star_{TV}/T^\star_{KL}$, then this means that we are in the low privacy regime and the algorithm tracks (privately) the KL characteristic time in the second phase (as AdaP-TT does). However, if the privacy budget $\epsilon$ is smaller than the empirical estimate of $T^\star_{TV}/T^\star_{KL}$, then it is the high privacy regime and the algorithm tracks (privately) the TV characteristic time in the second phase. For such a two-phase algorithm to achieve optimality, properly tuning the amount of time spent in its first phase is a crucial step. Whether it is possible to analyze such an algorithm and quantify the proper tuning (if it even exists) is an interesting direction for future work.

**Non-asymptotic sample complexity of** AdaP-TT. In the *non-private FC-BAI literature* (i.e. $\epsilon = +\infty$), *there is no tight lower bound in the non-asymptotic regime (i.e. for any value of $\delta$)*. This is the main open problem in FC-BAI, and hence in DP-FC-BAI too. In the class of asymptotically ($\beta$-)optimal algorithms, TTUCB [JD22] is one of the few to have non-asymptotic guarantees. Adapting

the non-asymptotic analysis of [JD22] to the private AdaP-TT is an interesting direction for research. We conjecture that such an adaptation is possible (up to technicalities) by adding concentration terms linked to Laplace distribution and losing at least a multiplicative four-factor compared to TTUCB due to doubling and forgetting.

### E.8    Analysis of non-private AdaP-TT: Proof of Theorem 6

The proofs detailed in Appendices E.2 and E.3 can easily by adapted to provide guarantees on the non-private AdaP-TT, which relies on the non-private leader/challenger defined in Equation (20) and Equation (21).

The key difference between the non-private AdaP-TT and the private AdaP-TT lies in the definition of the stopping threshold. While the private GLR stopping rule (Equation (16)) has an additive terms in $\mathcal{O}(\log(1/\delta)^2/\epsilon)$ to cope for the uncertainty due to the Laplace noise, the non-private GLR stopping rule (Equation (13)) scales simply as $\log(1/\delta)$. This has drastic consequences in terms of asymptotic upper bound on the expected sample complexity.

It is direct to see that most manipulations from Appendix E.5 still holds for the non-private AdaP-TT. Therefore, we use the notations and conditions defined therein. We show that the non-private GLR stopping rule (Equation (13)) yields

$$
\begin{aligned}
&\min\{\tau_\delta, T\} - \kappa T \\
&\le \sum_{T \ge T_{k_n}^+ \ge \kappa T} (T_{k_n+2}^+ - T_{k_n+1}^+) \mathbb{1}\left(\tau_\delta > T_{k_n+1}^+\right) \\
&\le \sum_{T \ge T_{k_n}^+ \ge \kappa T} (T_{k_n+2}^+ - T_{k_n+1}^+)\ \mathbb{1}\left(\exists a \ne a^\star,\ \frac{(\hat{\mu}_{k_{n,a^\star}+1,a^\star} - \hat{\mu}_{k_{n,a}+1,a})^2}{1/\tilde{N}_{k_{n,a^\star}+1,a^\star} + 1/\tilde{N}_{k_{n,a}+1,a}}\right. \\
&\hspace{6cm} \left. < 2c_{(k_{n,a^\star}+1)(k_{n,a}+1)}\left(\tilde{N}_{k_{n,a^\star}+1,a^\star}, \tilde{N}_{k_{n,a}+1,a}, \delta\right)\right) \\
&\le \sum_{T \ge T_{k_n}^+ \ge \kappa T} (T_{k_n+2}^+ - T_{k_n+1}^+)\ \mathbb{1}\left((T_{k_n}^- - 1)\frac{1-\zeta}{T_{\mathrm{KL},\beta}^\star(\boldsymbol{\mu})} < 4\log(4 + \log T_{k_n}^- + \log(2 + \eta))\right. \\
&\hspace{2cm} \left. + 2\mathcal{C}_G\left(\log(1/\delta)/2 + s\log(2 + \log_2 T_{k_n}^- + \log_2(2 + \eta)) + \log((K-1)\zeta(s)^2)/2)\right)\right),
\end{aligned}
$$

Let $T_\zeta(\delta)$ defined as the largest deterministic time such that the above condition is satisfied when replacing $T_{k_n}^-$ by $(1-\kappa)T$. Let $k_\delta$ be the largest random vector of phases such that that $T_{k_\delta+1}^+ \le T_\zeta(\delta)$ almost surely, hence $T_{k_\delta+2}^+ > T_\zeta(\delta)$ almost surely. Then, using the above yields that $\tau_\delta \le T_{k_\delta+2}^+$ almost surely, hence

$$
\limsup_{\delta\to 0} \frac{\mathbb{E}_{\boldsymbol{\nu}}[\tau_\delta]}{\log(1/\delta)} \le \limsup_{\delta\to 0} \frac{\mathbb{E}_{\boldsymbol{\nu}}[T_{k_\delta+2}^+]}{\log(1/\delta)} \le (2+\eta)^2 \limsup_{\delta\to 0} \frac{\mathbb{E}_{\boldsymbol{\nu}}[T_{k_\delta+1}^+]}{\log(1/\delta)} \le (2+\eta)^2 \limsup_{\delta\to 0} \frac{T_\zeta(\delta)}{\log(1/\delta)},
$$

**First-order terms.** Since we are only interested in upper bounding $\limsup_{\delta\to 0} \frac{T_\zeta(\delta)}{\log(1/\delta)}$, we can safely drop the second orders terms in $T$ and $\log(1/\delta)$. This allows us to remove the terms in $\mathcal{O}(\log\log T)$ and in $\mathcal{O}(\log\log(1/\delta))$. Using that $\mathcal{C}_G(x) = x + \mathcal{O}(\log x)$, tedious manipulations yields that

$$
\limsup_{\delta\to 0} \frac{T_\zeta(\delta)}{\log(1/\delta)} \le \frac{T_{\mathrm{KL},\beta}^\star(\boldsymbol{\mu})}{(1-\kappa)(1-\zeta)}\ .
$$

Letting $\kappa$, $\eta$ and $\zeta$ goes to zero yields that

$$
\limsup_{\delta\to 0} \frac{\mathbb{E}_{\boldsymbol{\nu}}[\tau_\delta]}{\log(1/\delta)} \le 4T_{\mathrm{KL},\beta}^\star(\boldsymbol{\mu})\ .
$$

## F Extended experimental analysis

In this section, we perform additional experiments to compare AdaP-TT and DP-SE for FC-BAI. We test the two algorithms in six bandit environments with Bernoulli distributions, as defined by [SS19], namely

$$\mu_1 = (0.95, 0.9, 0.9, 0.9, 0.5), \qquad \mu_2 = (0.75, 0.7, 0.7, 0.7, 0.7),$$
$$\mu_3 = (0, 0.25, 0.5, 0.75, 1), \qquad \mu_4 = (0.75, 0.625, 0.5, 0.375, 0.25)\},$$
$$\mu_5 = (0.75, 0.53125, 0.375, 0.28125, 0.25), \quad \mu_6 = (0.75, 0.71875, 0.625, 0.46875, 0.25)\}.$$

For each Bernoulli instance, we implement the algorithms with

$$\epsilon \in \{0.001, 0.005, 0.01, 0.05, 0.1, 0.2, 0.3, 0.4, 0.5, 0.6, 0.7, 0.8, 0.9, 1, 10\},$$

and a risk level $\delta = 0.01$. We verify empirically that the algorithms are $\delta$-correct by running each algorithm 100 times. In Figure 2, we plot the evolution of the average stopping time and standard deviation with respect to the privacy budget $\epsilon$. All the algorithms are implemented in Python (version 3.8) and are tested with an 8-core 64-bits Intel i5@1.6 GHz CPU.

All the experiments validate the same conclusions as the ones reached in Section 5, i.e.

1. AdaP-TT requires fewer samples to provide a $\delta$-correct answer,
2. there exists two privacy regimes, and in the low-privacy regime, the sample complexity is independent of the privacy budget.

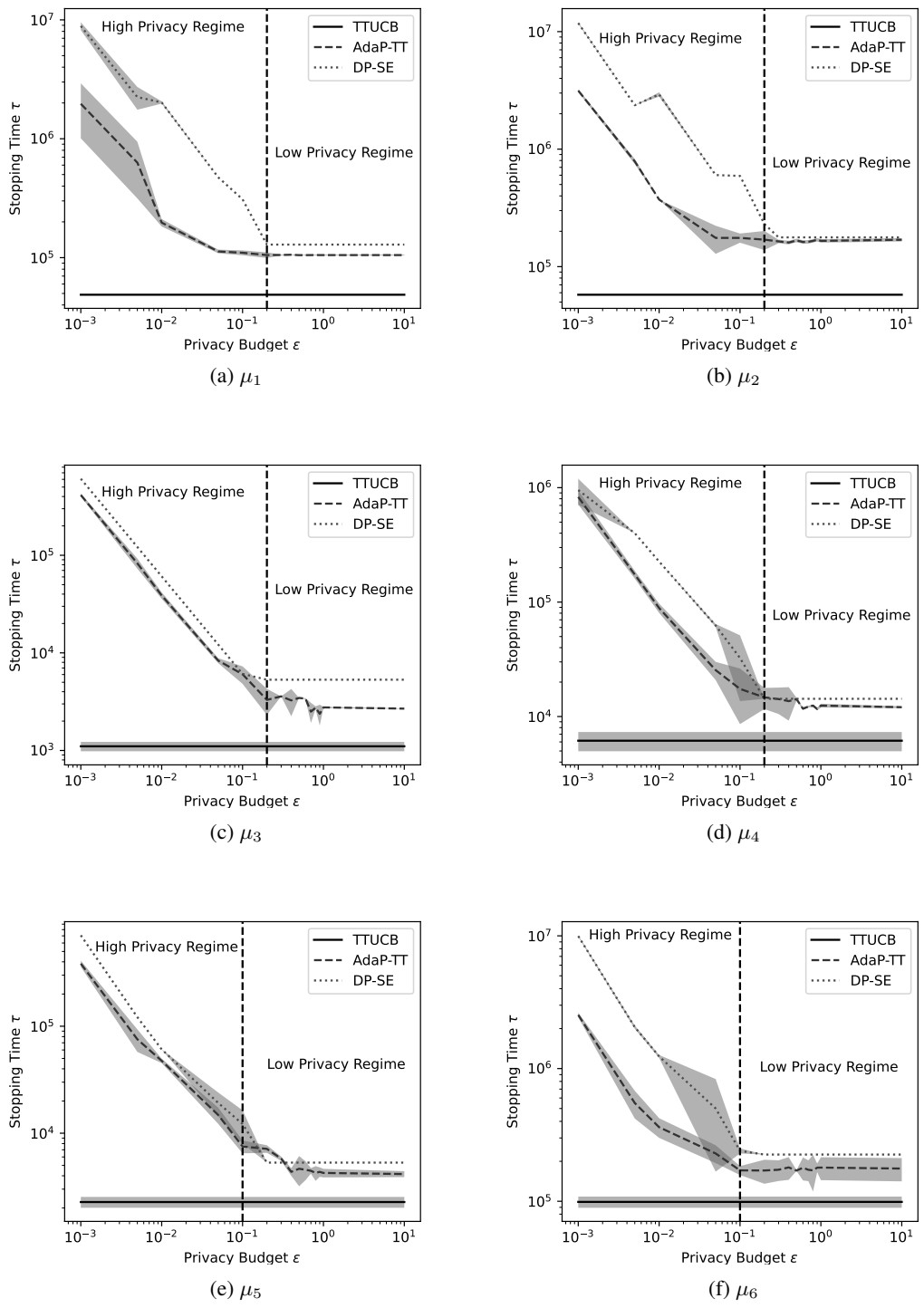

Figure 2: Evolution of the stopping time $\tau$ (mean $\pm$ std. over 100 runs) of AdaP-TT, DP-SE, and TTUCB with respect to the privacy budget $\epsilon$ for $\delta = 10^{-2}$ on different Bernoulli instances. The shaded vertical line separates the two privacy regimes. AdaP-TT outperforms DP-SE.

