# OpenReview forum: "On the Complexity of Differentially Private Best-Arm Identification with Fixed Confidence"
_NeurIPS.cc/2023/Conference — NeurIPS 2023 poster_

### Official Review · Reviewer_rwuX · 2023-07-04

**Soundness:** 3 good
**Presentation:** 3 good
**Contribution:** 2 fair
**Rating:** 6
**Confidence:** 3

**Summary:**

The paper considers the best arm identification (BAI) problem under differential privacy (DP) constraints, where the goal is to privately identify the arm with the highest expected reward with probability at least $1 - \delta$. The paper proposes a DP variant of the top two algorithm, which achieves near-optimal performance in the high privacy regime. The paper also proves an information-theoretic instance-dependent lower bound for the DP-BAI problem. The paper also conducts experiments on synthetic data to demonstrate the efficiency of the proposed algorithm.

**Strengths:**

The proposed upper and lower bounds match each other in the high-privacy regime when the true means are away from 0 or 1. This provides an up-to-constant characterization of the asymptotic sample complexity for DP-BAI.

The proposed algorithm empirically outperforms a natural DP variant of the successive elimination algorithm.

The paper is also nicely written and easy to read.


**Weaknesses:**

While DP-BAI is an interesting problem to study, the techniques used in the paper are primarily natural variants of previous work. In particular, nonprivate-BAI and the closely related DP regret minimization in the bandit setting have been studied. Hence the technical novelty of the paper is restricted.

It would be nice if the authors could comment on the non-asymptotic optimality of the proposed algorithm.

Minor comments:
1. Not sure if colored texts are supported for NeurIPS submissions.
2. Definition 1: The letter d is a bit overloaded. Maybe it is better to use $D$ or $\mathcal{D}$ to refer to datasets, which is used later in the paper.
3. Line 207: Might be good to have parentheses after sup.
4. Can you comment on how $T^*_{TV}(\nu)$ and  $T^*_{KL}(\nu)$ relate to each other?
5. Algorithm 2: $L_n$'s are not initialized.

**Questions:**

See comments above.

**Limitations:**

Yes.

---

> ### Author Rebuttal · Authors · 2023-08-09
>
> We thank the reviewer for the careful reading and astute observations to improve the manuscript.
>
> **Technical novelty.** Please refer to the general comment.
>
> **Non-asymptotic sample complexity of AdapTT.** Please refer to the general comment.
>
> **Minor Typos.** Thanks for detecting the typos. We will modify them in the final version of the paper. Specifically, we will change $d$ to $D$ in Definition 1, add parentheses after the sup in Line 207, and initialize $L_n$ to zero at the start of Algorithm 1. While colours are allowed in the figures, official guidelines do not specify anything for text. Since it is recommended to have a paper which is legible when printed in black/white, we will avoid the use of colour in the revised version.
>
> **Relation between $T^\star_{TV}$ and $T^\star_{KL}$.** One can use Pinsker's inequality to connect the TV characteristic time and the KL characteristic time: $T^\star_{TV}(\nu) \geq \sqrt{2 T^\star_{KL}(\nu) }.$
>
> We will add this comment in the final version of the paper.
>
> Here, we present the exact calculations for completeness. For every alternative mean parameter $\lambda$ and every arm $a$, using Pinkser's inequality, we have that $d_{TV} ( \mu_a, \lambda_a) \leq \sqrt{\frac{1}{2} d_{KL} ( \mu_a, \lambda_a)}$.
> Therefore, for every allocation over arms $\omega$, we have
> $
> \sum_a \omega_a d_{TV} ( \mu_a, \lambda_a) \leq \sum_a \omega_a \sqrt{\frac{1}{2} d_{KL} ( \mu_a, \lambda_a)} \leq \sqrt{\frac{1}{2}\sum_a \omega_a d_{KL} ( \mu_a, \lambda_a)}.
> $
> Taking the supremum over the simplex and the infimum over the set of alternative mean parameters yields $(T^\star_{TV}(\nu))^{-1} \leq \sqrt{\frac{1}{2} (T^\star_{KL}(\nu))^{-1} }$.
> This concludes the proof.
>
> Hope our response answers your concerns. Looking forward to addressing any further comments.

---

> > ### Comment · Reviewer_rwuX · 2023-08-16
> >
> > Thank the authors for the detailed response and addressing my concerns. I will increase my score. I would suggest the authors to include a more detailed discussion of the technical novelty in future revisions.

---

### Official Review · Reviewer_jHc4 · 2023-07-05

**Soundness:** 4 excellent
**Presentation:** 4 excellent
**Contribution:** 4 excellent
**Rating:** 8
**Confidence:** 1

**Summary:**

This work provides some insight into using the Laplace mechanism on the algorithms concerning the best arm identification problem bandits with a constraint of fixed confidence. The analysis carried out in this work enumerates general properties of differentially private FC-BAI and then proposes an algorithm based on those bounds.

**Strengths:**

Analysis carried out in this work is general to all algorithms that are designed to solve the BAI problem where there are some guarantees presented on the sample complexity of such class of algorithms.

**Weaknesses:**

No weaknesses found in this paper, as per the knowledge of the reviewer.

**Questions:**

-

**Limitations:**

No additional limitations can be seen in this work. All the limitations, to the reviewer's knowledge, have been addressed properly in the paper.

---

> ### Author Rebuttal · Authors · 2023-08-09
>
> We thank the reviewer for the time spent reviewing and the positive view of this work. We hope that the rebuttals done for the other reviewers will answer your future questions. Also, let us know if you have any other concerns.

---

### Official Review · Reviewer_RUVA · 2023-07-08

**Soundness:** 3 good
**Presentation:** 1 poor
**Contribution:** 3 good
**Rating:** 5
**Confidence:** 2

**Summary:**

This paper studies the problem of Best Arm Identification (BAI) for bandit problems under the constraints of differential privacy. There are some $K$ distributions and the solver interacts with the arms over a sequence of rounds. In each round they choose a distribution and receive a reward according to that distribution. The goal is to identify the arm with the highest expected reward with certainty $1-\delta$ where $\delta$ is a Fixed Confidence (FC) parameter given to the solver.

Motivated by real-world problems like clinical trials for testing doses/efficacy of drugs, the authors suggest studying this problem under the constraint of differential privacy - this is relevant when the rewards received might be private information, such as the medical data of a patient's response to the drug/dose they were administered.

The contributions of this paper are as follows:
1. The authors derive a lower bound on the sample complexity for DP-FC-BAI problem - their results suggest that the problem can be characterized in the low-privacy and high privacy regimes. In the former it seems that the sample complexity of the problem is unaffected by the DP constraint.
2. They present an algorithm, AdaP-TT, which is a differentially private version of the Top Two Upper Confidence Bound (TTUCB) algorithm from prior work. The sample complexity of this algorithm matches the lower bound asymptotically in the high-privacy regime.
3. The conduct some experiments showing that their algorithm has lower sample complexity compared to another differentially private approach (DP Successive-Elimination).

**Strengths:**

1. The problem being studied is very well-motivated - in particular the example of dose finding and the necessity of privacy preserving BAI is compelling.
2. The authors provide a lower bound and characterize the hardness of the problem.
3. The algorithm they provide (privatizing the optimal sample complexity achieving method in the non-private setting) seems to work well in practice based on their experiments. At a cursory look the performance of the algorithms seem to corroborate the constant factors in the sample complexity terms that the authors derived.

**Weaknesses:**

1. I found this paper very hard to read - the questions being considered by the authors make sense but the expressions derived are not easy to make sense of. The presentation is hard to follow - some terms are introduced without much context (such as ``Top Two algorithms" on the second page) and this paper seems hard for non-experts to follow.
2. On page 2 the authors say "A calibrated amount of noise, from a Laplace [DR14] or Gaussian distribution [DRS22], is injected into an algorithm to ensure DP. The noise scale is set to be proportional to the algorithm’s sensitivity and inversely proportional to the privacy budget $\varepsilon$." I think this statement is somewhat misleading - DP can be achieved in many ways and obfuscating noise is routinely drawn from many distributions (including the Gumbel distribuion, Poisson distributions, or other ad-hoc methods depending on the setting). Indeed, as the authors themselves find in their lower bound, sometimes privacy can be achieved for free. Similarly, the way the scale of the noise is chosen can be very involved - it is not always possible to derive a uniform bound on the sensitivity of an algorithm or such a method could give a very sub-optimal result. It is also important to note that not all problems admit DP solutions - there is no meaningful DP way to compute the XOR of $n$ private bits, for instance.
3. The expressions in the lower bound are hard to interpret - ideally one might want to characterize them only in terms of the problem parameters i.e. the number of arms $K$, the confidence parameter $\delta$ and the privacy parameter $\varepsilon$ - the given bound seems to be more general which is great (as it describes the hardness in terms of the expected rewards), but is it possible to instead substitute a hard instance for the $\mu_i$ and $\Delta_a$ values and derive a more readable corollary?
4. The approach does not seem to be very novel as it reduces to essentially just applying the Laplace mechanism at one point of the TTUCB algorithm from prior work. The sensitivity admits a straightforward bound as well.

**Questions:**

1. You define a low and high privacy regime based on $\varepsilon$, $T_{KL}^*$ and $T_{TV}^*$, but the expressions for these depend upon $\mu_i$ which the solver does not have access to. Operationally, is there any way for the solver to know whether they are in the low privacy or high privacy regime?
2. The claim that there is no additional cost of privacy in terms of sample complexity for the low-privacy regime seems to need some more justification - there is a matching upper bound in the high-privacy regime but since there is no matching upper bound for the low-privacy regime, couldn't there just be a stronger lower bound which holds that we are not yet aware of?

**Limitations:**

Potential negative societal impacts do not need to be discussed here as this is primarily a theoretical paper.

---

> ### Author Rebuttal · Authors · 2023-08-09
>
> We would like to thank you for your time, thorough feedback, and precise remarks.
>
> We first answer the points raised in the "weaknesses" section.
>
> **1. The paper is hard to read for non-experts.** The paper studies private BAI and builds upon recent advances in both domains. As such, for non-experts in both domains, it is harder to grasp the key novel insights. With the additional page available for the final version, we will improve the presentation by providing more context for non-experts in both domains, i.e. differential privacy and best arm identification, and providing more intuition behind the different results.
>
> **2. DP can be achieved in many ways.** We thank the reviewer for this precise remark. As correctly pointed out, DP can be achieved in many different ways. The goal of the sentence was to introduce the Laplace and Gaussian mechanisms. A more nuanced and correct way is to say "A popular way to achieve DP is to inject a calibrated amount of noise, from a Laplace [DR14] or Gaussian distribution [DRS22], into the algorithm. The scale of the noise is set to be proportional to the algorithm’s sensitivity and inversely proportional to the privacy budget $\epsilon$." We will modify the sentence in the revised version.
>
> **3. The expression of the lower bound is hard to interpret.** The sample complexity lower bound derived in this paper is problem dependent, i.e. depends on the hardness of the environment the policy is interacting with, which is characterized by the expected rewards $(\mu_i)_{i = 1}^K$. *The FC-BAI literature is mainly interested in problem-dependent bounds*. As pointed out by the reviewer, this is a more general result and more interesting when trying to characterise the hardness of a problem. It is always possible to derive a worst-case lower bound from the problem-dependent one, by optimizing for the worst environment (i.e. the environment with the highest characteristic time in the environment class). This may be even less interpretable than the problem-dependent lower bound. The lower bound of Theorem 2 is expressed to draw a direct parallel with the one known for FC-BAI (see e.g. [GK16]). We provide a simpler version in Corollary 1 and a full characterisation in Proposition 1.
>
> **4. The novelty of the approach.** Please refer to the general comment for the novelty and technical contributions. Applying the Laplace mechanism to the Top Two algorithm to have a sample complexity that matches the lower bound is not as straightforward as it seems. For example, using the tree-based mechanism [DNPR10], which is also based on the Laplace mechanism, to estimate the means privately does not achieve the lower bound. Also, the doubling and forgetting parts are important in order to have less sensitivity and add just the required amount of noise to achieve the lower bound.
>
> We now answer your other questions.
>
> **1. $T^\star_{KL}$ and $T^\star_{TV}$ depend on $(\mu_i)$? Is there a way for the policy to know in which privacy regime it resides?**
> - First, we would like to mention that $T^\star_{KL}$ is not our contribution. It is a classic and well-studied quantity in the BAI literature and constitutes a crucial part of the design of every optimal algorithm in this field (see e.g. [GK16]). If $T^\star_{KL}$ completely characterises the complexity of the non-private problem, $T^\star_{TV}$ dictates the sample complexity at the high privacy regime and can be seen as a private counterpart of $T^\star_{KL}$.
>
> - Both $T^\star_{KL}$ and $T^\star_{TV}$ depend on $(\mu_i)$, as they are *problem-dependent lower bounds*. Indeed, $(\mu_i)$ is unknown to the policy. This means that, for the policy to know in which privacy regime it resides, tracking $T^\star_{KL}$ and $T^\star_{TV}$ using the empirical means estimate $(\hat\mu_i)$ is a natural way. If $\epsilon$ is less than the estimated $T^\star_{TV}$ divided by the estimated $T^\star_{KL}$, it is the high privacy regime, otherwise, it is the low privacy regime. AdaP-TT is not doing that. It only tracks the non-private lower bound (i.e. $T^\star_{KL}$) as a non-private algorithm would do, and the additional cost tracking that privately is shown to be $T^\star_{TV}/\epsilon$, which comes from the added Laplace noise. Another approach would be to first perform a test to determine in which regime the policy resides, and then track (privately) the corresponding characteristic time. Such a two-phase algorithm requires properly choosing the amount of time spent in its first phase. Without proper tuning, it has no chance to reach asymptotic optimality. Whether it is possible to analyze such an algorithm and quantify the proper tuning is an interesting direction for future work.
>
> **2. No additional cost of privacy in terms of sample complexity for the low-privacy regime.** The sample complexity of AdaP-TT is shown to achieve $T^\star_{KL}$ of Gaussian distributions in the low-privacy regime. Since the rewards in the analysis are supposed Bernoulli, the mismatch from **exact** optimality is coming from the mismatch between the KL divergence of Bernoulli distributions and that of Gaussian, which is generally controllable in most instances where the means are far from the borders, i.e. $0$, and $1$. This is in essence, similar to the mismatch between UCB and KL-UCB in the regret-minimization literature (Chapter 10 in [LS18]). To overcome this mismatch, it is necessary to adapt the transportation costs to the family of distributions considered. In our setting, this can be done by using the Bernoulli KL rather than Gaussian KL in lines 12 and 16 of the algorithm. While the Top Two algorithms for Bernoulli distributions have been studied in [JDB+22], the analysis is more involved. Therefore, it would obfuscate where and how privacy is impacting the expected sample complexity.
>
> Hope that our response addresses the reviewer's questions. Let us know if you have any other queries. If our response convinces you, it would be helpful to raise the score.

---

> > ### Comment · Reviewer_RUVA · 2023-08-18
> > **Response to rebuttal**
> >
> > Thank you for your detailed response. After reading your meta rebuttal and the discussions with the other reviewers I would be happy to change my score from a 4 to a 5. As suggested elsewhere some more work could be done on general readability and highlighting the novelty of your work.

---

### Official Review · Reviewer_Qtw8 · 2023-07-17

**Soundness:** 3 good
**Presentation:** 3 good
**Contribution:** 4 excellent
**Rating:** 8
**Confidence:** 3

**Summary:**

This work provides novel theoretical guarantees to quantify the cost of privacy in DP-FC-BAI (and $\epsilon$ global DP). Specifically, the authors show that the complexity depends on an information theoretic total variation metric. Although, this holds in the high privacy regime, this is reasonable and possibly connected with results on strong data processing inequalities in prior works. Through their lower bound ,the authors show consistency and recover the non-private regime as well. Additionally, the authors propose a variant of existing algorithms with asymptotically matching upper bound.

**Strengths:**

The originality of the paper is clear. The presentation is clear and the contributions are significant. The total variation characteristic time is a novel quantity that it is introduced in this work. The proposed algorithm (AdaP-TT) is also novel and achieves matching guarantees with the proposed lower bound. The authors manage to characterize a phase transition effect: in the low privacy regime the privacy can be achieved for free. This is correct and has been observed in prior works as well (but for Thomson sampling). The results of the work are the first lower bounds for DP BAI, and they successfully characterize the hardness of the problem.

**Weaknesses:**

The proof techniques appear to be mostly applications of known proof techniques. Sketches of proofs for the lower and upper bounds can potentially improve further the quality of the paper. The sketches should provide details on the novelty of the proof technique and several steps of the proof (similarly to proof of Lemma 1 in the Appendix).


A few references are missing, for instance "On the Renyi differential privacy of the shuffle model" by Girgis et al., "Secure best arm identification in multi-armed bandits"  by Ciucanu at al.

Finally, the format of some references in the bibliography can be improved, some references point to older version of prior works on arxiv, while the papers have been published, for instance see [KNSS20].




**Questions:**

Which parts of the proofs technique are novel? Could you please include proof sketches that emphasize on the proof technique?

Does the total variation uniquely characterize the hardness of the problem? Can the lower bounds be expressed in terms of the KL divergence instead?

**Limitations:**

The authors adequately addressed the limitations.

---

> ### Author Rebuttal · Authors · 2023-08-09
>
> We would like to thank the reviewer for the time spent reviewing, careful reading, and kind words about the contributions' novelty and significance. It is really encouraging for us.
>
> **Missing references.** We thank the reviewer for pointing us to these references. We will add them to the Related Works section in the revised version, and update the reference to [KNSS20].
>
> **Technical novelty; "proof sketches that emphasize on the proof technique".** Please refer to the general comment for the technical novelty. We also thank the reviewer for the advice, indeed we will add more intuition in the proof sketches and emphasize more the technical contributions pointed out in the general comment.
>
> **Does the total variation uniquely characterize the hardness of the problem? Can the lower bounds be expressed in terms of the KL divergence instead?** We thank the reviewer for this excellent question.
>
> Our lower bound suggests that the hardness of the DP-FC-BAI problem is characterized by $T^\star_{TV}$, which is a total variation counterpart of the classic KL-based characteristic time $T^\star_{KL}$ in FC-BAI [GK16]. The total variation distance appears to be the natural measure to quantify the hardness of privacy in many other settings (regret minimization [1], Karwa-Vadhan lemma [2], Differentially Private Assouad, Fano, and Le Cam [3]). The high-level intuition is that: Pure DP can be seen as a multiplicative stability constraint of $e^\epsilon$ when one data point changes. Group privacy tells that if two datasets differ in $d_{ham}$ points, then one incurs a factor $e^ {d_{ham}~\epsilon}$. Now, if I have $n$ iid points from a distribution $P$ and $n$ iid points from a distribution $Q$, the Karwa-Vadhan lemma states that we incur a factor $e^{(nTV(P,Q)) ~ \epsilon}$. This is proved by building a maximal coupling, which is the coupling that minimizes the Hamming distance in expectation. In brief, *the total variation naturally appears since it is the quantity that characterises the hardness of the optimal transport problem minimizing the hamming distance*, i.e $TV(P, Q) = \inf_{(X,Y)\sim (P,Q)} E(1_{X \neq Y})$.
> However, it is possible that the problem can be characterized by other f-divergences/measures.
>
> Finally, one can always go from TV to KL using Pinsker's inequality though that would always be less tight than the TV-based lower bound. Please refer to the answer to the reviewer rwuX for the inequality between $T^\star_{TV}$ and $T^\star_{KL}$, i.e. $T^\star_{TV}(\nu) \geq \sqrt{2 T^\star_{KL}(\nu)}$.
>
> Hope that our response addresses the reviewer's questions. Let us know if you have any other queries.
>
> [1] Azize, A., & Basu, D. (2022). When privacy meets partial information: A refined analysis of differentially private bandits. Advances in Neural Information Processing Systems, 35, 32199-32210.
>
> [2] Karwa, V., & Vadhan, S. (2017). Finite sample differentially private confidence intervals. arXiv preprint arXiv:1711.03908.
>
> [3] Acharya, J., Sun, Z., & Zhang, H. (2021, March). Differentially private assouad, fano, and le cam. In Algorithmic Learning Theory (pp. 48-78). PMLR.

---

> > ### Comment · Reviewer_Qtw8 · 2023-08-10
> > **Response to the Rebuttal**
> >
> > Thank you for your detailed response. I have another question regarding the tightness of the bound in the high privacy regime. This property seems to hold for other settings as well. For instance in hidden probabilistic graphical models, where only noisy observations are available there exist similar properties. For instance, in the work "Optimal Rates for Learning Hidden Tree Structures" by Nikolakakis et al. the upper and lower bounds appear to match for the highly noisy regime (only) as well. I think that the role of the privacy cost $\epsilon$ has the quantity noisy information threshold as it is defined in the paper "Optimal Rates for Learning Hidden Tree Structures". Thus the property of tightness in the highly noisy regime can go beyond the setting of differential privacy and raise other information theoretic questions. Could the authors please verify if this is the case. If there is indeed a connection, it would be useful to compare this phenomenon with noisy probabilistic models in prior work "Optimal Rates for Learning Hidden Tree Structures" and highlight such similar properties.

---

> > > ### Author Response · Authors · 2023-08-16
> > > **Tightness in the high privacy regime**
> > >
> > > We thank the reviewer for the interesting question.
> > > Indeed, deriving matching upper and lower bounds for high privacy regimes is an interesting result that appears in different settings of differential privacy literature, such as regret minimization [1] and parameter estimation [2]. This also seems to be the fact for [3] that the reviewer mentioned.
> > > We speculate two facets of this phenomenon:
> > >
> > > 1. Explicit bounds in high and low privacy regimes: We have matching bounds only in the high or low privacy regimes because the lower bounds are generally harder to explicit and understand in transitional phases. Thus, it is harder to claim optimality in those phases.
> > >
> > > 2. Information-theoretic roots: We agree that there might be a more profound information-theoretic reason in relation to [3]. Indeed, there seems to be a link between the privacy budget $\epsilon$ and the information thresholds introduced in [3]. Specifically, if the randomized mapping $\mathcal{F}$ in [3] satisfies DP, then the noisy information threshold and noiseless information threshold can be similarly written as a function of $\epsilon$ and the total variation. Finding a rigorous link between these two quantities is an interesting question to explore. In this direction, it would be worth studying prior work [4] that aimed to express differential privacy as a mutual information constraint.
> > >
> > > We will add a remark about this phenomenon after the section about the comparison to the lower bound.
> > >
> > >
> > > [1] Azize, A., & Basu, D. (2022). When privacy meets partial information: A refined analysis of differentially private bandits. Advances in Neural Information Processing Systems, 35, 32199-32210.
> > >
> > > [2] Cai, T. T., Wang, Y., & Zhang, L. (2021). The cost of privacy: Optimal rates of convergence for parameter estimation with differential privacy. The Annals of Statistics, 49(5), 2825-2850.
> > >
> > > [3] Nikolakakis, K. E., Kalogerias, D. S., & Sarwate, A. D. (2019). Optimal Rates for Learning Hidden Tree Structures. arXiv preprint arXiv:1909.09596.
> > >
> > > [4] Cuff, P., & Yu, L. (2016). Differential privacy as a mutual information constraint. In Proceedings of the 2016 ACM SIGSAC Conference on Computer and Communications Security (pp. 43-54).

---

> > > > ### Comment · Reviewer_Qtw8 · 2023-08-16
> > > > **Thank you!**
> > > >
> > > > Thank you for your response. I believe that the suggested changes and general impact of the results are also important from an information theoretic point of view (and a very short discussion can be included in the paper). Parts of the proof or a proof sketch should be included in the main part of the paper. I think this work has a great impact also on problems beyond privacy.  I will raise the score.

---

### Official Review · Reviewer_Z6yo · 2023-07-19

**Soundness:** 3 good
**Presentation:** 1 poor
**Contribution:** 3 good
**Rating:** 5
**Confidence:** 3

**Summary:**

This paper studies the problem of BAI with fixed confidence under ϵ-global Differential Privacy (DP). The authors derive a lower bound on the sample complexity of any δ-correct BAI algorithm satisfying ϵ-global DP. In addition, the authors also design an ϵ-global DP variant of the Top Two algorithm, named AdaP-TT. AdaP-TT runs in arm-dependent adaptive episodes and adds Laplace noise to ensure a good privacy-utility trade-off. The authors provide an asymptotic upper bound on the sample complexity of AdaP-TT, which matches the lower bound up to multiplicative constants in the high-privacy regime. Finally, the authors also conduct experiments to validate their theoretical results.

**Strengths:**

1.	The studied problem, differential privacy in best arm identification, is interesting and finds applications such as clinical trials and hyper-parameter tuning.
2.	The provided lower bound result is very interesting. The authors give a deep discussion on the problem hardness in two regimes. In the high-privacy regime, the hardness depends on a coupled effect of privacy and a novel information-theoretic quantity, i.e., total variation characteristic time. In the low-privacy regime, the sample complexity lower bound is the same as that in classic (non-private) best arm identification.


**Weaknesses:**

1.	The writing of this paper should be improved. This paper is very dense and difficult to follow.
2.	The authors only provide asymptotic sample complexity for their algorithm AdaP-TT. Can the authors provide a non-asymptotic sample complexity, or give some ideas?
3.	Can the authors discuss the technical novelty compared to existing works on DP with regret minimization? What is the unique challenge brought by the best arm identification setting?


**Questions:**

Please see the weaknesses above.

**Limitations:**

Please see the weaknesses above.

---

> ### Author Rebuttal · Authors · 2023-08-09
>
> We thank the reviewer for the constructive feedback and kind words concerning the importance of the derived lower bound.
>
> **1. Dense paper and difficult to follow.** The paper investigates private BAI and draws on very recent developments in both fields (like AKR21, AB22, JDB+22, JD22 etc.). As a result, we understand that the non-experts in either of the fields may find the paper difficult to grasp. Thus, even under the space constraint of the submission, we tried to provide proof sketches, connect the relevant results, and discuss the consequences of every theoretical result.  Using the extra page in the final version, we will focus on the paper's presentation by providing additional context for the non-experts in both domains, namely differential privacy and best arm identification, as well as offering more intuition behind the various results and proofs.
>
> **2. Non-asymptotic sample complexity of AdapTT.** Please refer to the general comment for a detailed response.
>
> **3. Technical novelty.** Please refer to the general comment for the technical contribution.
>
> We hope that our response addresses the reviewer’s questions. We look forward to responding to further queries.

---

> > ### Comment · Reviewer_Z6yo · 2023-08-15
> > **Thank the authors for their response.**
> >
> > Thank the authors for their response. I tend to keep my score.

---

### Official Review · Reviewer_NTgq · 2023-07-25

**Soundness:** 2 fair
**Presentation:** 3 good
**Contribution:** 2 fair
**Rating:** 7
**Confidence:** 4

**Summary:**

This paper studies the problem of best arm identification with fixed confidence under ϵ-global Differential Privacy. First, they provide a lower bound of the BAI-DP problem showing that two different regimes depend on the privacy budget of $\epsilon$. They propose an ϵ-global DP variant of the Top Two algorithm, called AdaP-TT, and provide its asymptotic sample complexity. They also conduct experiments that confirm the theoretical validity.

**Strengths:**

(originality)

-They first derive the hardness result giving a lower bound for the problem of best arm identification with fixed confidence under ϵ-global Differential Privacy.
For the high-private regime, the lower bound depends on the Total Variation Characteristic Time, which shares a similar spirit with the regret minimization with ϵ-global DP [AB22]. For low-private regimes, the lower bound reduces to the classical no-private lower bound.

-As an algorithmic contribution, they devise variants of Top Two algorithms, called AdaP-TT, based on non-private TTUCB[JD22]. They propose a sample complexity in an asymptotic regime, which is the order-optimal up to some constants.


(quality) The paper is well-structured and easy to follow.

**Weaknesses:**

-As discussed in the paper, in BAI setting the only Information is leaked when publishing all the sequence of actions $\{a_s\}_{s<t}$ and the final recommendation a^*, which might not be frequently common situations, since BAI setting, the decision of final output is more interested. The presented problem is more of an artificial problem, although theoretical results are valid and new insights.

-Related above, the privacy issues should be more important for the case where multiple players may exist rather than only a single player is involved, as in federated learning. With regard to multi-agent bandit problems, recently the following paper studied the best arm identification with DP setting and propose the DP successive elimination based on [SS19].


 [Rio et al 2023] Multi-Agent Best Arm Identification with Private Communications.
Alexandre Rio, Merwan Barlier, Igor Colin, Marta Soare. Proceedings of the 40th International Conference on Machine Learning, PMLR 202:29082-29102, 2023.


I believe that a comparison with this paper should be required before the publication, even though [Rio et al 2023] might have been public after the NeurIPS submission date (So lacking this comparison in the current is not the reason for rejection).
 [Rio et al 2023]  consider a lower bound but only for the extreme non-private setting, where agents share all their raw reward samples with the central coordinator. Therefore, the submitted paper’s analysis for lower bound might be still novel. However, DP successive elimination proposed in [Rio et al 2023] could cover the single agent BAI-DP. Therefore the authors could do a comparison with this paper.
======After rebuttal======
My concerns are fully addressed. Hence I changed my score from 5 to 7.

**Questions:**

Could you provide comparison with [Rio et al 2023]?

---

> ### Author Rebuttal · Authors · 2023-08-09
>
> We would like to thank the reviewer for the constructive feedback and the interesting questions.
>
> **Real application motivating privacy at the level of intermediate actions and the single-agent setting.** We respectfully disagree with the assessment of the reviewer regarding the practical grounding of the setting studied in this work. Considering privacy at the level of intermediate actions for a single agent is **not** an artificial problem. It is based on a real-world application of adaptive dose-finding trials (see Example 1 in lines 39--50).
>
> First, in many small-scale clinical trials, only one physician decides on which dose to give to a patient taken from a local pool of volunteers. Therefore, it is truly a single-agent setting. While the multi-agent setting and federated learning can tackle large-scale clinical trials taking place at several locations simultaneously, we also need to provide tools for small-scale ones.
>
> Second, both small-scale and large-scale clinical trials have to publish their experimental findings in order to obtain the right to sell their medicine. This can be done publicly through a medical journal, or confidentially by providing the necessary information to local/international health authorities. The experimental findings of a clinical trial are not reduced solely to the final recommendation $\hat a_{\tau_{\delta}}$, and the scientists need to detail the experimental protocol. This includes the dose allocated to each patient, which are the intermediate actions $(a_t)$ for ${1 \leq t < \tau_\delta}$. Since every application of a dose level and the patient's reaction to it exposes information regarding the medical conditions of the patient, privacy has to be enforced at the level of the intermediate actions too. Indeed, one may propose another definition where the policy only publishes the final recommendation and the intermediate actions are supposed to be "hidden". This boils down to defining different trust barriers.
>
> In our definition, the adversary may look inside the execution of the policy and see the intermediate actions as well. This can be seen as a version of pan privacy [1] and thus would protect against intrusions too. Thus, it is clear that our definition provides better privacy protection and is well-motivated. On the other hand, we agree that an interesting future question may be to see what could be gained in sample complexity if only the final recommendation is published. We would mention it as a possible future work.
>
>
> **Comparison to [Rio et al 2023].** We thank the reviewer for pointing us to [Rio et al 2023]. At the time of the submission, we could not be aware of this concurrent work which was published in ICML this year, i.e. two months after the submission for NeurIPS. On the other hand, [Rio et al 2023] studies a different setup, with multiple agents, while our work is on a single agent. The proposed algorithm in [Rio et al 2023] is a generalisation of DP-SE from single to multi-agent. Since we provide a full theoretical and experimental comparison of our algorithm AdaP-TT compared to DP-SE, we already present the comparison with the single-agent version of the algorithm of [Rio et al 2023]. Here, we provide a statement that we would like to add in the Related Work section while referring to [Rio et al 2023]:
>
> "
> [Rio et al 2023] also studies privacy for best arm identification under fixed confidence but for multiple agents. They propose and analyse the sample complexity of DP-MASE, a multi-agent version of DP-SE. They show that multi-agent collaboration leads to better sample complexity than independent agents, even under privacy constraints.
> "
>
> Hope we have addressed your concerns. Let us know if you have any further comments.
>
> [1] Dwork, Cynthia, et al. "Differential privacy under continual observation." Proceedings of the forty-second ACM symposium on Theory of computing. 2010.

---

> > ### Comment · Reviewer_NTgq · 2023-08-19
> > **Thanks for the feedback**
> >
> > Thank you very much for your detailed feedback.
> > I am grateful for the authors to discuss the theoretical and empirical comparison with Multi-Agent DP-BAI-FC [Rio et al 2023].
> > Also, I acknowledge the motivation for DP-BAI problem in real-world scenarios.
> > I read other reviewers' points and believe that this submission is in good form.

---

### Author Rebuttal · Authors · 2023-08-09

We would like to thank the reviewers for acknowledging the strengths and soundness of the contribution as well as for their thoughtful comments and efforts towards improving the manuscript. In the following, we address the general concerns of reviewers that were common.

As pointed out in the reviews, our main goal is to propose a thorough examination of Best Arm Identification under Fixed Confidence with Differential Privacy, i.e. DP-FC-BAI.

(a) We provide a lower bound on the sample complexity of any $\delta$-correct BAI algorithm satisfying $\epsilon$-global DP. This lower-bound quantifies the additional cost of privacy through a novel information theoretical quantity, i.e. the Total Variation Characteristic Time ($T^\star_{TV}$). As also pointed out in the reviews, *this is the first lower bound in the literature on DP-FC-BAI*.

(b) We also propose an $\epsilon$-global DP variant of the Top Two algorithm, called AdaP-TT. We provide its asymptotic sample complexity and show that AdaP-TT enjoys both theoretical near-optimality and good experimental performance.

Now, we would like to address two recurring comments.

**Novelty of the results and technical contributions.** We resolve three layers of technical challenges through our contributions.

**1. Lower bound.**
(a) To derive the lower bound, we first provide an $\epsilon$-global DP version of the transportation lemma (Lemma 1). To prove this lemma, we construct a sequential coupling between the probability distribution of the triplet $(\tau, \underline{A}, \hat{A})$ when the BAI strategy interacts with two different environments. Here, $\tau$ is the stopping time, $\underline{A}$ is the sequence of action recommended and $\hat{A}$ is the recommended final action. Extra care is needed *to deal with the (random) stopping times in the coupling, compared to a fixed horizon $T$ in regret minimization*. This is a result that could be of general use beyond our work. We combine this lemma with a data-processing inequality to prove the final lower bound.
(b) We also provide a full characterisation of the lower bound by studying the TV characteristic time (Proposition 1), which requires solving exactly the optimization problem defining $T^\star_{TV}$. This provides a closed form expression of $T^\star_{TV}$ with respect to the mean rewards of arms $(\mu_a)_{a = 1}^K$.

**2. Algorithm design.** We propose a *generic wrapper which adapts the existing FC-BAI algorithm to tackle DP-FC-BAI*. It builds on two components.
(a) Adaptive episodes with per-arm doubling and forgetting, which yields the $\epsilon$-global DP property of the algorithm (Theorem 3).
(b) A private GLR stopping rule, obtained by plugging in private empirical means in the non-private GLR stopping rule used of FC-BAI with Gaussian distributions.

To prove the $\delta$-correctness of this choice (Theorem 4), we need to combine concentration results for Bernoulli distributions and Laplace distributions, and account for per-arm phase indices which are random variables. As a result, *the dependency in $(n,\delta)$ of the obtained stopping threshold is fundamentally different than the ones obtained in FC-BAI* (see lines 309--314). This has a profound impact on the asymptotic upper bound on the expected sample complexity.

To use this proposed wrapper, one can choose among the numerous existing sampling rules to tackle FC-BAI. In this work, we considered the Top Two algorithms since they have good theoretical guarantees and empirical performance.

**3. Upper bound.**
(a) We highlight exactly the cost of each algorithmic design choice (doubling, forgetting, the private cost in the GLR stopping) and how they affect the final asymptotic sample complexity. Those remarks will hold for most asymptotically ($\beta$-)optimal algorithms when combined with our wrapper.

(b) Building on [JDB+22], we provide a generic analysis of the class of Top Two algorithms when combined with our wrapper.

Compared to [JDB+22], the **key challenge** lies in controlling the per-arm phases. As a consequence, the proof of sufficient exploration had to be adapted. More importantly, in order to obtain convergence towards the optimal allocation, *we had to prove that the phase switches of the arms happen in a round-robin fashion after a large enough time*. This means that an arm switches phase for a second time after all other arms first switch their own phases.

**Non-asymptotic sample complexity of AdaP-TT.**
In the *non-private FC-BAI literature* (i.e. $\epsilon = + \infty$), *there is no tight lower bound in the non-asymptotic regime (i.e. for any value of $\delta$)*. This is the main open problem in FC-BAI. Therefore, it is also an open problem in DP-FC-BAI. In the class of asymptotically ($\beta$-)optimal algorithms, few have non-asymptotic guarantees.
[JD22] proposed TTUCB and showed the first non-asymptotic upper bound on the expected sample complexity of a Top Two algorithm.
Adapting the non-asymptotic analysis of [JD22] to the private AdaP-TT is an interesting direction for research.
Based on our understanding of their proof, we conjecture that such an adaptation is possible (up to technicalities).
Intuitively, in Theorem 2.4 in [JD22], the $\delta$-dependent term will be modified by adding a concentration term linked to Laplace distribution to the $\log(n)$ term stemming from sub-Gaussian concentrations.
Similar to our asymptotic analysis, this will have consequences on the dependency in the upper bound.
For the $\delta$-independent terms in Theorem 2.4 in [JD22], they will also be impacted since Lemma 3.2 in [JD22] should be modified to account for Laplace noise.
In the low-privacy regime, we expect also to lose at least a multiplicative four-factor compared to TTUCB due to doubling and forgetting.

Hope this commentary addresses the concerns regarding technical novelty and challenges. Below, we address the comments specific to each reviewer by responding to them directly.

---

### Decision · Program_Chairs · 2023-09-21

**Decision:**

Accept (poster)

**Comment:**

Although some reviewers found that the paper is dense and hard to parse, others (experts in the area) mentioned that the paper contains sufficient contributions to be published. In particular, the authors derive a hardness result giving a lower bound for the problem of BAI with fixed confidence under $\varepsilon$-global Differential Privacy. For the high-privacy regime, the lower bound depends on the Total Variation Characteristic Time, which shares a similar spirit with the regret minimization with $\varepsilon$-global DP [AB22]. For low-private regimes, the lower bound reduces to the classical no-private lower bound. This is a neat observation. The algorithmic contributions based on top-two algorithms are also novel.

For the final version of the paper, it would be ideal if the authors can make the presentation more reader-friendly, especially to non-experts.